

# Optimisation of the World Ocean Model of Biogeochemistry and Trophic-dynamics (WOMBAT) using surrogate machine learning methods.

Pearse J. Buchanan[1,2], P. Jyoteeshkumar Reddy[1], Richard Matear[1], Matthew A. Chamberlain[1], Tyler
Rohr[2,3], Dougal Squire[1,4], and Elizabeth H. Shadwick[1,2].

[1]CSIRO Environment, Hobart, 7004, Australia.
[2]Australian Antarctic Program Partnership, Hobart, 7000, Australia.
[3]Institute for Marine and Antarctic Studies, University of Tasmania, Hobart, 7004, Australia.
[4]ACCESS-NRI, Australian National University, Canberra, 2601, Australia.

*Correspondence to*: Pearse J. Buchanan (pearse.buchanan@csiro.au)

**Abstract.** The introduction of new processes in biogeochemical models brings new model parameters that must be set. Optimisation of the model parameters is crucial to ensure model performance based on process representation, rather than poor parameter values. However, for most biogeochemical models, standard optimisation techniques are not viable due to computational cost. Typically, (tens of) thousands of simulations are required to accurately estimate optimal parameter

values of complex non-linear models. To overcome this persistent challenge, we apply surrogate machine learning methods to optimise the model parameters of a new version of the World Ocean Model of Biogeochemistry and Trophic dynamics (WOMBAT), which we call WOMBAT-lite. WOMBAT-lite has undergone numerous updates described herein with many new model parameters to prescribe. A computationally inexpensive surrogate machine learning model based on Gaussian Process Regression was trained on a set of 512 simulations with WOMBAT-lite. These simulations explored model fidelity

to 8 observation-based target datasets by varying 26 uncertain parameters across their *a priori* ranges. The surrogate model, trained on these 512 simulations, facilitated a global sensitivity analysis to identify the most important parameters and facilitated Bayesian parameter optimisation. Our approach returned optimal posterior distributions of 13 important parameters that, when input to WOMBAT-lite, ensured excellent fidelity to the target datasets. This process improved the representation of chlorophyll-a concentrations, air-sea carbon dioxide fluxes and patterns of phytoplankton nutrient

limitation. We present an optimal parameter set for use by the modelling community. Overall, we show that surrogate-based calibration can deliver optimal parameter values for the biogeochemical components of earth system models and can improve the simulation of key processes in the global carbon cycle.



## 1 Introduction

Ocean biogeochemical models are crucial tools for unravelling the complex interactions between the physical transport of properties, the chemical reactions of compounds and the biological conversions between inorganic and organic matter (e.g., Fennel et al., 2022). They are key for understanding and quantifying the impact of climate change on ocean ecosystems and biogeochemical cycles. This includes both the natural pulses of climate variation, such as the El Nino Southern Oscillation, and the pervasive long-term climate change, such as that induced by accumulating greenhouse gas emissions. For instance,

ocean biogeochemical models are used to estimate the ocean's uptake of carbon dioxide ($CO_2$) (Doney et al., 2003; Friedlingstein et al., 2023; Joos et al., 2013; Orr et al., 2001; Terhaar et al., 2024), to understand the controls on interior oxygen concentrations (Buchanan and Tagliabue, 2021; Oschlies et al., 2018), to quantify changing volumes of oxygen minimum zones (Busecke et al., 2022), for projecting change in ocean primary productivity (Kwiatkowski et al., 2020; Tagliabue et al., 2021), to evaluate shifts in marine ecosystem community composition (Cael et al., 2021a; Follows et al.,

2007) and fisheries production (Lotze et al., 2019; Stock et al., 2017), and most recently to evaluate the efficacy of marine $CO_2$ removal strategies (Fennel et al., 2023; Kwiatkowski et al., 2023; Siegel et al., 2021).

At their core, ocean biogeochemical models include an ecosystem component. This component represents the growth of phytoplankton via uptake of nutrients and photosynthesis, their mortality via zooplankton grazing and respiration, and the

routing of dead biomass from both phytoplankton and zooplankton to detritus. The detritus sinks through the water column and is acted on by heterotrophic remineralisation to return the organic matter to the inorganic nutrients from which phytoplankton biomass was initially constructed. This ecosystem component, at its simplest, is known as a nutrient-phytoplankton-zooplankton-detritus (NPZD) model (e.g., Fennel et al., 2022). Other components may accompany it, such as those that encode the chemical reactions of the carbon system (Orr et al., 2017), exchanges with external reservoirs (i.e.,

rivers, sediments, and atmosphere), trace metals (Tagliabue et al., 2023), isotopes (Buchanan et al., 2021), or biogenic aerosols (Gantt et al., 2012). Some models consider different types of nutrients, phytoplankton, zooplankton and detritus, with some including dozens of types defined by distinct traits and/or sizes (Follett et al., 2022; Follows and Dutkiewicz, 2011; Serra-Pompei et al., 2022). Whether simple or complex, a defining feature of ocean biogeochemical models is their ecosystem component, which controls how elements cycle between inorganic and organic phases.

Despite their critical applications, the construction of biogeochemical models suffers from numerous sources of uncertainty. Model simulations of air-sea fluxes of $CO_2$, for instance, suffer from considerable seasonal biases, particularly in the Southern Ocean (Hauck et al., 2020) due, in part, to biases in the phasing and magnitude of biological activity (Mongwe et al., 2018). These biases stem from poor mechanistic understanding of the processes being modelled, the complex interplay of



those processes and a lack of observational constraint (Denman, 2003; Fennel et al., 2022; Matear, 1995; Rohr et al., 2023; Ward et al., 2010). However, even if our understanding and observational network were complete, there exist many tuneable and potentially inter-dependent parameters that control many target outcomes (air-sea $CO_2$ fluxes, nutrient fields, chlorophyll concentrations, etc.) that must be reproduced simultaneously. One optimisation approach has been to reduce the number of processes being represented, both physical and biogeochemical, such that a smaller number of parameters requires

optimisation (DeVries and Weber, 2017; Holzer and Primeau, 2013). While this approach has skill for reproducing the ocean's large-scale fields in an equilibrium state, it arguably has less skill in emulating the many inter-dependent upper ocean processes that operate on higher frequencies. Optimising a model with these higher frequency processes would ideally involve: (1) a global sensitivity analysis that identified the most important parameters, followed by (2) a Bayesian optimisation procedure to constrain their optimal values from within their *a priori* ranges. Being Bayesian, the optimal

parameter values would be taken from posterior distributions, recognising that many "optimal" parameter combinations are possible. Even so, the sheer number of parameters, their non-linear interactions and an objective function composed of many targets (i.e., trying to reproduce many features at once) makes this approach impossible without large and typically unfeasible computational costs.

Machine learning techniques now offer a means to overcome this key challenge (Reddy et al., 2024b, a). Synthetic output may be generated by a surrogate machine learning model trained on a smaller set of real model output. The surrogate machine learning model is computationally cheap and can generate tens to hundreds of thousands of samples required for a detailed exploration of the parameter space. Such a high number of samples is critical to identify both the first-order and interactive effects of different parameters (Reddy et al., 2024b; Saltelli et al., 2019), as well as a means to undergo Bayesian

optimisation (Reddy et al., 2024a). The surrogate-based calibration has been successful with physical and terrestrial biosphere components of climate system models (Li et al., 2018; Reddy et al., 2024a; Xu et al., 2022, 2018), but its application to marine biogeochemical components is in its infancy.

In this study, we optimise version "lite" of the World Ocean Model of Biogeochemistry And Trophic dynamics (WOMBAT-
lite) using surrogate machine learning techniques (Fig. 1). This surrogate approach is crucial. Although WOMBAT-lite has few tracers and is computationally efficient, making it viable for high resolution configurations (Kiss et al., 2020; Matear et al., 2015; Menviel and Spence, 2024; Oke et al., 2013) and large ensembles (Mackallah et al., 2022; Rashid, 2022; Ziehn et al., 2020), it is nonetheless a global, three-dimensional, biogeochemical model. This makes it computationally demanding enough to prevent parameter calibration via traditional techniques. Second, surrogate-based optimisation has been

successfully applied to physical and terrestrial components of climate models (Li et al., 2018; Reddy et al., 2024a; Xu et al., 2022, 2018) and so offers real potential for, but has not been widely applied to, biogeochemical models. Finally, biogeochemical models are constantly undergoing major updates and new process development. The need for efficient and accurate optimisation is therefore constant. As an apt example, we made major updates during this study (detailed in



Appendix A) and focussed on improving Southern Ocean air-sea $CO_2$ fluxes, which, shows persistent biases in ocean

biogeochemical models (Hauck et al., 2020, 2023; Mongwe et al., 2018).

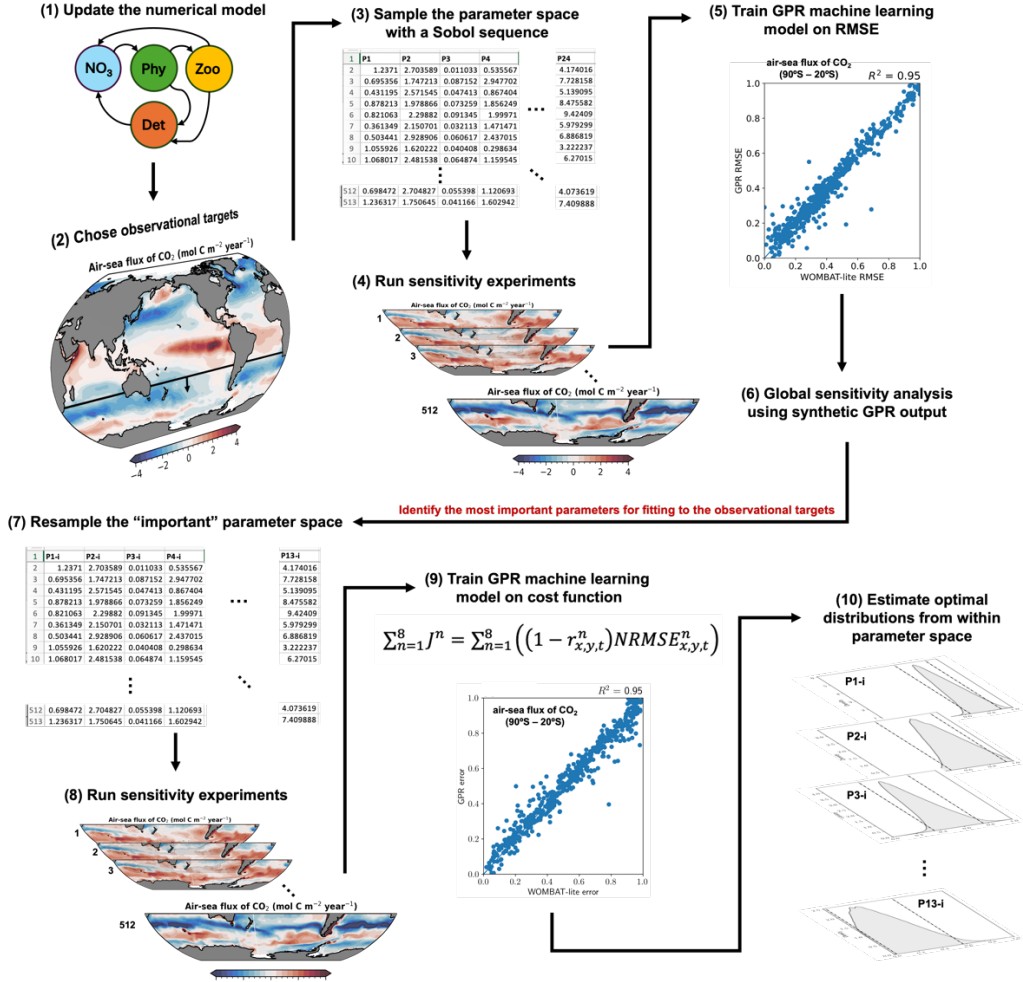

**Figure 1. Flow chart representation of the methodology.** (1) Any updates to the numerical model, in this case a biogeochemical model, are finalized. Because the model has been altered, it requires optimization. (2) The target observations are chosen against which the model

performance will be assessed and, eventually, optimized. (3) Sobol sequencing of the full parameter space selects 512 unique parameter sets from *a priori* ranges of the uncertain parameters. In this case, we chose 24 uncertain parameters. (4) The 512 unique parameter sets are used to run the numerical model forward to obtain 512 "simulated" solutions for each of the target observations. (5) Using a metric of model performance, in this case the root mean square error (RMSE), we train a surrogate machine learning model based on Gaussian Process Regression (GPR) to synthetically reproduce the model performance (RMSE) given a parameter set as input. (6) With a the GPR model we

create thousands of synthetic model simulations to conduct a global Sobol sensitivity analysis. This tells us what the most important parameters are for model performance against our observational targets. (7) We resample the parameter space using only the most important parameters and (8) run a new set of simulations with these unique parameter sets. (9) The GPR model is then trained to reproduce the global cost-function (denoted "error" above), which accounts for the performance of the model across all observational targets, given a unique parameter set as input. (10) Thousands of synthetic cost-function results, estimated by the GPR model, are used to perform Bayesian

optimization that solves for the optimal posterior distributions of the important parameters from within the *a priori* ranges.





## 2. Methods

### 2.1 Optimisation Summary

We have developed a new ocean biogeochemical model called WOMBAT-lite (Fig. 2). These updates (detailed below) necessitated a thorough sensitivity analysis and optimisation to a chosen set of observations. We performed Sobol sensitivity

analysis, which gave an understanding of which parameters were most important to the model outcomes, followed by Bayesian optimisation to fine-tune parameter values and improve model accuracy. However, both Sobol sensitivity analysis and Bayesian optimisation require many thousands of samples to be reliable. To overcome this challenge, we employed a machine learning model based on Gaussian Process Regression to act as a surrogate of WOMBAT-lite. This computationally inexpensive surrogate, trained on hundreds of real simulations with WOMBAT-lite, was able to produce large samples of

synthetic results that enabled sensitivity analysis optimisation (Fig. 1).

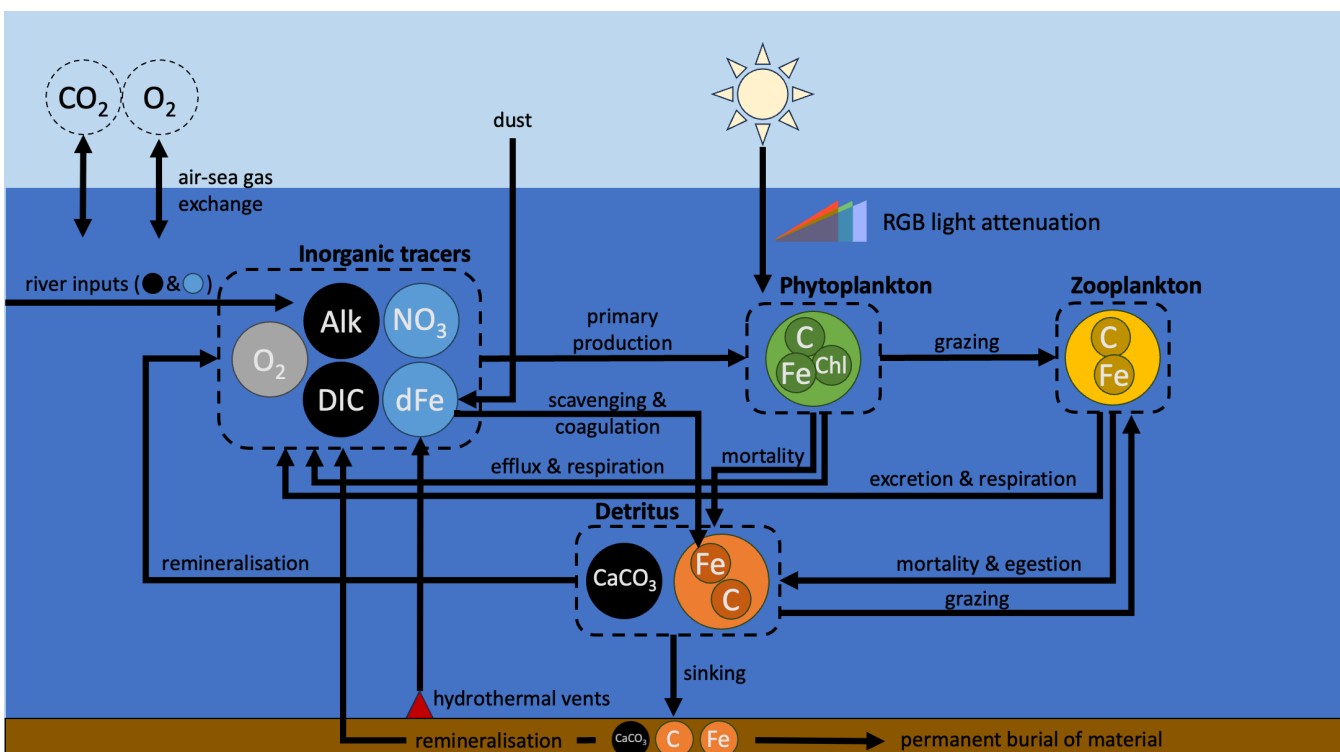

**Figure 2. Schematic representation of WOMBAT-lite.** Tracers and biomass pools are represented by circles of different colours. Black tracers represent the carbon system, blue are inorganic nutrients. Inner circles of C, Chl and Fe within each biomass pool represent the units

of carbon, chlorophyll and iron that are explicitly tracked. Major components are grouped within the dashed outlines. Although dissolved inorganic carbon (DIC), alkalinity (Alk) and oxygen ($O_2$) are connected to primary production, they are only affected by it and do not limit primary production of phytoplankton. In contrast, both nitrate ($NO_3$) and dissolved iron (dFe) are biogeochemical tracers whose availability both controls and is affected by phytoplankton growth. Dust and hydrothermal iron input dFe, while rivers input DIC, Alk, $NO_3$ and dFe. Atmospheric concentrations of carbon dioxide ($CO_2$) and $O_2$ are not explicitly tracked by the ocean model (dashed circles).






## 2.2 Model development summary

WOMBAT-lite (Fig. 2) considers 13 tracers: two nutrients, being nitrate ($NO_3$) and dissolved iron (dFe), the carbon system (dissolved inorganic carbon (DIC), alkalinity (Alk), calcium carbonate ($CaCO_3$)), oxygen ($O_2$), the biomass pools of one phytoplankton ($B_p$), one zooplankton ($B_z$) and one sinking detrital ($B_d$) functional type, prognostic chlorophyll ($B_p^{Chl}$), as

well as biogenic iron in phytoplankton ($B_p^{Fe}$), zooplankton ($B_z^{Fe}$) and sinking detritus ($B_d^{Fe}$). In WOMBAT-lite, photosynthetically active radiation (PAR) is split into three wavelength bands associated with blue, green and red light, and each of these bands attenuates differently through the water column. The explicit chlorophyll content of phytoplankton allows for photoacclimation and the formation of deep chlorophyll maxima. The attenuation of blue, green and red light is affected by chlorophyll concentrations, and we implicitly account for the "packaging effect" by assuming a positive

relationship between chlorophyll concentration and community mean cell size, where larger cells have less effect on light absorption. Phytoplankton limitation by nutrients is affected by an implicit positive relationship between cell size and cell density, where larger cells have less affinity for nutrients. Limitation by iron is modelled via variable quotas, allowing for luxury uptake in dFe-rich conditions and the export of Fe-rich detritus. Phytoplankton increase their iron requirements as their intracellular quota of chlorophyll increases, generating a co-limitation of light and iron on growth. Cycling of dFe now

explicitly considers free, ligand-bound and colloidal iron, and is lost via nanoparticle formation, scavenging and colloidal coagulation. Zooplankton grazing is via a type III disk formulation that substantially dampens the temperature effect on grazing activity, aligning with rapid consumption rates in polar and tropical waters alike. The sinking of detritus is spatiotemporally variable and is dependent on phytoplankton biomass, emulating community shifts in mean cell size and bloom conditions, and depth, emulating a power law rather than an exponential decay associated with acceleration due to

packaging with increasing pressure. A fraction of the detritus (and $CaCO_3$) reaching the sediment is now permanently buried. In addition, WOMBAT-lite considers inputs of nitrate, dissolved inorganic carbon and alkalinity from rivers, simplistic sedimentary denitrification and nitrogen fixation routines, and the flux of dissolved iron from hydrothermal vents. All biogeochemical cycles in WOMBAT-lite are therefore open and their inventories can change. The basic unit of biomass is carbon with a fixed $C:N:O_2$ of 122:16:-172. A full description of WOMBAT-lite is in Appendix A.


## 2.3 Model experiments and evaluation

### 2.3.1 Observational target fields for assessment

We use 8 observational products/databases to assess the performance of WOMBAT-lite (Fig. 3) including the gridded, global products of surface nitrate (Garcia et al., 2024b) and dissolved iron (Huang et al., 2022). While nutrient distributions

are useful, they have limited power by themselves for assessing biogeochemical models, since similar distributions of nutrients can be achieved for different rates of phytoplankton growth and recycling (Fennel et al., 2022). Remotely sensed chlorophyll is an important constraint that can be considered a proxy for the total stock of phytoplankton biomass and



features heavily in biogeochemical model evaluation (Fennel et al., 2022). We use the Copernicus three-dimensional
chlorophyll-a product that combines remotely-sensed, hydrographic and BGC-Argo measurements of fluorescence to
generate a depth-resolved climatology of chlorophyll (Sauzède et al., 2015). The extension of chlorophyll to depth allows for
an assessment of patterns in the vertical, including spatiotemporal variations in the position of the deep chlorophyll
maximum. Other important observations include a gridded and global product of air-sea $CO_2$ fluxes for the year 1985 (Chau
et al., 2022), the earliest year available, and vertically integrated net primary production (Westberry et al., 2008). Because
the carbon-based productivity model (CbPM; Westberry et al., 2008) is based on backscatter and explicitly accounts for
growth rates separate from biomass its patterns are more orthogonal to chlorophyll than the Vertically Generalized
Production Model (VGPM; Behrenfeld and Falkowski, 1997), and so offers greater potential as an independent constraint on
model performance (Westberry et al., 2023). In addition to these gridded products, we also use a database of the primary
limiting nutrient for phytoplankton growth (Browning and Moore, 2023) and sediment trap records of the sinking flux of
detrital particles through the ocean interior (Mouw et al., 2016).


While we recognise that these datasets are themselves subject to uncertainty, their combination allows for a powerful model
assessment. Furthermore, our assessment focusses on the reproduction of large-scale, seasonal patterns. It is also worth
noting that having too many target fields compounds the difficulty associated with parameter optimisation, while having too
few risks poor performance in unconsidered targets. Target fields must therefore be chosen carefully. All gridded datasets as
well as the particle flux database are resolved on a global 1º by 1º degree grid and on a monthly temporal resolution, while
the primary limiting nutrient for phytoplankton, due to data scarcity, is represented annually (unchanging in time).

**2.3.2 Sensitivity experiments for evaluation and surrogate model training**

Our first goal was to understand which parameters in WOMBAT-lite were most important for model performance. We
undertook 512 simulations that each sampled randomly from predefined ranges of 24 key parameters related to the
ecosystem component of the model (Table 1; Fig. 1). Each experiment carried a unique biogeochemical parameter set that
altered the biogeochemical behaviour, but all experiments had identical physical conditions and initial conditions. Physical
fields were initialised from a previous spin up with the same ocean model (Kiss et al., 2020) forced by JRA55-do (Tsujino et
al., 2018). We used a repeat "normal" year forcing of the JRA55-do to avoid inter-annual variability and extremes in climate
modes (Stewart et al., 2020). Atmospheric $CO_2$ was maintained at 315.2 ppm (i.e., levels at calendar year 1958). Nitrate,
dissolved iron, dissolved oxygen, dissolved inorganic carbon and alkalinity fields were initialised from globally gridded
datasets at the month of December (Garcia et al., 2024b, a; Huang et al., 2022; Lauvset et al., 2016). Concentrations of
phytoplankton, zooplankton, detritus, and calcium carbonate were initialised at globally homogenous values of 0.1 mmol C
m$^{-3}$. Fe:C ratios of phytoplankton, zooplankton and detritus were initialised at 7 µmol per mol. Chlorophyll was initialised at
0.004 mg per mg of phytoplankton carbon biomass.





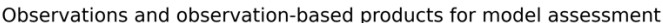

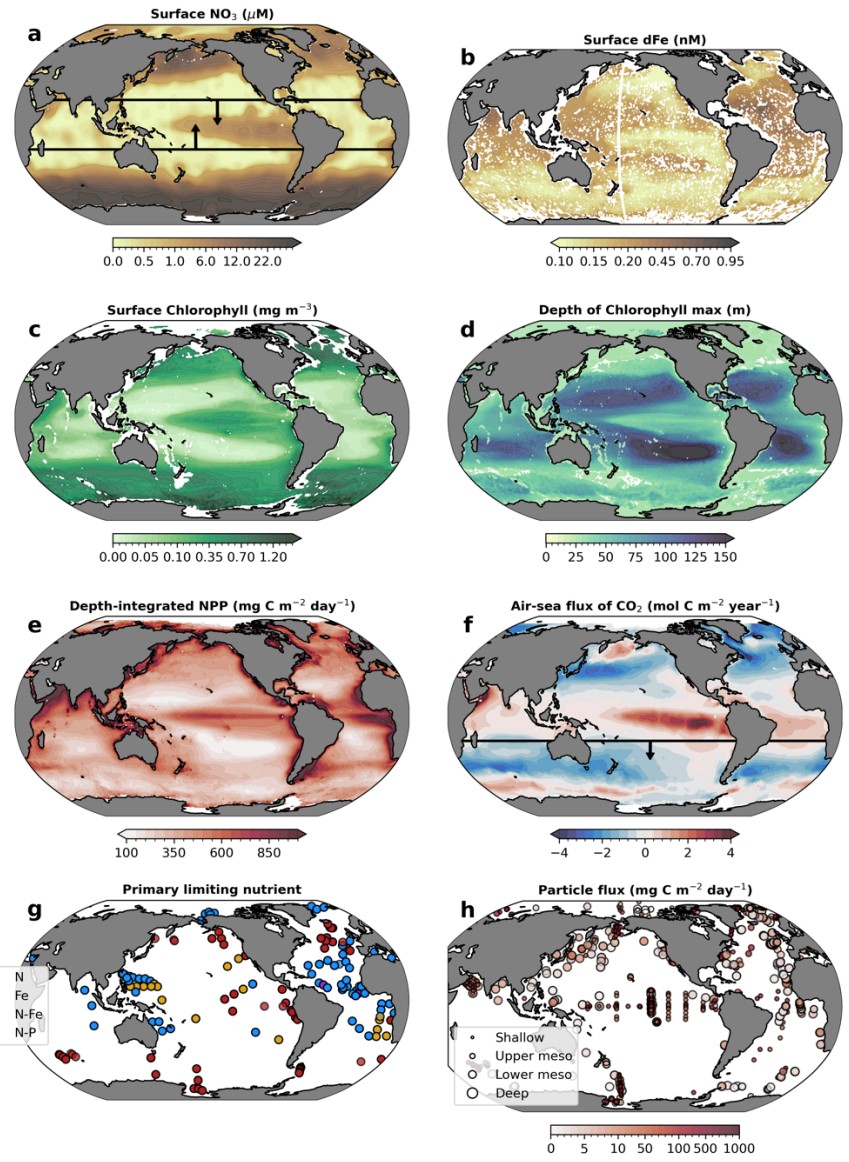


**Figure 3. Observation-based target fields used for assessment (sensitivity analysis and optimisation) of WOMBAT-lite.** Annual means are shown here for illustration purposes, but monthly resolution was used for all products except the Primary Limiting Nutrient dataset. Boxes in maps of surface nitrate ($NO_3$) and the air-sea flux of carbon dioxide ($CO_2$) encapsulate the specific regions of focus when assessing model performance and optimization, being between 20ºS and 20ºN for surface $NO_3$ and south of 20ºS for air-sea $CO_2$ fluxes. $CO_2$ fluxes 205 out of the ocean are positive.




**Table 1. Key ecosystem parameters for WOMBAT-lite and their predefined ranges for ocean-only experiments.** Parameter values for other configurations, include for the Earth System Model (ACCESS-ESM1.6), are available at https://github.com/ACCESS-NRI.

| Component | Parameter | *a priori* range | Default (optimal range) | Description | Units |
|---|---|---|---|---|---|
| Phytoplankton | $\alpha_a$ | 0.25 - 1.25 | 1.0 (0.89 – 1.16) | Scaler control on phytoplankton maximum growth rates | day$^{-1}$ |
| | [i]$\beta_a$ | 1.040 - 1.080 | 1.050 (1.041 – 1.063) | Base for temperature-dependent autotrophy | - |
| | $PI^0$ | 1.5 - 3.0 | 2.25 | Initial slope of the photosynthesis-irradiance curve | (W m$^{-2}$)$^{-1}$ (mg Chl m$^{-3}$)$^{-1}$ |
| | [ii]$B_p^{thresh}$ | 0.01 - 1.0 | 0.6 (0.48 – 0.94) | Biomass threshold of phytoplankton for implicit allometric scaling | mmol C m$^{-3}$ |
| | $K_p^{N_0}$ | 0.01 - 3.0 | 2.0 (1.04 – 2.30) | Half-saturation coefficient for nitrogen uptake | mmol N m$^{-3}$ |
| | $K_p^{dFe_0}$ | 0.01 - 3.0 | 2.5 (2.07 – 2.97) | Half-saturation coefficient for dissolved iron uptake | μmol Fe m$^{-3}$ |
| | $Q_p^{"\frac{Fe}{C}}$ | 20 - 100 | 50 (39 – 64) | Maximum Fe:C quota of the cell | μmol Fe (mol C)$^{-1}$ |
| | $Q_p^{*\frac{Fe}{C}}$ | 4 - 15 | 10 | Optimal Fe:C quota of the cell | μmol Fe (mol C)$^{-1}$ |
| | $Q_p^{'\frac{Chl}{C}}$ | 0.001 - 0.01 | 0.004 | Minimum Chl:C quota of the cell | mg Chl (mg C)$^{-1}$ |
| | $Q^{*\frac{Chl}{C}}$ | 0.02 - 0.06 | 0.036 (0.020 – 0.038) | Optimal Chl:C quota of the cell | mg Chl (mg C)$^{-1}$ |
| | $\gamma_p^0$ | 0.01 - 0.10 | 0.01 (0.010 – 0.016) | Linear mortality rate of phytoplankton | day$^{-1}$ |
| | $\Gamma_p^0$ | 0.01 - 0.10 | 0.05 | Quadratic mortality rate of phytoplankton | (mmol C m$^{-3}$)$^{-1}$ day$^{-1}$ |
| Grazing | $g_h$ | 2.0 - 4.0 | 3.0 | Scaler control on maximum zooplankton grazing rate | day$^{-1}$ |
| | $\varepsilon$ | 0.05 - 1.5 | 0.05 (0.05 – 0.15) | Zooplankton prey capture rate coefficient | m$^6$ (mmol C)$^{-2}$ day$^{-1}$ |
| | [iii]$\phi_z^p$ | 1.0 | 1.0 | Preference of zooplankton for phytoplankton | - |
| | $\phi_z^d$ | 0.01 - 0.50 | 0.25 | Preference of zooplankton for detritus | - |
| | [iii]$\lambda$ | 0.6 | 0.6 | Zooplankton assimilation efficiency | - |
| | $\gamma_z^0$ | 0.01 - 0.10 | 0.05 | Linear mortality of zooplankton (respiration) | day$^{-1}$ |
| | $K_z^Y$ | 0.01 – 0.5 | 0.25 | Half-saturation coefficient of zooplankton mortality | mmol C m$^{-3}$ |
| | $\Gamma_z^0$ | 0.1 - 1.0 | 0.9 (0.61 – 0.99) | Quadratic mortality rate of zooplankton (predation) | (mmol m$^{-3}$)$^{-1}$ day$^{-1}$ |



| Component | Parameter | *a priori* range | Default (optimal range) | Description | Units |
|---|---|---|---|---|---|
| | $\beta_h$ | 1.060 - 1.080 | 1.065 (1.060 − 1.075) | Base for temperature-dependent heterotrophy | - |
| Detritus | $\omega_d^0$ | 5 - 20 | 18 (12.7 − 19.9) | Scaler to sinking speed of detritus | m day$^{-1}$ |
| | $\omega_d^{max}$ | 20 - 50 | 35 | Maximum sinking speed of detritus | m day$^{-1}$ |
| | $\gamma_d^0$ | 0.025 - 0.1 | 0.09 (0.064 − 0.099) | Linear rate of (implicit) bacterial remineralisation | day$^{-1}$ |
| | $R_{\frac{CaCO_3}{detritus}}$ | 0.01 – 0.15 | 0.050 | CaCO$_3$ to organic detrital ratio | mol C / mol C |
| | $\omega_{CaCO_3}^0$ | 3 - 10 | 6.0 | Scaler to sinking speed of CaCO$_3$ | m day$^{-1}$ |
| | $\gamma_{CaCO_3}^0$ | 0.0005 - 0.01 | 0.01 | Scaler control on (implicit) CaCO$_3$ dissolution rate | day$^{-1}$ |
| Iron cycling | [iii]$Lig$ | 0.7 | 0.7 | Concentration of Fe-binding organic ligand | µmol m$^{-3}$ |
| | [iii]$K_{nanop}^{Fe}$ | 0.01 | 0.01 | Precipitation of $Fe'$ as nanoparticles (in excess of solubility) | day$^{-1}$ |
| | [iii]$K_{scav}^{Fe}$ | 0.00005 | 0.00005 | Scavenging of $Fe'$ onto biogenic particles | (mmol C m$^{-3}$)$^{-1}$ day$^{-1}$ |
| | [iii]$K_{coag}^{Fe}$ | 0.0001 | 0.0001 | Coagulation of dissolved Fe into colloidal Fe | (mmol C m$^{-3}$)$^{-1}$ day$^{-1}$ |

[i]Parameter variations not included in initial sensitivity experiments for sensitivity analysis (steps 3-6 in Fig. 1). Only in optimization (steps 7-10 in Fig. 1).

[ii]Parameter range was set equal to 0.01 to 0.1 in the initial sensitivity experiments for sensitivity analysis (step 3-6 in Fig. 1).

[iii]Parameter space not explored.






We chose to run the experiments for only 10 years, making a total of 5120 model years and at a nominal horizontal resolution of 1º. This short timescale was enough to assess the skill of the biogeochemical model, at least regarding its ecosystem component. Marine phytoplankton contribute half of all primary production in the Earth system (Field, 1998) but

represent less than 1% of photosynthetic biomass (Friedlingstein et al., 2023; Le Quéré et al., 2005), meaning that they turn over quickly. Changes to key parameters within the ecosystem component therefore result in a rapid realisation of different patterns in biological states (e.g., chlorophyll and net primary production, among others). Our analyses and optimisation thus focus on the ecosystem component using 10-year model runs. We do acknowledge that longer-term, low frequency modes of variation exist in biogeochemical models, and to partially address this we completed 100-year simulations with optimal

parameter sets. However, we also note that longer integrations risk the compounding of physical and biogeochemical model errors.

### 2.3.3 Measures of performance

Output from the final year of the experiments (year 10) was compared directly to the target datasets (Fig. 3). Univariate measures of performance were calculated, including the correlation coefficient (Pearson's), root mean square error, global

mean bias and the normalised standard deviation (Stow et al., 2009). These were calculated across all grid cells and time points. For surface nitrate concentrations, we only calculated these statistics between 20ºS to 20ºN to focus on achieving a realistic transition of higher concentrations to lower concentrations from the equatorial to subtropical biome. We stress that fidelity in extra-tropical regions was captured independently via the limiting nutrient dataset. Also, surface nitrate in the equatorial region is highly responsive to changes in the ecosystem component due to warmer temperatures that accelerate

metabolism. After only 10 years our simulations diverged most in this region. Additionally, surface nitrate concentrations were $\log_{10}$ transformed prior to calculating the measures of performance due to a skewed distribution towards very low values. This substantially improved the ability of our optimisation approach to select parameters that reproduced the high nitrate tongue in the equatorial region (see below). For air-sea fluxes of $CO_2$, we assessed model performance exclusively in the Southern Ocean south of 20ºS. This region was also highly sensitive to the parameterisations of the model, showing

positive or negative fluxes depending on our parameter set and aligning with findings that biological activity in this region is of high importance for model skill (Mongwe et al., 2018).



## 2.4 Details of the sensitivity analysis and model optimisation

### 2.4.1 Global sensitivity analysis

Sensitivity analysis (SA) methods are broadly categorized into local and global approaches. Local SA examines how small perturbations in parameters around certain reference points affect outputs, making it computationally feasible and widely used (Rakovec et al., 2014). However, for models where parameters interact or have non-linear impacts on the outputs, local SA can introduce substantial bias, underestimating parameter importance (Saltelli et al., 2019). The model parameters in this study are anticipated to exhibit complex, non-linear interactions influencing the outputs (Denman, 2003; Fennel et al., 2022;

Matear, 1995; Ward et al., 2010), thus justifying the use of global SA to capture these dynamics accurately. One of the most effective global SA methods, Sobol sensitivity analysis, is widely adopted due to its precision in addressing interaction effects, discontinuities, and non-linear influences of parameters on model outputs (Baki et al., 2022; Reddy et al., 2024b). Based on the Hoeffding-Sobol decomposition, Sobol SA leverages Analysis of Variance (ANOVA) to decompose output variance into contributions from individual parameters, interactions between parameter pairs, and so on across increasing

levels of dimensionality (Saltelli et al., 2010; Sobol′, 2001). Sobol sensitivity indices, representing the importance of parameter interactions in the context of total output variance, are then computed by evaluating ratios of these variances. To perform Sobol SA on the WOMBAT-lite outputs requires extensive parameter sampling, a computationally expensive task. To efficiently manage this, we use a surrogate Gaussian Process Regression (GPR) model (Williams and Rasmussen, 1995, 2006), which is trained on a limited number of runs (sensitivity experiments; section 2.3.2; visualised in Fig. 1). The GPR

model, designed for accuracy, provides predictions over a large sample space, allowing SA to be performed without extensive and computationally unfeasible simulations.

The analysis focuses on the target fields detailed in Fig. 3 and section 2.3.1. Initially, a Quasi-Monte Carlo Sobol sequence is applied to generate 512 parameter samples using the Uncertainty Quantification Python Laboratory package (Wang et al.,

2020). Simulations are then run with these parameter samples, and root mean square error (RMSE) values are calculated by comparing the model outputs with observational data in space and time. RMSE is normalized via min-max scaling. A sample size of 512 is selected based on the sample size sensitivity experiments (Fig. S1). Next, the GPR model is trained using these parameter samples as inputs and the normalized RMSE as the output for each observation. K-fold cross-validation (K=8) is used to evaluate the GPR model's accuracy. The data is split into K folds, and the model is trained on K-1 folds, with the

left-out fold serving as the test set. This is repeated across all folds, and predictions are aggregated. The GPR model accuracy is assessed through the goodness-of-fit ($R^2$) metric by comparing GPR predictions with WOMBAT-lite RMSE data, which indicates high accuracy (Fig. S2). Using this validated GPR model, RMSEs for 53,248 new parameter samples (generated via Sobol sequence) are predicted for each target field (all eight), consistent with methods from previous studies





(Baki et al., 2022; Reddy et al., 2024b). Finally, Sobol sensitivity indices are calculated based on these predictions for all
eight target fields, offering insight into the relative influence of each parameter on the ability of WOMBAT-lite to reproduce
the targets.

### 2.4.2 Parameter optimisation

This study uses Gaussian Process Regression-based Bayesian Optimisation (G-BO) (Reddy et al., 2024a) to identify the
optimal parameter distributions so that WOMBAT-lite can best reproduce the eight target fields simultaneously. The process
begins by generating 512 parameter samples via a Quasi Monte-Carlo (QMC) Sobol sequence design implemented through
the Uncertainty Quantification Python Laboratory (UQ-PyL) package (Wang et al., 2020). For the optimisation, we explore
only the parameter space of the most sensitive parameters identified by the global sensitivity analysis (Fig. 1). These 512
sample parameter sets are used as input to WOMBAT-lite, and the model is run forward for 10 years (see above). A sample
size of 512 is selected based on the sample size sensitivity experiments (Fig. S3). Then, the GPR surrogate model, trained on
these 512 samples, predicts a normalized cost function,

$$J = \left(1 - r_{x,y,t}\right) \cdot NRSME_{x,y,t} \,, \tag{1}$$

Where $J$ is the cost function for a given target field, $NRSME_{x,y,t}$ is the normalized root mean square error (scaled by the
max-min) and $r_{x,y,t}$ is the Pearson's correlation coefficient evaluated across all longitude (subscript $x$), latitude (subscript $y$)
and time (subscript $t$) points (time in this case being monthly in resolution). This cost function penalizes poor correlations, as
well as bias and error in the variance because we use an uncentered NRMSE. Rather than optimise for the parameters that
best reproduce each target field in isolation, we chose to optimise to a global cost function

$$\sum_{n=1}^{8} J^n = \sum_{n=1}^{8} \left(\left(1 - r_{x,y,t}^n\right) NRMSE_{x,y,t}^n\right) \,, \tag{2}$$

Where superscript $n$ is the $n^{th}$ target field. Therefore, we aim to select parameter sets that optimise overall model
performance. Using a composite kernel—constant, Matern, and white noise kernels—the GPR model accurately predicts the
normalized cost function values, confirmed by R² scores > 0.8 from K-fold cross-validation (K=8) (Fig. S4). The trained
GPR model is then used to estimate the normalized cost function value for optimisation purposes.

Bayesian optimisation enables iterative learning of optimal model parameters using observational data. Here, a uniform prior
is assumed for parameters, and the normalized cost function value predicted by the GPR model serves as the likelihood
function (Reddy et al., 2024a). Since computing marginal likelihood (p(z)) directly is often complex, Markov Chain Monte
Carlo (MCMC) sampling is employed, which estimates the posterior distribution without explicit calculation of this constant
(Issan et al., 2023). Among MCMC methods, Affine invariant ensemble sampling, implemented using the "emcee" Python
package (Foreman-Mackey et al., 2013), is selected for its efficient convergence properties. This method uses an ensemble



of chains to simplify sampling from anisotropic distributions. Fifty walkers and a stretch move of two are applied, with the first 10,000 steps used as a burn-in phase to ensure convergence, followed by 90,000 additional steps to achieve stable posterior distribution estimates (Foreman-Mackey et al., 2013; Goodman and Weare, 2010).

## 3. Results

### 3.1 Performance

The 512 sensitivity experiments, each with a unique parameter set (Table 1), produced 512 unique realisations of biogeochemical and ecosystem dynamics. We compared year 10 of the experiments at all grid cells and at monthly temporal resolution with the target datasets (i.e., the observations). The skill of these simulations ranged widely (Fig. 4; Table 2). Global surface chlorophyll showed the greatest variation, ranging in correlations from -0.45 to 0.8, followed by the primary limiting nutrient of phytoplankton growth (-0.29 to 0.82) and the depth of the chlorophyll maximum (-0.25 to 0.69). Most

experiments underestimated the variability in surface chlorophyll, and many produced surface chlorophyll concentrations that were low compared with the observation-based product. Unlike surface chlorophyll, there was too much variation in the depth of the chlorophyll maximum and many experiments had chlorophyll maxima that were positioned too deep (i.e., positive bias).

For the air-sea flux of $CO_2$ in the Southern Ocean (20ºS – 90ºS) and the depth-integrated rate of net primary production, our simulations showed a narrow range of correlations between 0 and 0.5. The narrower range potentially reflects the identical physical state across our experiments, which strongly influences air-sea gas exchange and nutrient delivery to the surface. For net primary production, the weaker correlations might reflect significant errors in the observation-based products themselves (Westberry et al., 2023) that limit the potential for agreement. Like chlorophyll, net primary production showed a

negative bias in many experiments and a chronic inability to capture the observed magnitude of variations. The same general underestimation of values and variability was the case for the sinking flux of detritus. No experiment was able to reproduce the observed spatiotemporal variations in sinking detrital flux, although this is perhaps expected given that this dataset captures higher frequency variations in particle flux that are lost in a coarse resolution model.

For surface dissolved iron, the experiments produced correlations ranging from -0.13 to 0.52, and thus the best performing experiments compared well with other biogeochemical models (Huang et al., 2022). Finally, the primary limiting nutrient of phytoplankton growth (i.e., nitrogen or iron) showed biases and correlations ranging from negative to positive, indicating that some were too nitrogen limited, some were too iron limited, and some performed well.



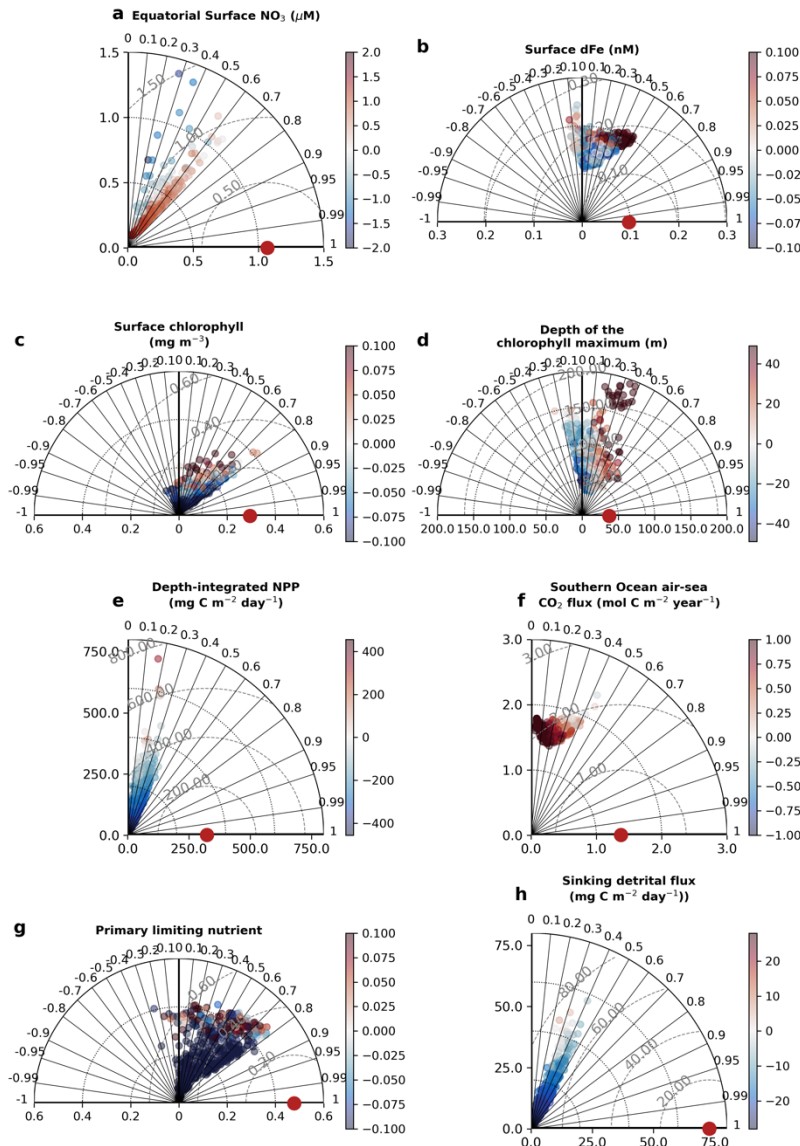


**Figure 4. Performance of the 512 sensitivity experiments with WOMBAT-lite against the 8 key observational targets.** Taylor Diagrams (Taylor, 2001) represent the agreement between a dataset and the target (red dot) by visualising the dataset in terms of its correlation (radii), normalised standard deviation (x and y axes), and the centered root mean square error (dashed grey contours). We also colour each experiment by its global mean bias. Positive bias in the Southern Ocean air-sea flux of $CO_2$ signifies too much outgassing or not enough ingassing. These statistics were computed on the global 1º by 1º grid at monthly resolution, except the Primary Limiting Nutrient dataset which was computed as an annual mean.






**Table 2. Performance ranges of experiments for key observations.** Minimum and maximum values of correlations, normalised standard deviations and bias across all 512 sensitivity experiments. Surface nitrate was $\log_{10}$ transformed. $NO_3$ = nitrate. dFe = dissolved iron. $CO_2$ = carbon dioxide. POC = particulate organic carbon.

| Observation | Correlation | Normalised standard deviation | Bias |
|---|---|---|---|
| Surface $NO_3$ (20ºS – 20ºN) (µM) | 0.20 \| 0.66 | 0.09 \| 1.30 | -2.14 \| 1.33 |
| Surface dFe (nM) | -0.13 \| 0.52 | 1.15 \| 2.51 | -0.10 \| 0.27 |
| Surface Chlorophyll (mg m$^{-3}$) | -0.45 \| 0.80 | 0.01 \| 1.41 | -0.27 \| 0.21 |
| Depth of the chlorophyll maximum (m) | -0.25 \| 0.69 | 0.89 \| 5.19 | -49.4 \| 271.4 |
| Depth-integrated net primary production (mg C m$^{-2}$ day$^{-1}$) | 0.02 \| 0.49 | 0.00 \| 2.26 | -456 \| 325 |
| Air-sea flux of $CO_2$ (20ºS – 90ºS) (mol C m$^{-2}$ year$^{-1}$) | 0.03 \| 0.44 | 1.01 \| 1.73 | -1.40 \| 0.26 |
| Primary limiting nutrient (1=N, 2=Fe) | -0.29 \| 0.82 | 0.00 \| 1.01 | -0.46 \| 0.51 |
| Sinking flux of POC (mg C m$^{-2}$ day$^{-1}$) | 0.08 \| 0.59 | 0.00 \| 0.79 | -27.6 \| 6.2 |

## 3.2 Global sensitivity analysis

Global sensitivity analysis with our first set of 512 experiments and supplemented by the surrogate machine learning model revealed that the overall performance of WOMBAT-lite was sensitive to 11 of the 24 parameters tested, based on an arbitrary threshold of a 5% contribution to variation (Fig. 5). These were the scaler on the maximum growth ($\alpha_a$) and linear mortality rates of phytoplankton ($\gamma_p^0$), the half-saturation coefficients for phytoplankton uptake of dissolved iron ($K_p^{dFe}$) and nitrate ($K_p^N$), the maximum quota of iron ($Q_p^{"\frac{Fe}{C}}$) and optimal quota of chlorophyll ($Q_p^{*\frac{Chl}{C}}$), the prey capture rate coefficient of zooplankton ($\varepsilon$) and their quadratic mortality rate ($\Gamma_z^0$), the sinking ($\omega_d^0$) and remineralisation ($\gamma_d^0$) rate of detritus, and the temperature sensitivity of heterotrophy ($\beta_h$). For each of the 8 target fields, typically only a few of these key parameters were influential. We step through these parameters here.



All target fields were highly sensitive to the phytoplankton maximum growth rates ($\alpha_a$) and their linear mortality ($\gamma_p^0$) (Fig. 5). Of these, only the depth of the chlorophyll maximum and surface dFe concentrations were sensitive to $\alpha_a$ and $\gamma_p^0$ via

interactive effects with each other or other variables (i.e., higher-order interactive effects). All other target fields were directly affected by these parameters, making $\alpha_a$ and $\gamma_p^0$ master parameters with largely predictable effects for controlling the performance and output of WOMBAT-lite. For example, while the air-sea flux of $CO_2$ in the Southern Ocean (south of 20ºS) was sensitive to several parameters, the model's ability to reproduce the observations was primarily controlled by the ability of phytoplankton to accumulate biomass rapidly in the spring and summer. If $\gamma_p^0$ was too high then too much biomass

was lost over the winter, causing a lag in the spring bloom. If $\alpha_a$ was too low, then the bloom would be too weak. The link between $CO_2$ ingassing in the summer and the phytoplankton bloom also meant that the prey capture rate coefficient of zooplankton ($\varepsilon$), the half-saturation coefficient for iron uptake by phytoplankton ($K_p^{dFe}$) and the sinking rate of detritus ($\omega_d^0$) were also important controls on Southern Ocean $CO_2$ fluxes.

Surface nitrate concentrations in the tropics (20ºS – 20ºN), depth-integrated net primary production and the sinking flux of detritus were all affected by similar parameters. After $\alpha_a$ and $\gamma_p^0$, the parameters of influence were the sinking and remineralisation rates of detritus ($\omega_d^0$ and $\gamma_d^0$). Elevated sinking rates and decelerated remineralisation both deepen the nitracline by stripping more nitrate out of the upper ocean and shrinking the large tongue of high nitrate water that spreads west across the equatorial Pacific. Surface nitrate in the equatorial band was also marginally affected by the temperature

sensitivity of heterotrophy ($\beta_h$), which amplifies remineralisation rates (set also by $\gamma_d^0$) in warmer waters and so can increase substantially how much detritus is returned to inorganic nutrients.





**Figure 5. Sensitivity of WOMBAT-lite performance to variations in the parameters listed in Table 1**. Performance is measured by the root mean square error (RMSE). Darker colours indicate a greater sensitivity of a target field to the parameter in question. In (a) we show first-order sensitivities and in (b) the higher-order sensitivities, otherwise referred to as an interaction effect, where the effect on the target is dependent on the variations in other parameters. Note that the parameter $\beta_a$ was not included at this stage but only later during the optimisation process (steps 7-10 in Fig. 1).

Two key parameters controlling the model's ability to reproduce surface dissolved iron (after $\alpha_a$ and $\gamma_p^0$) and the primary limiting nutrient dataset were the half-saturation coefficient for iron uptake ($K_p^{dFe}$) and the maximum Fe:C quota of phytoplankton ($Q_p^{"\frac{Fe}{C}}$). Variations in these parameters had strong interactive effects. Elevating iron quotas increased the ability of phytoplankton to take excess dFe into their biomass, reduced surface dFe concentrations and strengthened iron





limitation as more Fe-rich biomass was exported as sinking detritus. However, if we also increased the half-saturation

coefficient of dFe uptake, this slowed phytoplankton luxury uptake of dFe and made them less likely to achieve high intra-

cellular Fe:C quotas. As such, setting higher Fe:C quotas had little effect on surface dFe concentrations when $K_p^{dFe}$ was high.

On the other hand, setting high Fe:C quotas can have a substantial effect on dFe concentrations when $K_p^{dFe}$ is low. For the

primary limiting nutrient dataset, however, we found that increasing both Fe:C quotas and $K_p^{dFe}$ elevated dFe limitation

regardless of what happened to the dFe concentrations.

Performance in surface chlorophyll was the most inter-dependent metric, meaning that many parameters were influential. In

addition to the master parameters of $\alpha_a$ and $\gamma_p^0$, surface chlorophyll was affected by the maximum Fe:C quota ($Q_p^{\overline{\frac{Fe}{C}}}$), the

half-saturation of dFe uptake ($K_p^{dFe}$), the maximum quota of chlorophyll to carbon in phytoplankton ($Q_p^{\overline{\frac{Chl}{C}}}$), the prey capture

rate coefficient of zooplankton ($\varepsilon$), and marginally by the quadratic mortality coefficient of zooplankton ($\Gamma_z^0$). Thus, 7 of the

11 important parameters were influential. The fact that many parameters were crucial for determining the performance of

surface chlorophyll reflects that a delicate balance between phytoplankton growth and mortality must be struck to reproduce

overall biomass.

Finally, the depth of the chlorophyll maximum was overwhelmingly influenced by $\alpha_a$ and $\gamma_p^0$, with weaker influences from

the half-saturation coefficient for nitrogen uptake ($K_p^N$), cell quotas of iron and chlorophyll ($Q_p^{\overline{\frac{Fe}{C}}}$ and $Q_p^{\overline{\frac{Chl}{C}}}$), as well as

parameters related to the sinking and remineralisation of detritus ($\beta_h$, $\omega_d^0$ and $\gamma_d^0$). All these parameters affected the depth of

the nitracline in direct and indirect ways, to which the depth of the chlorophyll maximum was strongly linked.

**3.3 Parameter optimisation**

Our optimisation procedure involved another 512 experiments that varied 13 parameters: the 11 parameters identified in the

sensitivity analysis plus 2 additional parameters that were missed by our sensitivity analysis. All insensitive parameters were

held at their default values (Table 1). The additional 2 parameters included wider variations in the biomass threshold of

phytoplankton for allometric scaling ($B_p^{thresh}$) from 0.01 to 1.0 (previously this had been varied from 0.01 to 0.1), as well as

variations in the base scaler of temperature-dependent autotrophy ($\beta_a$), which was previously held constant during the

sensitivity experiments at a value of 1.066 ($Q_{10}$ = 1.89), following Eppley (1972), but have nonetheless been shown to vary

between phytoplankton types (Anderson et al., 2021a). We took the opportunity during the optimisation to vary these

parameters. In our optimisation step, we explored a range of 1.040 to 1.080 ($Q_{10}$ from 1.48 to 2.16) in $\beta_a$ motivated by the

results of Anderson et al. (2021a). Briefly, we explored variations in 13 parameters and sought their optimal values for best



reproducing the 8 target fields (Fig. 3) by minimizing the global cost function (Eq. 2). The 512 experiments were used to calibrate the global cost function synthetically using the machine learning model.

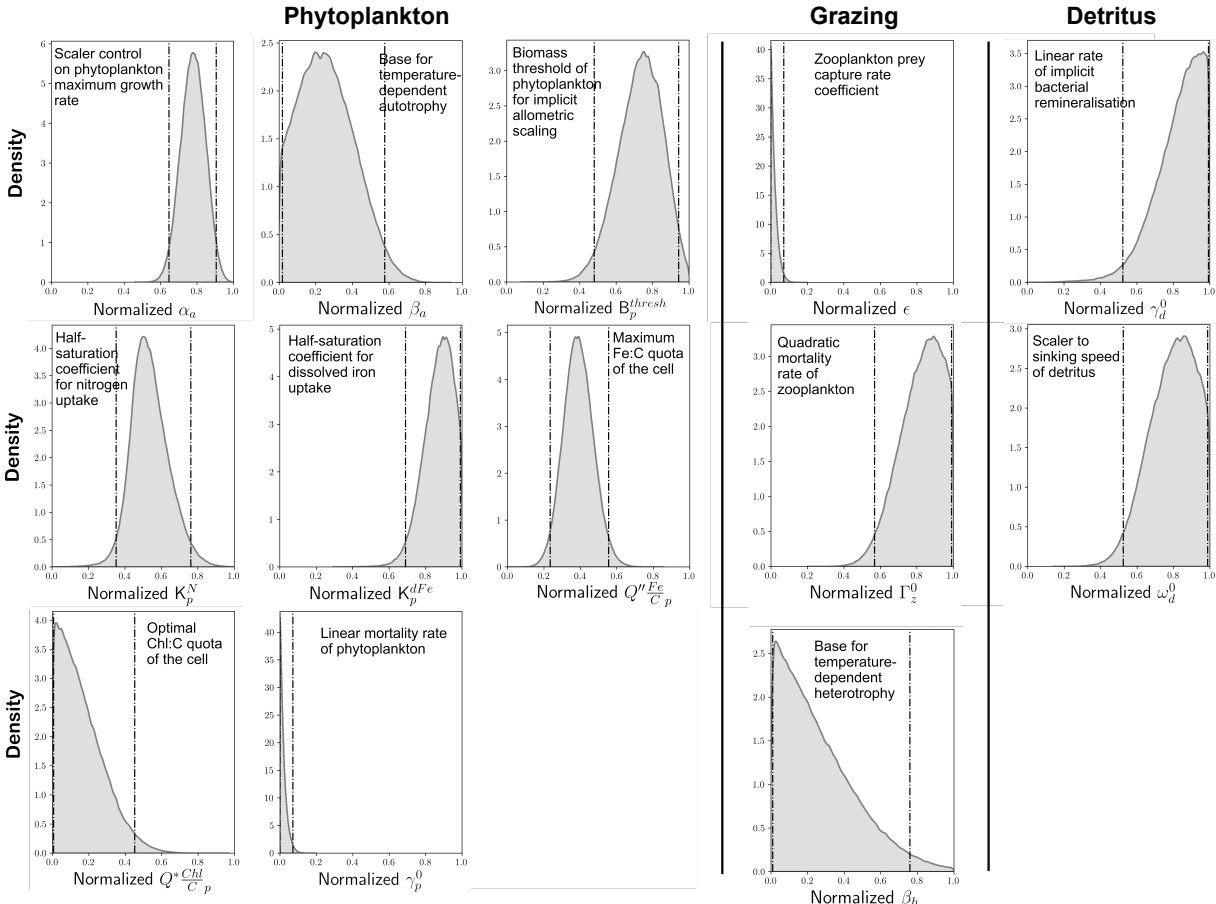

**Figure 6. Optimal probability density functions of 13 important parameters.** Optimisation involved minimizing the global (summed)
cost function (Eq. 2) of model performance across the 8 target variables shown in Figure 3. We show normalized distributions of each parameter here and refer the reader to the ranges (*a priori* and optimal) shown in Table 1 for their actual values. Parameters are organised according to whether they control phytoplankton, zooplankton or detritus.

We identified posterior distributions of each of the 13 parameters from which optimal values could be chosen (Fig. 6).
Optimal values for each of these parameters and their 95% confidence interval range are detailed in Table 1. For most of the parameters, the probability distributions showed peaks away from the edges of the predefined ranges, suggesting that our *a priori* ranges were sufficiently wide to capture the optimal values. For the scaler on maximum growth rates of phytoplankton ($\alpha_a$), for instance, the model predicted optimal values with 95% confidence between 0.89 and 1.16 day$^{-1}$, a range that sits within our *a priori* range of 0.25 to 1.25 day$^{-1}$ (Table 1) and suggests that the rapid accumulation of biomass during the



growth season is crucial for model performance. We also note that the model predicted optimal values that often aligned well

with ecological theory. Higher values of $K_p^{dFe}$ over $K_p^{N}$ (half-saturation coefficients for uptake) reflect the lesser

bioavailability of dFe relative to that of inorganic nitrogen due to its complexation with organics (Shaked et al., 2020;

Tagliabue et al., 2017). Fidelity to observations was also better when the temperature sensitivity of autotrophy ($\beta_a$) was

much lower than the temperature sensitivity of heterotrophy ($\beta_h$), consistent with ecological theory on metabolism (Brown et

al., 2004) and experimental data (Chen et al., 2012).

Optimal values for two parameters were predicted at the lower edge of their range. These were the linear mortality rate of

phytoplankton at 0ºC ($\gamma_p^0$) and the prey capture rate coefficient of zooplankton ($\varepsilon$), for which the optimal values were

predicted at 0.01 day$^{-1}$ and 0.05 m$^6$ (mmol C)$^{-2}$ day$^{-1}$, respectively (Table 1). Given that $\gamma_p^0$ was strongly influential to the

ability of WOMBAT-lite to reproduce all 8 of our target fields, it is likely that lower values of this parameter than explored

herein would produce a better fit to the observations. Rates of mortality considerably lower (< 0.001 day$^{-1}$) than the lowest

value in our *a priori* range (0.01 day$^{-1}$) were observed in phytoplankton cultures grown within their thermal niche (Baker and

Geider, 2021). However, mortalities also increase severely at higher temperatures (Baker and Geider, 2021), and our choice

of higher values in our *a priori* range was motivated by an attempt to account for the small proportion of phytoplankton taxa

placed above their thermal niche or affected by other sources of environmental stress at any given time. The fact that our

optimisation always chose the lowest values (near 0.01 day$^{-1}$) suggests that the proportion of the community that is stressed

is considerably lower than we assumed. Similarly, optimal values of $\varepsilon$, the zooplankton prey capture rate coefficient, tended

towards the lowest of our predefined range. This parameter was a strong control on the model's ability to reproduce the

observed surface chlorophyll and the summer uptake of $CO_2$ in the Southern Ocean (Fig. 5). Values less than 0.05 m$^6$ (mmol

C)$^{-2}$ day$^{-1}$ suggest grazing pressure more in line with a zooplankton community with a large representation of

mesozooplankton (Rohr et al., 2022). This type of community is typical for eutrophic and high latitude regions but may not

be representative of zooplankton grazers in the oligotrophic gyres (Rohr et al., 2024). Importantly, the lower grazing

pressure associated with this type of zooplankton community would allow phytoplankton growth during the spring to

outpace zooplankton grazing for longer, which we note again was important for achieving summer $CO_2$ uptake in the

Southern Ocean. While we cannot say whether an expanded range would have selected for even lower values, we note that a

value near 0.05 m$^6$ (mmol C)$^{-2}$ day$^{-1}$ sits at the global median of empirical estimates across a large range of zooplankton taxa

(Rohr et al., 2022).



## 3.4 Outcomes

### 3.4.1 Finding the optimal parameter set

After our optimisation, we chose 20 randomly sampled parameter sets from the optimal posterior distributions shown in Figure 6 and ran WOMBAT-lite forward for 10 years from initial conditions. These optimal versions of WOMBAT-lite show good fidelity to the target fields, with all registering good performances in terms of the global cost function that were as good as or better than the best of the 512 sensitivity experiments (yellow bars in Fig. 7). Continuing to run the model forward for 100 years post initialization showed some degradation in the performance (red bars in Fig. 7). This is expected, since our optimisation procedure was trained on model output only 10 years post initialization due to computational constraints. Model outcomes drift further away from the target fields with longer integrations. Lower frequency variability and trends are thus missed by the optimisation that are nonetheless present in the biogeochemical model, and these play out as the model is integrated forward for longer. After 100 years, we chose our best performing parameter set, detailed in Table 1 (red star in Fig. 7). This experiment showed good performance across all observational metrics in its 100th year and we hereafter show output from this experiment.

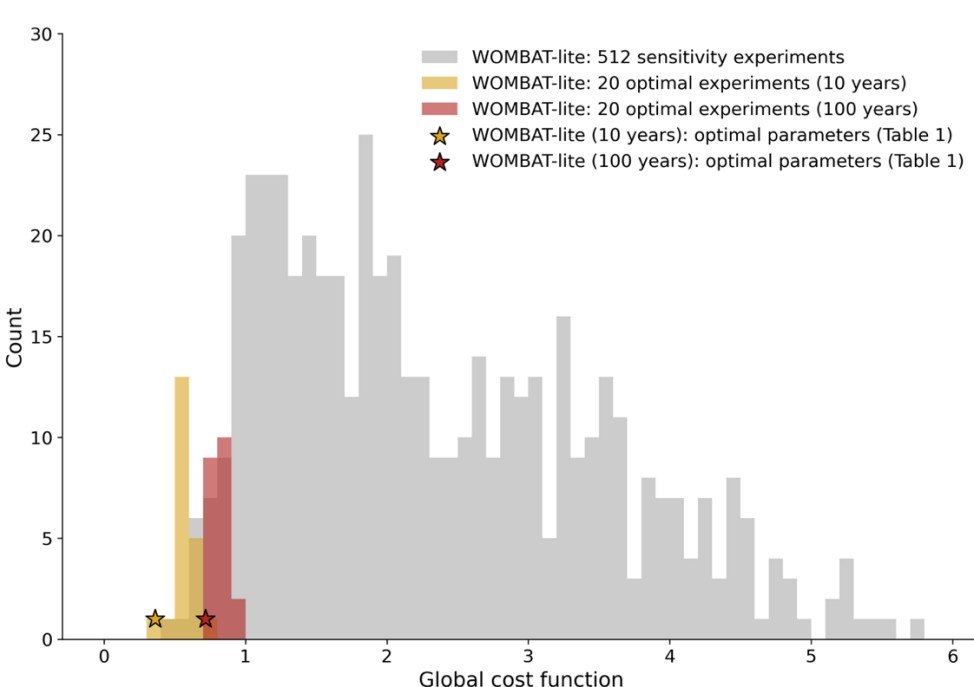

**Figure 7. Overall performance of WOMBAT-lite in terms of the global cost function (summed across all target variables; Eq. 2).** We show all 512 sensitivity experiments (grey), the 20 optimal experiments that selected parameter values randomly from the optimal probability density functions (Fig. 6) after 10 years (gold) and 100 years (red), and the performance of the optimal parameter set after 100 years (red star).



### 3.4.2 Performance improvements

A key challenge in model development is that the addition of new processes can degrade performance if the new processes are implemented poorly and/or if these processes have complex interactive effects once introduced. Our sensitivity analysis and optimisation procedure provided a means to constrain both the effects and values of the WOMBAT-lite parameter set, so that any advances in functionality are accompanied by an improvement in performance. We take the opportunity to compare the optimised WOMBAT-lite with an unoptimised biogeochemical model (Appendix A.1), run under the same conditions and without the functional improvements described in Appendix A.2. The optimised WOMBAT-lite shows a better tropical distribution of surface nitrate (Fig. 8a-c), lower concentrations of dissolved iron at the surface (Fig. 8d-f) and, consequently, the appearance of iron-limited regimes for phytoplankton growth in the Southern Ocean, subarctic North Pacific and Atlantic, eastern Equatorial Pacific, and the upwelling centres of the Benguela, Arabian and Canary current systems (Fig. 8s-u). Note that the surface iron distribution shown in Figure 8 is of the annual average, which includes higher concentrations caused by winter mixing (Tagliabue et al., 2014b), whereas much of the observations will have been taken during the polar summer when dFe concentrations are drawn down by biology. WOMBAT-lite also shows a 50% increase in globally integrated net primary production (from 18.5 Pg C yr$^{-1}$ to 27.9 Pg C yr$^{-1}$) compared to the unoptimised model, and less diffuse peaks of primary production in the highly productive upwelling zones, consistent with observations (Fig. 8m-o). The increase in primary production combined with the spatially variable sinking scheme, which includes a linear increase in sinking speeds with depth, likely contributed to elevating the flux of detritus into the deep ocean (Fig. 8v-x). We note, however, that any improvements to sinking of detritus are marginal and deep ocean organic particle fluxes are still underestimated, likely because WOMBAT-lite still underestimates globally integrated net primary production (Buitenhuis et al., 2013) and stocks of sinking detritus (Fox et al., 2024). Finally, WOMBAT-lite shows good agreement with surface chlorophyll and the broad patterns in the depth of the chlorophyll maximum (Fig. 8g-l), although the chlorophyll maxima are situated too deep in the gyres and are too shallow in the Southern Ocean.



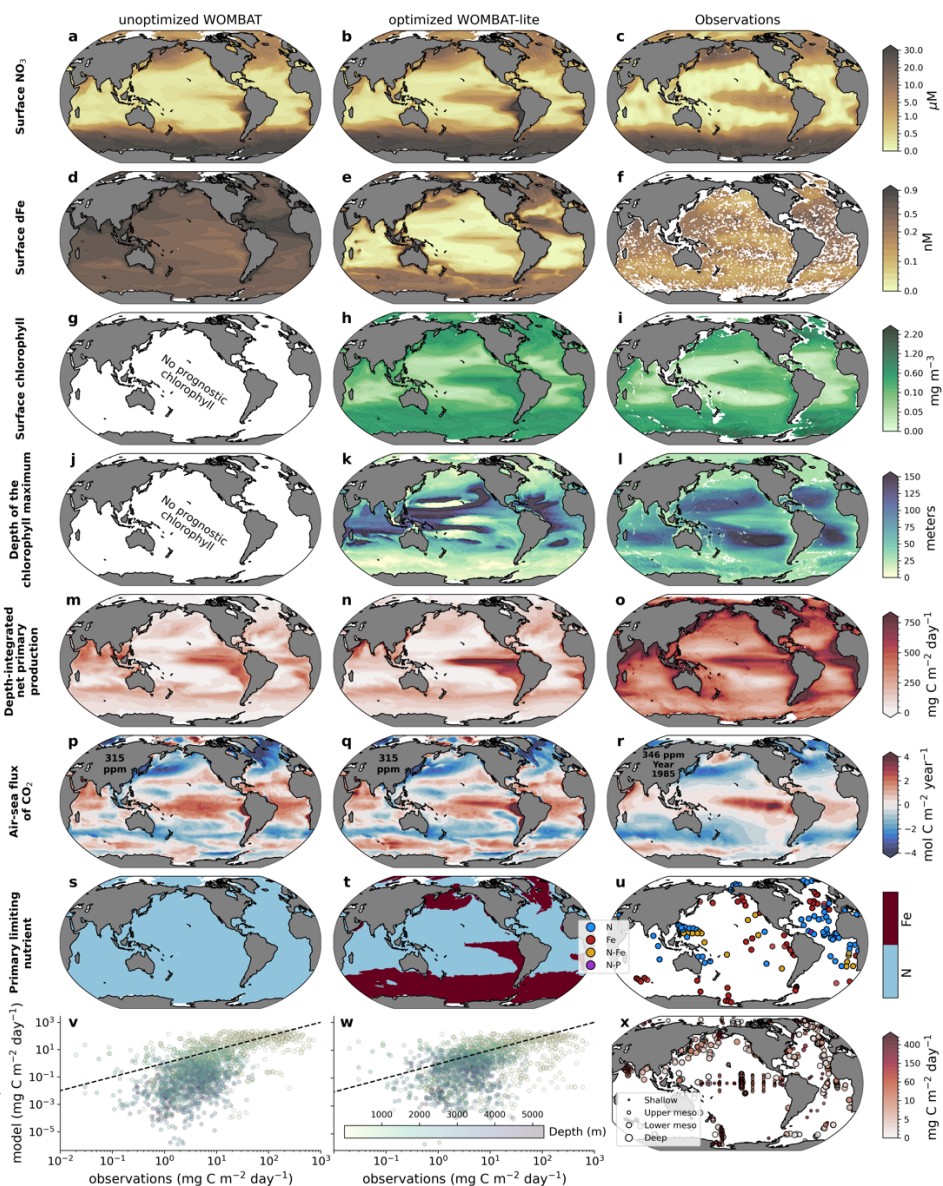

**Figure 8. Comparison of the previous, unoptimised WOMBAT (left) and WOMBAT-lite (middle) with the 8 target observations (right).** (**a-c**) Surface nitrate concentration (μM). (**d-f**) Surface dissolved iron concentration (nM). (**g-i**) Surface chlorophyll concentration (mg m$^{-3}$). (**j-l**) Depth of the chlorophyll maximum (m). (**m-o**) Depth-integrated net primary production (mg C m$^{-2}$ day$^{-1}$). (**p-r**) Downward flux of $CO_2$ (mol C m$^{-1}$ year$^{-1}$) (μM). (**s-u**) Primary limiting nutrient to phytoplankton growth at the surface. (**v-x**) Downward flux of particulate organic carbon (mg C m$^{-1}$ day$^{-1}$). We show annual means here even though statistical measures of performance shown in Figure 6 are computed including temporal variations on a monthly resolution. Output after 100 years of forward simulation from initialization with repeat atmospheric forcing for calendar year 1990-1991 conditions using the JRA-55do (Tsujino et al., 2018). Atmospheric $CO_2$ set at 315 ppm in the experiments. Note that the ocean-only hindcast runs performed here with the previous, unoptimized WOMBAT used a parameter set detailed in Law et al. (2017) that is intended for the ACCESS-1.5 Earth System Model.





### 3.4.3 Phytoplankton bloom phenology

A major update to WOMBAT-lite has been to inclusion of prognostic chlorophyll to carbon ratios within its phytoplankton functional type. This allows for direct comparison with remotely sensed chlorophyll products, including those that investigate the phenology of phytoplankton blooms (Nicholson et al., 2024). WOMBAT-lite was not optimised for its representation of phytoplankton phenology, but nonetheless performs well in respect to the timing of its annual blooms (Fig. 9a,b). The model captures the sharp change in bloom initiation (using the cumulative sum method) between the subtropical

and subpolar regimes in both hemispheres, with autumn-winter subtropical blooms and spring-summer polar blooms. WOMBAT-lite also shows a general increase in duration of its blooms in the tropics compared to the polar regions (Fig. 9c,d), as well as increases in the mean and integrated chlorophyll concentrations in the polar and upwelling regions compared to the subtropical gyres (Fig. 9e-h).

That said, WOMBAT-lite shows some clear biases compared with the Nicholson et al. (2024) dataset, which was built from the Ocean Color – Climate Change Initiative remotely-sensed chlorophyll product (Sathyendranath et al., 2019). Blooms at the poles start too early and their duration is too long. Overly long blooms in the Southern Ocean contributed to an overestimate of mean and integrated concentrations of chlorophyll than that calculated by Nicholson et al. (2024) (Fig. 9i). This bias may be associated with an excess of dFe due to a ferricline that is placed too shallow (Fig. 8e,f), a common model

bias (Tagliabue et al., 2016) that is also present in our simulations, and which amplifies iron supply and chlorophyll accumulation. Meanwhile, in the subtropics and northern high latitudes, the phytoplankton blooms in WOMBAT-lite appear to be too low in mean and integrated chlorophyll (Fig. 9i). This is possibly caused by bloom durations that are too short in the subtropics (they are > 300 days in the remote-sensing product), although this is not the case in the north polar region. Equally, though, these biases in bloom chlorophyll metrics may be due to our optimisation against a different chlorophyll

product (Sauzède et al., 2016), with which the model shows good agreement. Capturing the bloom phenology of phytoplankton is important because there is evidence of multi-decadal trends in their start, end, and duration in both the Southern and Arctic Oceans (Ardyna and Arrigo, 2020; Thomalla et al., 2023), and according to the results herein, the timing and duration of the bloom is influential to air-sea $CO_2$ fluxes.





**Figure 9. Comparison of WOMBAT-lite (left) with phenological indicators of the annual phytoplankton bloom observed via chlorophyll-a (right).** (**a-b**) Bloom initiation day. (**c-d**) Bloom duration (days). (**e-f**) Mean bloom chlorophyll-a across duration (mg m$^{-3}$). (**g-h**) Integrated bloom chlorophyll (mg m$^{-3}$ bloom$^{-1}$). (**i**) Zonal mean integrated bloom chlorophyll for the observations (black) and WOMBAT-lite (red). Observations are provided by Nicholson et al. (2024). Phenological metrics are calculated via the cumulative sum method. For WOMBAT-lite these results come from the optimized model after 100 years of forward simulation from initialization with repeat atmospheric forcing for calendar year 1990-1991 conditions using the JRA-55do (Tsujino et al., 2018).



### 3.4.4 Carbon fluxes

Previous assessments of ocean biogeochemical models show that the Southern Ocean air-sea $CO_2$ fluxes are strongly biased (Hauck et al., 2020, 2023). We therefore sought to improve this aspect within WOMBAT-lite. To properly assess the performance of WOMBAT-lite for reproducing global $CO_2$ fluxes, we performed a hindcast simulation with the optimal version of WOMBAT-lite from 1958 to 2019 forced by the inter-annually changing JRA55-do atmospheric fields and with the historical increase in atmospheric $CO_2$. This hindcast simulation was initialised with biogeochemical fields at the end of a

200-year spin-up simulation with the same repeat "normal" year forcing of the JRA55-do (Stewart et al., 2020), with atmospheric $CO_2$ set at 315.2 ppm (equivalent calendar year 1957) and where alkalinity and preindustrial DIC budgets were at quasi-equilibrium. Budgets of major tracers at year 200 are presented in Table 3. This hindcast simulation did not strictly adhere to the recommendations of the OMIP2 protocol (Orr et al., 2017), but was sufficient to assess the seasonality of air-sea $CO_2$ fluxes in WOMBAT-lite.

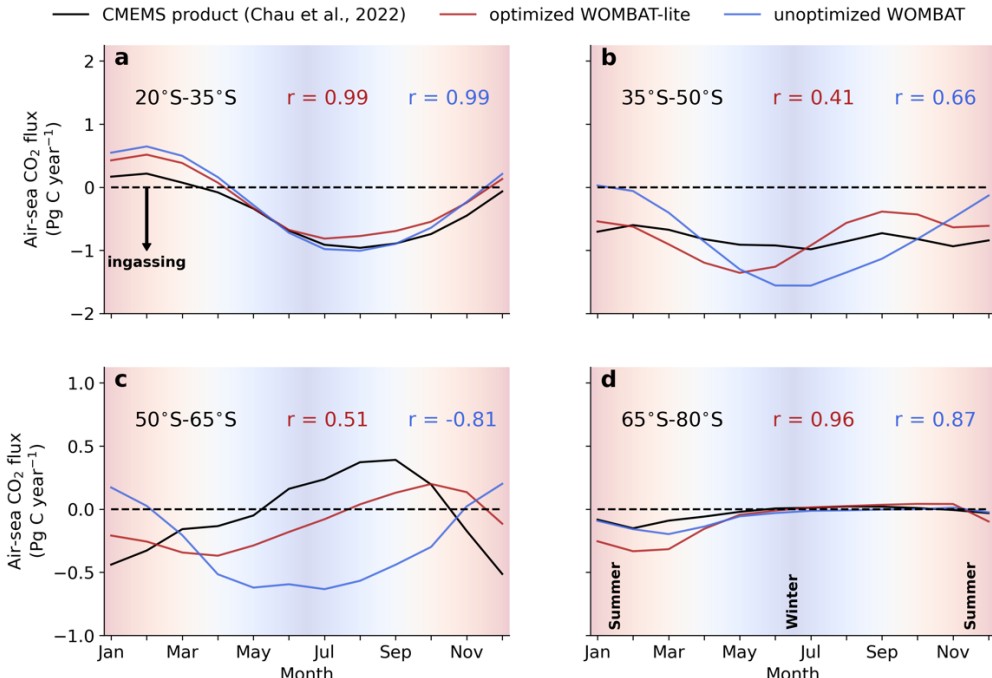

**Figure 10.**
**Climatological evolution of the integrated air-sea $CO_2$ flux over the latitudinal bands in the Southern Ocean averaged over years 1990 to 2010.** (**a**), Observational product from the Copernicus Marine Environmental Monitoring Service (black) are compared with fluxes from WOMBAT-lite (red) and an unoptimized, previous version of WOMBAT (blue) integrated between 20ºS-35ºS (subtropics). Pearson's correlations are shown for both models. Negative values are net ingassing of $CO_2$ into the ocean. Background shading denotes the seasonal

transition from summer to winter to summer. Panels (**b**), (**c**), and (**d**) are the same as in (**a**), but integrated within 35ºS-50ºS for the subtropical-subantarctic transition, 50ºS-65ºS for the Antarctic Circumpolar Current and 65ºS-80ºS for Antarctic zone, respectively.



**Table 3. Key rates in WOMBAT-lite after 200-year spin-up under repeat normal year forcing with optimised parameters.**

| Cycle | Description | Units | WOMBAT-lite |
|---|---|---|---|
| Carbon | Net primary production | Pg C yr$^{-1}$ | 27.9 |
| | Organic export (100 m) | Pg C yr$^{-1}$ | 6.2 |
| | CaCO$_3$ export (100 m) | Pg C yr$^{-1}$ | 0.60 |
| | Preindustrial air-sea flux | Pg C yr$^{-1}$ | -0.358 |
| | Sedimentary burial | Pg C yr$^{-1}$ | -0.20 |
| | CaCO$_3$ sedimentary burial | Pg C yr$^{-1}$ | -0.012 |
| | Riverine flux | Pg C yr$^{-1}$ | 0.587 |
| Alkalinity | Sedimentary burial | Pmol Eq. yr$^{-1}$ | 0.002 |
| | CaCO$_3$ sedimentary burial | Pmol Eq. yr$^{-1}$ | -0.002 |
| | Riverine flux | Pmol Eq. yr$^{-1}$ | 0.018 |
| | Sedimentary denitrification | Pmol Eq. yr$^{-1}$ | 0.0028 |
| | Nitrogen fixation | Pmol Eq. yr$^{-1}$ | -0.0015 |
| Nitrogen | Sedimentary burial | Tg N yr$^{-1}$ | -30.8 |
| | Riverine flux | Tg N yr$^{-1}$ | 35.8 |
| | Nitrogen fixation | Tg N yr$^{-1}$ | 20.4 |
| | Sedimentary denitrification | Tg N yr$^{-1}$ | -39.6 |
| Oxygen | Preindustrial air-sea flux | Pmol O$_2$ yr$^{-1}$ | -0.04 |
| | Volume of hypoxia (< 60 μM) | $10^{15}$ m$^3$ | 79.1 |
| Iron | Atmospheric deposition | Gmol Fe yr$^{-1}$ | 1.1 |
| | Hydrothermal flux | Gmol Fe yr$^{-1}$ | 9.9 |
| | Sedimentary burial | Gmol Fe yr$^{-1}$ | 0.60 |




With optimal parameters, WOMBAT-lite shows improvement in its seasonality and regional agreement of $CO_2$ fluxes compared with an unoptimised version of the model (Fig. 10). While $CO_2$ fluxes are strongly controlled by thermal processes
in the subtropics and are thus well approximated by optimised and unoptimised versions alike (Fig. 10a), $CO_2$ fluxes at higher latitudes are, however, more affected by biological drawdown and release (Mongwe et al., 2018; Takahashi et al., 2002). In the transition from subtropical to subantarctic zones (35ºS-50ºS) the observations show overall oceanic uptake of $CO_2$, but importantly a greater uptake in the summer (Fig. 10b). Our optimised WOMBAT-lite manages to show some improvement over the unoptimised model, with lower uptake in the winter and a trend towards uptake in the spring/summer.
Nonetheless, this improvement is marginal in this zone and suggests that further improvement can be made in the future. The best match between WOMBAT-lite and the data is achieved in the Antarctic Circumpolar Current zone (50ºS-65ºS), where WOMBAT-lite shows good climatological correlations with the observations, while the unoptimised model shows negative correlations. The flip from poor to good performance is caused by the net outgassing in the late winter and a trend towards oceanic uptake in the spring summer (Fig. 10c). Better seasonal correlations (0.87 → 0.96) are also achieved in the Antarctic
Zone (65ºS-80ºS), although with WOMBAT-lite potentially overestimating the summer flux of $CO_2$ into these waters (Fig. 10d).

These seasonal improvements in air-sea $CO_2$ fluxes are clear by looking at correlations at each grid cell (Fig. 11). We directly compared the monthly $CO_2$ fluxes between an observation product (Chau et al., 2022) and WOMBAT-lite as well as
the unoptimised model from January 1985 to December 2018. Negative correlations across a broad swath of the Southern Ocean are evident in the unoptimised model, as well as the subarctic Pacific and Atlantic Oceans. The optimised WOMBAT-lite, however, shows many positive correlations in these regions. Negative and near-zero correlations do still exist in WOMBAT-lite, suggesting room for further improvement, but the area and intensity of these poorly performing locations are considerably reduced.


Improvement in air-sea $CO_2$ fluxes is noteworthy from a zonally integrated perspective that incorporates the Northern Hemisphere (Fig. 12). North of 40ºN, the oceanic uptake of $CO_2$ in the unoptimised version of the model exceeded the observations. Winter-time uptake north of 40ºN in WOMBAT-lite is, however, substantially reduced (Fig. 12b), while summer uptake is increased (Fig. 12c), resulting in a better match to observed $CO_2$ fluxes in the Northern Hemisphere (Fig.
12a) that is also visible in the temporal correlations (Fig. 11). Meanwhile, there is little difference between the optimised and unoptimised versions of the model in the low latitudes, emphasising how thermal changes dominate air-sea $CO_2$ fluxes in this region (Takahashi et al., 2002). Once again, in the Southern Ocean, we see clear improvements from a zonally integrated perspective. Winter outgassing is now achieved in WOMBAT-lite, although the zone of peak outgassing occurs too far north (Fig. 12c). Similarly, summer oceanic uptake is now achieved and is a closer match to the observations, although again the





zones of maximum uptake are shifted too far north by roughly ~5° (Fig. 12b). Overall, the changes in the biogeochemical

functionality of WOMBAT-lite show some improvements in reproducing observed air-sea $CO_2$ fluxes.

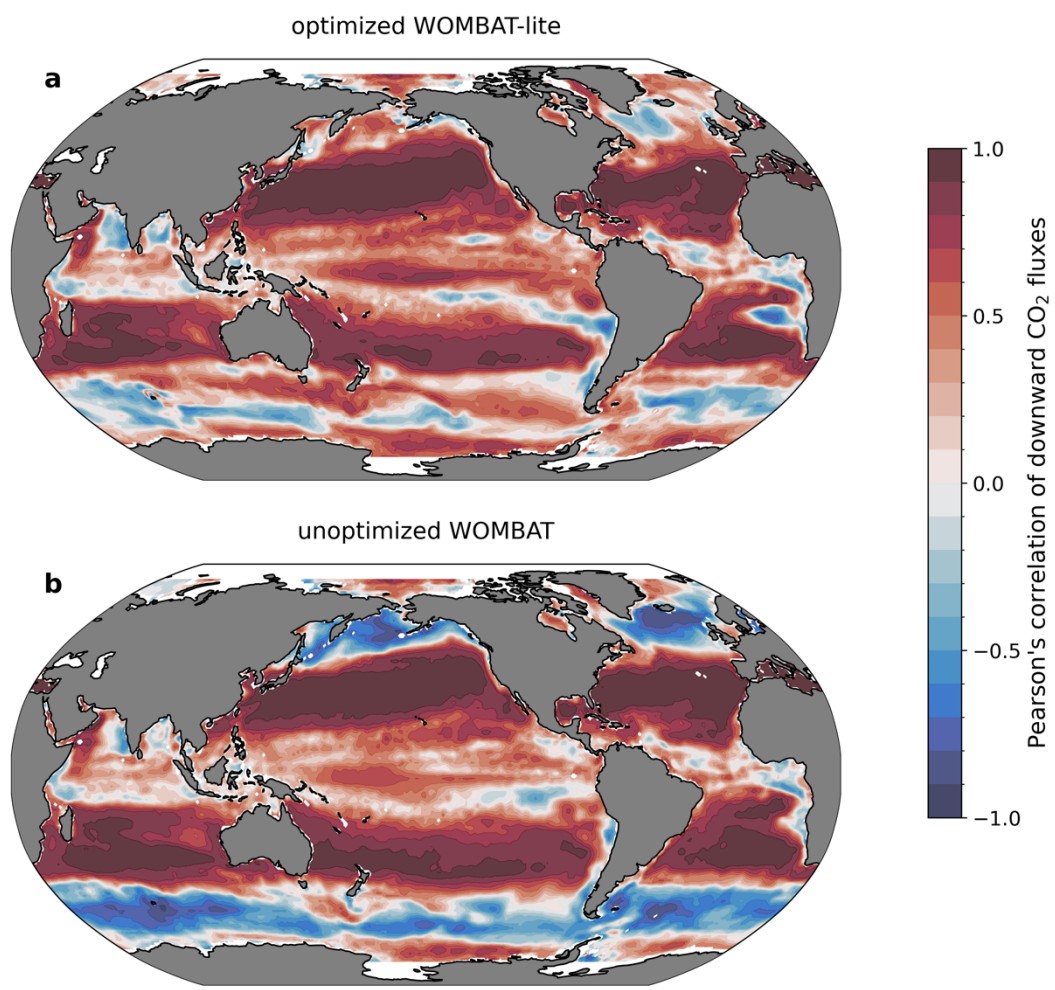

**Figure 11. Temporal correlations in monthly $CO_2$ fluxes with an observation product** (Chau et al., 2022)**. (a)**, optimised WOMBAT-
lite and (**b**), unoptimized, previous WOMBAT over years 1985 to 2019.



**Figure 12. Zonally integrated air-sea CO₂ fluxes.** (**a**), Annual mean observations (black) averaged over years 1990-2010 are compared with fluxes from WOMBAT-lite (red) and a previous, unoptimized version of WOMBAT (blue). Panels (**b**) and (**c**) are the same as in (**a**) but are averages over the months of January-March and July-September, respectively. Positive fluxes are out of the ocean.



**Summary**

We have updated The World Ocean Model of Biogeochemistry And Trophic dynamics (WOMBAT) in a new version called
WOMBAT-lite. Although "lite", this version is more complex than the previous WOMBAT. Updates include an implicit
representation of the packaging effect of cell size, which explicitly controls light attenuation through the water column;
explicit photoacclimation of phytoplankton; a dynamic colimitation of iron and light, whereby phytoplankton require
increases in their iron quotas as they photoacclimate (i.e., increase their chlorophyll quotas) in low light conditions; spatial
variations in nutrient affinities of phytoplankton for nitrogen and iron; expanded iron cycling, with additional sinks due to
scavenging onto detrital particles, colloidal coagulation and nanoparticle precipitation; riverine sources offset by permanent
burial in sediments; a spatially varying sinking rate of detritus that increases with an implicit estimate of mean community
cell size and also with depth; simplistic sedimentary denitrification and nitrogen fixation routines; and dampened
temperature-dependence of zooplankton grazing.

These updates necessitated optimisation of the many model parameters now present in WOMBAT-lite. To do so, we used a
surrogate machine learning model trained on a limited sample of real model output. It is critical for accurate global
sensitivity analysis (Saltelli et al., 2019) and Bayesian parameter optimisation techniques (Reddy et al., 2024a) to have a
large number of samples. These surrogate-based sensitivity analyses and parameter optimisation techniques are therefore
gaining traction in climate and environmental sciences where computational overhead severely restricts the number of direct
model simulations that can be done (Li et al., 2018; Reddy et al., 2024a, b; Xu et al., 2022, 2018). Ocean biogeochemical
models are no different. Here we applied the surrogate-based method to a new model, WOMBAT-lite, optimised the model
parameters and delivered improved performance in reproducing 8 target datasets. Our approach is a powerful way to
optimise the parameter set of complex models with large computation overhead and to identify the most important
parameters for realistic simulations. This was done in ocean-only simulations with repeating atmospheric conditions to limit
physical biases and focus evaluation on the biogeochemical component and its influence on upper ocean fields.


The improvements showcased herein included surface distributions of iron and nitrate, surface chlorophyll concentrations,
the representation of deep chlorophyll maximums, phytoplankton phenology, and particularly the seasonality of air-sea $CO_2$
fluxes in the high latitudes, where in some regions the seasonality flipped from a negative correlation in the unoptimised
model to a positive correlation with our optimised, new model. Surface chlorophyll was also well reproduced, as was the
distribution of iron-limited regions of primary productivity. Additionally, global net primary production was increased by
50%, partially rectifying a low bias in the previous unoptimised model, although our simulated rate of 28 Pg C yr$^{-1}$ remains
low compared with data-based estimates of ~50 Pg C yr$^{-1}$ (Buitenhuis et al., 2013).



Despite these improvements, biases do remain. Chief among them is a difficulty in reproducing the seasonality of air-sea
$CO_2$ exchange in the Southern Ocean, despite the improvements achieved here. WOMBAT-lite does manage to represent the
austral winter outgassing of $CO_2$ from the polar frontal region but fails to absorb enough $CO_2$ in the summer, particularly in
the subantarctic zone. Other models struggle with representing the seasonality of $CO_2$ exchange in the Southern Ocean, with
some absorbing too much (e.g., Yool et al., 2021) and others, like WOMBAT-lite, too little (see Hauck et al., 2020, 2023).
Physical biases in the model are no doubt important here, such as those that have insufficient winter mixing, as has been
proposed (Hauck et al., 2023). Our sensitivity analysis and optimisation procedure, however, would also suggest that how we
chose to represent the biology contributes substantially to model skill in air-sea $CO_2$ fluxes (also see Mongwe et al., 2018).
Two parameters of importance for Southern Ocean $CO_2$ fluxes were optimised at the lower edge of their *a priori* ranges: the
linear mortality coefficient for phytoplankton ($\gamma_p^0$) and the prey capture rate coefficient of zooplankton ($\varepsilon$). This suggests that
further improvement in Southern Ocean $CO_2$ fluxes can be gained by exploring lower values for these parameters than those
explored here, or alternatively introducing spatial variations in $\varepsilon$ that would capture transitions from nano- to meso-
zooplankton grazing from oligotrophic to eutrophic regimes (Rohr et al., 2024). Interestingly, lowering these parameters
would serve to accelerate the phytoplankton bloom at the beginning of the growth season, suggesting that the phenology of
the annual spring bloom is a primary control on Southern Ocean $CO_2$ fluxes. According to a recent analysis, this phenology
is changing (Thomalla et al., 2023), which may imply a changing strength and/or seasonality of air-sea $CO_2$ fluxes in the
region.

As a final word, we note that a surrogate-based optimisation of a complex numerical model can only be as good as the initial
sample set on which the surrogate is trained. A clear example of this limitation is evident in Figure 7. Even with its optimal
parameters, WOMBAT-lite suffered a loss in performance when run over 100 years compared to when run over only 10
years. Future iterations of surrogate-based optimisation would therefore benefit from extending the length of simulations
done by initial set of sensitivity experiments. That said, significant savings in computation efficiency would be needed
before this is possible with computationally demanding models, such as ocean biogeochemical models, but could be feasible
by running the biogeochemical model offline from the ocean physics (e.g., Séférian et al., 2013). This approach would also
eliminate any confounding errors caused by an evolution of the ocean's physical state since the physical state would not be
allowed to evolve. Future versions of WOMBAT, including WOMBAT-lite, WOMBAT-mid and WOMBAT-full, and their
deployment into different configurations (e.g., higher resolution versions) would benefit from this methodology of
optimisation.



**Code availability**

Model code of WOMBAT-lite developed for the simulations done in this paper is available at
https://github.com/pearseb/WOMBAT_dev. Future versions will be implemented using the GFDL "generic tracer"
framework to enable its use in MOM5 and MOM6. The code is available at https://github.com/ACCESS-NRI/GFDL-
generic-tracers. Code for the sensitivity analysis and optimisation routines are at https://github.com/Jyoteesh38/Bayesian-
Optimization-of-the-World-Ocean-Model-of-Biogeochemistry-and-Trophic-dynamics-WOMBAT-. Code for the analysis in
sections 3.1 and 3.4 and the figures therein is available at https://github.com/pearseb/WOMBAT-lite_optimisation_analysis.

**Data availability**

Model output will be made available upon request due to the size of the output (> 50 Gb).

**Competing Interests**

The authors declare that they have no conflict of interest.

**Acknowledgements**

This research was undertaken with the assistance of resources from the National Computational Infrastructure (NCI
Australia), an NCRIS enabled capability supported by the Australian Government. TR is an ARC DECRA recipient
(DE240100115) funded by the Australian Government. The authors wish to acknowledge use of the Ferret program
(http://ferret.pmel.noaa.gov/Ferret/), climate data operators (https://code.mpimet.mpg.de/projects/cdo/), NetCDF Operators
(http://nco.sourceforge.net/) and Python (www.python.org) for the analysis and graphics in this paper. Thanks to Thomas
Moore, Tilo Ziehn and Rachel Law for their support and for generously providing computational resources and storage.

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





**Appendix A**

**A.1 Key equations of the previous WOMBAT**

The previous version of WOMBAT is a simple nutrient-phytoplankton-zooplankton-detritus model (Law et al., 2017; Oke et al., 2013) simulating the biomass pools of one phytoplankton, one zooplankton and one detrital functional type, two nutrients of phosphate * 16, which is often referred to conveniently as nitrate, ($NO_3$, mmol m$^{-3}$) and dissolved iron (dFe, μmol m$^{-3}$), the carbonate system (dissolved inorganic carbon, alkalinity, and calcium carbonate; mmol C m$^{-3}$), and dissolved oxygen (mmol $O_2$ m$^{-3}$). The basic currency of the ecosystem component is 16 times phosphorus (i.e., nitrogen but without external sources and sinks). Biological stoichiometry is fixed at a C:N:$O_2$ ratio of 106:16:-172, and the Fe:N of phytoplankton is 20:1 μmol:mol. The minimum of nutrient availability or availability of photosynthetically activate radiation (PAR, W m$^{-2}$) control phytoplankton growth rates. Growth of phytoplankton is calculated by first solving for a maximum growth rate ($\mu^{max}$, day$^{-1}$), dependent on temperature (T, ºC), and then applying the minimum of nutrient ($L^{nutrient}$) and light ($L^{PAR}$) limitation terms to scale down from the maximum to a realised growth rate ($\mu$, day$^{-1}$).

The budgets for the previous WOMBAT are as follows:

$$\frac{\partial NO_3}{\partial t} = \gamma_d + \gamma_z + \gamma_p - \mu B_p \,, \tag{A1}$$

$$\frac{\partial dFe}{\partial t} = \left(\gamma_d + \gamma_z + \gamma_p - \mu B_p\right)\left(\frac{Fe}{N}\right) - \text{Scavenging} \,, \tag{A2}$$

$$\frac{\partial O_2}{\partial t} = \left(\mu B_p - \gamma_d - \gamma_z - \gamma_p\right)\left(\frac{O_2}{N}\right) \,, \tag{A3}$$

$$\frac{\partial B_p}{\partial t} = \mu B_p - \gamma_p - \Gamma_p - g B_z \,, \tag{A4}$$

$$\frac{\partial B_z}{\partial t} = g B_z \lambda - \gamma_z - \Gamma_z \,, \tag{A5}$$

$$\frac{\partial B_d}{\partial t} = \Gamma_p + \Gamma_z + g B_z (1 - \lambda) - \gamma_d \,, \tag{A6}$$

$$\frac{\partial B_{CaCO_3}}{\partial t} = \left(\Gamma_p + \Gamma_z + g B_z (1 - \lambda)\right)\left(\frac{C}{N}\right) f_{inorg} - \gamma_{CaCO_3} \,, \tag{A7}$$

$$\frac{\partial DIC}{\partial t} = \left(\gamma_d + \gamma_z + \gamma_p - \mu B_p\right)\left(\frac{C}{N}\right) - \left(\left(\Gamma_p + \Gamma_z + g B_z (1 - \lambda)\right)\left(\frac{C}{N}\right) f_{inorg} - \gamma_{CaCO_3}\right) \,, \tag{A8}$$

$$\frac{\partial Alk}{\partial t} = -\left(\gamma_d + \gamma_z + \gamma_p - \mu B_p\right) - 2\left(\left(\Gamma_p + \Gamma_z + g B_z (1 - \lambda)\right)\left(\frac{C}{N}\right) f_{inorg} - \gamma_{CaCO_3}\right) \,. \tag{A9}$$





Regarding phytoplankton growth rates ($\mu$, units of day$^{-1}$) the individual terms are

$$\mu^{max} = \alpha\beta^{(T)} , \tag{A10}$$

$$L^{PAR} = \left(1 - e^{-\frac{PI \cdot PAR}{\mu^{max}}}\right) , \tag{A11}$$

$$L^{nutrient} = \min\left(\frac{NO_3}{NO_3 + K^N}, \frac{dFe}{dFe + K^{Fe}}\right) , \tag{A12}$$

$$\mu = \min\left(\mu^{max}L^{PAR}, \mu^{max}L^{nutrient}\right) , \tag{A13}$$

where $\alpha$ and $\beta$ control the non-linear temperature-dependency of phytoplankton metabolism (Eppley, 1972), PI is the slope of the photosynthetic-irradiance curve (mg N (mg Chl)$^{-1}$ (W m$^{-2}$)$^{-1}$ day$^{-1}$), and $K^N$ and $K^{Fe}$ are the half saturation coefficients for nutrient limitation by $NO_3$ and $dFe$ (mmol m$^{-3}$ and µmol m$^{-3}$, respectively). The realised growth of phytoplankton ($\mu$, day$^{-1}$) is multiplied by their biomass ($B_p$, mmol N m$^{-3}$) to retrieve growth of phytoplankton (mmol N m$^{-3}$ day$^{-1}$). PAR is solved for as 0.43 of incoming shortwave radiation and then attenuated at a constant rate through the water column.

Phytoplankton mortality occurs via zooplankton grazing, as well as linear ($\gamma$) and quadratic ($\Gamma$) mortality terms. The specific grazing rate for zooplankton (g, units of day$^{-1}$) is described by a Type III disk equation (Gentleman and Neuheimer, 2008)

$$g = \frac{g^{max}\varepsilon B_p^2}{g^{max} + \varepsilon B_p^2} , \tag{A14}$$

where $g^{max}$ is the constant, temperature-independent, maximum (prey replete) specific grazing rate in units of day$^{-1}$ and $\varepsilon$ describes how fast the prey capture rates increase with the ambient phytoplankton (or prey) concentration in units of m$^6$ / (mmol N)$^{-2}$ day$^{-1}$ (Rohr et al., 2022). Phytoplankton losses to grazing then occur according to the product of g and zooplankton biomass ($B_z$, mmol N m$^{-3}$). A fraction of the grazed phytoplankton biomass is routed to zooplankton according to $gB_z\lambda$, where $\lambda$ is the assimilation efficiency (akin to gross growth efficiency). The fraction that is not assimilated into zooplankton ($1 - \lambda$) is routed to detritus ($B_d$, mmol N m$^{-3}$). Linear mortality is temperature dependent, such that

$$\gamma_p = \gamma_p^0 B_p \beta^T , \tag{A15}$$

whereas quadratic mortality is density dependent, but temperature independent, such that

$$\Gamma_p = \Gamma_p^0 B_p^2 . \tag{A16}$$

Both $\gamma_p^0$ and $\Gamma_p^0$ are constant parameters in units of day$^{-1}$ and m$^3$ mmol$^{-1}$ day$^{-1}$, respectively. Zooplankton are also affected by linear ($\gamma_z$) and quadratic ($\Gamma_z$) mortality terms of the same form according to predefined values of $\gamma_z^0$ and $\Gamma_z^0$. All quadratic mortality of phytoplankton and zooplankton biomass is routed to detritus, while linear mortality losses are routed to



dissolved nutrients and inorganic carbon. Remineralisation of detritus follows the same form as linear mortality (Eq. A15),
with its own rate controlled by the $\gamma_d^0$ coefficient, which is halved at depths below 180 metres.

Both detritus and calcium carbonate sink at constant rates. Calcium carbonate is produced at the same time as detritus (i.e., via unassimilated phytoplankton and quadratic mortality losses of phytoplankton and zooplankton), but at a constant rate controlled by $f_{inorg}$, which is equal to 6.2% of the organic detritus being produced. In contrast to the detrital pool, the calcium carbonate pool ($B_{CaCO_3}$, mmol C m$^{-3}$) is remineralised (dissolved) at a constant rate ($\gamma_{CaCO_3}$) that is not temperature
dependent.

In addition to its biological cycling, dissolved iron is affected by abiotic scavenging. This is represented implicitly via a relaxation term ($K_{scav}^{dFe}$, day$^{-1}$) to a background dFe concentration set by an assumed concentration of ligand-bound iron ($Fe_{Lig}$, μmol m$^{-3}$), where

$$\text{Scavenging} = K_{scav}^{dFe} \cdot \max(0, dFe - Fe_{Lig}). \tag{A17}$$

dFe is also reduced to a maximum of 1 μmol m$^{-3}$ at the continental shelves where the sediments are less than 200 metres deep.

## A.2 Key equations of WOMBAT-lite

The budgets for the full set of ecosystem equations of WOMBAT-lite are as follows:

$$\frac{\partial NO_3}{\partial t} = \left(\gamma_d + \gamma_z + \gamma_p - \mu B_p\right)\left(\frac{N}{C}\right), \tag{A18}$$

$$\frac{\partial dFe}{\partial t} = \left(\gamma_d Q_d^{\frac{Fe}{C}} + \gamma_z Q_z^{\frac{Fe}{C}} + \gamma_p Q_p^{\frac{Fe}{C}} - \mu_{Fe}\right) \cdot 1000 - Fe_{precip}^{dFe\to} - Fe_{scav}^{dFe\to} - Fe_{coag}^{dFe\to B_d^{Fe}}, \tag{A19}$$

$$\frac{\partial O_2}{\partial t} = \left(\mu B_p - \gamma_d - \gamma_z - \gamma_p\right)\left(\frac{O_2}{C}\right), \tag{A20}$$

$$\frac{\partial B_p}{\partial t} = \mu B_p - \gamma_p - \Gamma_p - g B_z \left(\frac{B_p}{B_{prey}}\right), \tag{A21}$$

$$\frac{\partial B_p^{Fe}}{\partial t} = \mu_{Fe} - \left(\gamma_p + \Gamma_p + g B_z \left(\frac{B_p}{B_{prey}}\right)\right) Q_p^{\frac{Fe}{C}}, \tag{A22}$$

$$\frac{\partial B_p^{Chl}}{\partial t} = \mu_{Chl} - 12 \left(\gamma_p + \Gamma_p + g B_z \left(\frac{B_p}{B_{prey}}\right)\right) Q_p^{\frac{Chl}{C}}, \tag{A23}$$

$$\frac{\partial B_z}{\partial t} = g B_z \lambda - \gamma_z - \Gamma_z, \tag{A24}$$

$$\frac{\partial B_z^{Fe}}{\partial t} = g B_z \lambda \left(\frac{B_p}{B_{prey}}\right) Q_p^{\frac{Fe}{C}} + g B_z \lambda \left(\frac{\varphi_z^d B_d}{B_{prey}}\right) Q_d^{\frac{Fe}{C}} - (\gamma_z + \Gamma_z) Q_z^{\frac{Fe}{C}}, \tag{A25}$$





$$\frac{\partial B_d}{\partial t} = \Gamma_p + \Gamma_z + gB_z(1-\lambda) - \gamma_d - gB_z\left(\frac{\varphi_z^d B_d}{B_{prey}}\right), \tag{A26}$$

$$\frac{\partial B_d^{Fe}}{\partial t} = \Gamma_p Q_p^{\frac{Fe}{C}} + \Gamma_z Q_z^{\frac{Fe}{C}} + gB_z(1-\lambda)\left(\frac{B_p}{B_{prey}}\right)Q_p^{\frac{Fe}{C}} + gB_z(1-\lambda)\left(\frac{\varphi_z^d B_d}{B_{prey}}\right)Q_d^{\frac{Fe}{C}} - \left(\gamma_d + gB_z\left(\frac{\varphi_z^d B_d}{B_{prey}}\right)\right)Q_d^{\frac{Fe}{C}}, \tag{A27}$$

$$\frac{\partial B_{CaCO_3}}{\partial t} = \left(\Gamma_p + \Gamma_z + gB_z(1-\lambda)\left(\frac{B_p}{B_{prey}}\right)\right)f_{inorg} - \gamma_{CaCO_3}, \tag{A28}$$

$$\frac{\partial DIC}{\partial t} = \left(\gamma_d + \gamma_z + \gamma_p - \mu B_p\right) - \left(\left(\Gamma_p + \Gamma_z + gB_z(1-\lambda)\left(\frac{B_p}{B_{prey}}\right)\right)f_{inorg} - \gamma_{CaCO_3}\right), \tag{A29}$$

$$\frac{\partial Alk}{\partial t} = -\left(\gamma_d + \gamma_z + \gamma_p - \mu B_p\right) - 2\left(\left(\Gamma_p + \Gamma_z + gB_z(1-\lambda)\left(\frac{B_p}{B_{prey}}\right)\right)f_{inorg} - \gamma_{CaCO_3}\right). \tag{A30}$$

### A2.1 Phytoplankton growth

Growth of phytoplankton ($B_p$) is controlled by a combination of temperature, light and nutrient supply. Temperature, T (ºC), sets the maximum potential growth rate of phytoplankton, $\mu^{max}$ (day$^{-1}$), following the Eppley curve (Eppley, 1972),

$$\mu^{max} = \alpha_a \beta_a^{(T)}, \tag{A31}$$

and where both $\alpha_a$ and $\beta_a$ (subscript a for autotrophy) are predefined (Table 1).

### A2.2 Phytoplankton growth: light limitation

Incoming shortwave radiation at the surface (W m$^{-2}$) is multiplied by 0.43 to return the photosynthetically active radiation (PAR), which is further split into three major wavelength bands associated with blue, green, and red light. Each band has unique attenuation through the water column according to the power law coefficients ($\chi$ and e) provided by Morel and Maritorena (2001) in their Table 2, which change depending on the chlorophyll concentration and implicitly account for the packaging effect of larger cells, assuming that more chlorophyll brings with it an increase in the mean community cell size. Attenuation of blue, green and red light is increased as chlorophyll concentrations increase, but as cells grow larger they absorb less light per unit chlorophyll. We calculate the depth of the euphotic zone to be the depth at which PAR is 1% of its incident intensity, resulting in typical depths between 50 and 150 metres.

The maximum potential growth rate is multiplied by a light limitation term, $L^{PAR}$, to return light-limited primary production, $\mu^{PAR}$. $L^{PAR}$ depends on the availability of PAR, the ratio of the euphotic depth to mixed layer depth ($\frac{z_{eup}}{z_{MLD}}$), the linear



mortality rate ($\gamma$, day$^{-1}$) and on the chlorophyll quota of the cell ($Q^{\frac{Chl}{C}}$, mg mg$^{-1}$). First, we solve for the initial slope of the

photosynthetic-irradiance curve, PI, which is altered by $Q^{\frac{Chl}{C}}$ such that

$$PI = PI^0 Q^{\frac{Chl}{C}} , \qquad (A32)$$

and where $PI^0$ can be altered to increase or decrease the response of phytoplankton to light. Following PI, we calculate $L^{PAR}$ via

$$L^{PAR} = \left(1 - e^{-\frac{PI \cdot PAR}{1 + \gamma p}}\right) L^{eup} , \qquad (A33)$$

$$L^{eup} = \min\left(1, \frac{z_{eup}}{z_{MLD}}\right) , \qquad (A34)$$

where $L^{eup}$ is scaled down when the mixed layer is deeper than the euphotic zone, representing the disadvantageous mixing of cells into darkness due to deep mixed layers. Light limited phytoplankton growth, $\mu^{PAR}$ (units of mmol C m$^{-3}$ day$^{-1}$), is then

$$\mu^{PAR} = \mu^{max} L^{PAR} . \qquad (A35)$$

### A2.3 Phytoplankton growth: nutrient limitation

The minimum of multiple nutrient limitation terms ($L^{nutrient}$) is then multiplied against $\mu^{PAR}$ to return the realised growth rate, $\mu$ (day$^{-1}$)

$$\mu = \mu^{PAR} L^{nutrient} = \mu^{max} L^{PAR} L^{nutrient} . \qquad (A36)$$

The limitation terms include growth limitation by nitrogen ($L^N$) and dissolved iron ($L^{Fe}$), such that

$$L^{nutrient} = \min(L^N, L^{Fe}) . \qquad (A37)$$

The nutrient limitation terms are dependent on the availability of resource, R, and the half-saturation coefficient for uptake of that resource by the phytoplankton ($K_p^R$), which itself is dependent on the biomass of phytoplankton in terms of carbon ($B_p$).

Initial $K_p^R$ values are predefined ($K_p^{R_0}$; Table 1) but are made to vary with phytoplankton biomass. We relate the biomass concentration of phytoplankton to the mean community cell size, which then affects the half-saturation coefficients for resource uptake. Using compilations of marine phytoplankton and zooplankton communities, Wickman et al. (2024) show that the nutrient affinity, aff, of a phytoplankton cell is related to its volume, V, via

$$aff = V^{-0.57} . \qquad (A38)$$



Additionally, the authors demonstrate that the volume of the average phytoplankton cell is related to the density (i.e.,

concentration; here $B_p$) of phytoplankton via

$V = B_p^{0.65}$ (combining panels c and f of their Figure 1), making                        (A39)

$aff = B_p^{-0.37}$ .                                                            (A40)

Finally, the affinity of phytoplankton for a given nutrient is proportional to the inverse of the half-saturation coefficient, $K_p^R$,

such that we can relate $K_p^R$ to the biomass concentration of phytoplankton via

$K_p^R = K_p^{R_0} B_p^{0.37}$ .                                                (A41)

For nitrogen limitation we follow the simple Monod formulation of

$L^N = \frac{N}{N + K_p^N}$ .                                                 (A42)

For iron we follow Aumont et al. (2015), who use the Droop formulation to assign growth limitation according to the quota

model (Droop, 1983; Flynn, 2003). First, a minimum cellular iron requirement in terms of an Fe:C ratio is solved for, $Q'^{\frac{Fe}{C}}$,

dependent on chlorophyll content ($Q^{\frac{Chl}{C}}$), the prescribed N:C ratio of 122:16, and nitrogen limitation terms (Flynn and

Hipkin, 1999).

$Q'^{\frac{Fe}{C}} = \left(\frac{0.0016}{55.85} 12 Q^{\frac{Chl}{C}}\right) + \left(1.21 \times 10^{-5} \frac{14}{55.85} \frac{N}{C} \cdot 0.5 \cdot 1.5 L^N\right) + \left(1.15 \times 10^{-4} \frac{14}{55.85} \frac{N}{C} \cdot 0.5 \, L^N\right)$ .     (A43)

Limitation of growth by iron ($L^{Fe}$) is then calculated as the difference between the current quota ($Q^{\frac{Fe}{C}}$) and the minimum

requirements of the cell ($Q'^{\frac{Fe}{C}}$) divided by a predefined optimal iron quota ($Q^{*\frac{Fe}{C}}$) assigned according to estimates from the

literature (Hopkinson et al., 2013; Strzepek et al., 2012; Sunda et al., 1991; Twining et al., 2021), such that

$L^{Fe} = \min\left(1, \max\left(0, \frac{\left(Q^{\frac{Fe}{C}} - Q'^{\frac{Fe}{C}}\right)}{Q^{*\frac{Fe}{C}}}\right)\right)$ .                                   (A44)

**A2.4 Phytoplankton growth: chlorophyll**

Chlorophyll concentration in phytoplankton ($B_p^{Chl}$, mg m⁻³) is explicitly considered as a tracer in WOMBAT-lite (Eq. A23).

It has a direct influence on phytoplankton growth. The concentration of chlorophyll in the water column increases light

attenuation, affecting light availability, and the chlorophyll quota of phytoplankton ($Q^{\frac{Chl}{C}} = \frac{B_p^{Chl}}{B_p}$, mg Chl (mg C)⁻¹) influences



the slope of the photosynthetic-irradiance curve, PI (Eq. A32). Also, an elevated chlorophyll quota increases the iron demand
of phytoplankton (Eq. A43). Phytoplankton attempting to reduce light limitation through photoacclimation therefore have a
higher iron demand.

Growth of chlorophyll occurs similarly to growth of biomass carbon but with its own light limitation term, $L_{Chl}^{PAR}$, where

$$L_{Chl}^{PAR} = \left(1 - e^{-\left(\frac{PI \cdot PAR_{MLD}}{\mu^{max}(1 - L^{nutrient})}\right)}\right) L^{eup} \, . \tag{A45}$$

Note the adjustments to $L_{Chl}^{PAR}$ (Eq. A45) relative to $L^{PAR}$ (Eq. A33). These adjustments are responsible for photoacclimation.
They increase chlorophyll accumulation in phytoplankton at depth, where there is less light, waters are cooler and where
there are more nutrients, but cause chlorophyll depletion near the surface in warm oligotrophic waters.

We step through why this is the case. $PAR_{MLD}$ is the average availability of light within the mixed layer. $PAR_{MLD}$ is
therefore less than PAR near the surface, but greater than PAR towards the bottom of the mixed layer. Beneath the mixed
layer $PAR_{MLD} = PAR$. That chlorophyll sees $PAR_{MLD}$ makes light limitation of chlorophyll stronger than phytoplankton near
the surface where $PAR_{MLD} < PAR$, but weaker than phytoplankton limitation at depth where $PAR_{MLD} > PAR$. Additionally,
the placement of $\mu^{max}(1 - L^{nutrient})$ in the denominator decreases $L_{Chl}^{PAR}$ more so than respiration $(1 + \gamma_p$ in Eq. A33)
decreases $L^{PAR}$ for phytoplankton in warm waters (where $\mu^{max}$ is high) and in nutrient deplete waters (where $(1 - L^{nutrient})$
is high).

Growth of chlorophyll is then calculated similarly to growth of phytoplankton, where a maximum growth rate is scaled down
by limitations associated with light and nutrient availability. However, chlorophyll production is affected by the minimum
$(Q'^{\frac{Chl}{C}})$ and optimal $(Q^{*\frac{Chl}{C}})$ quotas (units mg Chl (mg C)$^{-1}$). Minimum and optimal chlorophyll production rates, $\mu_{Chl}^{min}$ and
$\mu_{Chl}^{opt}$ (mg Chl m$^{-3}$ day$^{-1}$), ensure that phytoplankton chlorophyll growth always stays within specified bounds, where

$$\mu_{Chl}^{min} = \mu \, B_p \, 12 \, Q'^{\frac{Chl}{C}} \, , \text{ and} \tag{A46}$$

$$\mu_{Chl}^{opt} = \mu \, B_p \, 12 \, Q^{*\frac{Chl}{C}} \, . \tag{A47}$$

Chlorophyll growth rate is then calculated as

$$\mu_{Chl} = \mu_{Chl}^{min} + \frac{\left(\mu_{Chl}^{opt} - \mu_{Chl}^{min}\right) L_{Chl}^{PAR} L^{nutrient}}{PI \cdot PAR_{MLD}} \, , \tag{A48}$$

where we include the light response in the denominator to further accelerate chlorophyll growth in low light environments
and depress it in high light environments. This effectively increases or decreases the maximum quota that is attainable by a



phytoplankton cell around its optimal quota that is predefined ($Q^{*\frac{Chl}{C}}$, Table 1). Losses of chlorophyll occur in the same way

as losses of phytoplankton but are multiplied by the chlorophyll quota (Eq. A23).

**A2.5 Phytoplankton growth: iron uptake**

Like chlorophyll, the iron content of phytoplankton ($B_p^{Fe}$, mmol m$^{-3}$) is explicitly tracked as a tracer in WOMBAT-lite (Eq.

A22). First, an uptake rate is found dependent on the maximum quota of Fe within the phytoplankton type ($Q^{''\frac{Fe}{C}}$) and the

maximum phytoplankton growth rate via

$$\mu_{Fe}^{max} = \mu^{max} B_p Q^{''\frac{Fe}{C}} .$$ (A49)

Following Aumont et al. (2015), this rate is scaled by three terms relating to (i) michaelis-menten type affinity for dFe, (ii)

up-regulation of dFe uptake representing investment in transporters when cell quotas are limiting to growth, and (iii) down-

regulation of dFe uptake associated with enriched cellular quotas:

    i.    $\frac{dFe}{dFe + K_p^{dFe}}$ ,

            (A50)

    ii.    $4 - \frac{4.5 L^{Fe}}{0.5 + L^{Fe}}$ , (A51)

    iii.    $\max\left(0, 1 - \frac{\frac{Q^{\frac{Fe}{C}}}{Q^{''\frac{Fe}{C}}}}{abs\left(1.05 - \frac{Q^{\frac{Fe}{C}}}{Q^{''\frac{Fe}{C}}}\right)}\right)$ , (A52)

such that dFe uptake by phytoplankton is simulated as

$$\mu_{Fe} = \mu_{Fe}^{max} \cdot \left(\frac{dFe}{dFe + K_p^{dFe}}\right) \cdot \left(4 - \frac{4.5 L^{Fe}}{0.5 + L^{Fe}}\right) \cdot \max\left(0, 1 - \frac{\frac{Q^{\frac{Fe}{C}}}{Q^{''\frac{Fe}{C}}}}{abs\left(1.05 - \frac{Q^{\frac{Fe}{C}}}{Q^{''\frac{Fe}{C}}}\right)}\right) .$$ (A53)

The iron to carbon ratios of phytoplankton are passed to zooplankton and detritus and are also tracked in these pools.

**A2.6 Zooplankton grazing**

Grazing is represented as a Holling Type III function of prey biomass (Holling, 1959). This choice assumes that at very low

prey concentrations grazing is impaired by increased searching (i.e., slower clearance rate (Gentleman and Neuheimer,

2008)); at moderate prey concentrations zooplankton grazing accelerates exponentially to account for their learning to feed





on a growing prey source; at high prey density zooplankton handling time becomes the limiting factor (Gentleman and Neuheimer, 2008; Rohr et al., 2022). This formulation allows for greater stability in the ecosystem and elongates the phytoplankton spring bloom. The Holling Type III formulation requires two basic parameters to estimate grazing, g, in units

of day$^{-1}$. These are the maximum grazing rate, $g^{max}$ (day$^{-1}$), and a prey capture rate coefficient, $\varepsilon$ (m$^6$ (mmol C)$^{-2}$ day$^{-1}$). The grazing formula is

$$g = \frac{g^{max}\varepsilon B_{prey}^2}{g^{max}+\varepsilon B_{prey}^2} \ . \tag{A54}$$

Both WOMBAT-lite and legacy WOMBAT therefore use the same grazing formulation. However, an important distinction is that the maximum grazing rate, $g^{max}$, is now made dependent on temperature (T, in °C) according to

$g^{max} = g_h\beta_h^{(T)} ,$ $\qquad\qquad\qquad\qquad\qquad\qquad\qquad\qquad\qquad\qquad$ (A55)

where both $g_h$ and $\beta_h$ (subscript h for heterotrophy) must be predefined (Table 1). The application of $g^{max}$ in both the numerator and denominator make this grazing formula unique (Rohr et al., 2023) and equivalent to a disk formulation, rather than a Michaelis Menten formulation (Rohr et al., 2022). Practically, this amplifies grazing in warmer climes, but to a lesser extent than other formulations that apply the temperature amplification (i.e., $\beta_h^{(T)}$) only in the numerator of Eq. A54 (Rohr et

al., 2023). This dampens the effect that variations in temperature have on grazing activity, amplifying the effect of $\varepsilon$, and aligning with observations that the ratio of grazing to phytoplankton growth varies little between tropical and polar climes (Calbet and Landry, 2004). Theoretically, this assumes some evolutionary adaptation to account for the physiological effects of temperature across environmental niches, such that the efficiency of prey capture and handling become more important to grazers than metabolic constraints due to temperature.


Both phytoplankton ($B_p$) and detritus ($B_d$) contribute to the available prey biomass ($B_{prey}$), scaled by the preference of zooplankton for these prey types ($\varphi_z^p$, $\varphi_z^d$; Table 1), such that

$B_{prey} = \varphi_z^p B_p + \varphi_z^d B_d \ .$ $\qquad\qquad\qquad\qquad\qquad\qquad\qquad\qquad\qquad$ (A56)

Finally, an assimilation efficiency, $\lambda$, controls how efficiently prey biomass is ingested by the grazer, with the 25% of remainder being lost to the environment as detritus (i.e., sloppy feeding) and 75% as inorganic nutrients (i.e., excretion). For our experiments, $\lambda$ is set to 0.6 to align with measurements of gross growth efficiency from the literature (Anderson et al., 2021b), such that sloppy feeding is 10% and excretion is 30% of what is grazed. Variations in gross growth efficiency and excretion associated with food quality and quantity (Anderson et al., 2021b) will be considered in a future version of

WOMBAT-lite.



## A2.7 Non-grazing losses of biomass

Phytoplankton, zooplankton and organic detrital functional types are affected by both linear ($\gamma_p$ and $\gamma_z$) and quadratic ($\Gamma_p$ and $\Gamma_z$) mortality coefficients. These terms are of the form


$$\gamma = \gamma^0 \beta_h^{(T)} B \text{ , and} \tag{A57}$$

$$\Gamma = \Gamma^0 B^2 \text{ ,} \tag{A58}$$

Where $\gamma^0$ and $\Gamma^0$ are predefined (Table 1) and are different for different biomass pools (i.e., $\gamma_p^0 \neq \gamma_z^0 \neq \gamma_d^0$), and $\beta_h^{(T)}$ is a temperature-dependent amplifier for heterotrophic processes (Table 1). The linear mortality term for phytoplankton emulates thermally dependent losses of biomass that escalate as a greater proportion of the phytoplankton community is pushed above

their thermal niche (Baker and Geider, 2021). These losses are also associated with increased respiration and efflux, for instance of exopolymers (Bar-Zeev et al., 2013; Thornton, 2014), that route biomass to the inorganic nutrient pool. The quadratic mortality term emulates density-dependent losses of phytoplankton biomass that are not accounted for by grazing, for instance due to viral lysis (Brussaard et al., 2008; Suttle, 1994), and is not thermally dependent but density dependent. These quadratic losses are routed to sinking detritus.


Linear mortality for zooplankton represents rates of respiration (i.e., losses to inorganic nutrients) that are thermally dependent due to metabolism scaling positively with temperature (Ikeda, 1985; Ikeda et al., 2001), while the quadratic mortality closure term represents density-dependent predation by higher trophic levels not included in the model. As for phytoplankton, linear and quadratic loss terms are routed to inorganic nutrients and sinking detritus, respectively. For

zooplankton, the linear losses associated with respiration are reduced by a Michaelis-Menten function of zooplankton biomass,

$$\gamma_z = \gamma_z^0 B_z \left( \frac{B_z}{B_z + K_z^\gamma} \right) \text{ ,} \tag{A59}$$

such that in environments with little zooplankton their losses are reduced. This additional term ensures more stable population dynamics of zooplankton, and ecologically, mimics the greater prevalence of gelatinous zooplankton (tunicates,

ctenophores, cnidarians, etc.) in oligotrophic regions with lower metabolic demands.

Detritus undergoes remineralisation at a rate that is linearly dependent on the concentration of detritus (Eq. A57). There are no quadratic losses (Eq. A58; i.e., no explicit variations in bacterial biomass are considered) but remineralisation rates are halved beneath 200 metres depth to account implicitly for more intense heterotrophic bacterial activity in the upper ocean.




**A2.8 Sinking detritus**

One sinking detrital pool $B_d$ is considered. One quarter of the prey that is not assimilated into zooplankton biomass (i.e. sloppy feeding) and the product of quadratic mortality of both phytoplankton ($\Gamma_p B_p^2$) and zooplankton ($\Gamma_z B_z^2$) are routed to this detrital pool. Since we only account for one detrital pool, we consider that base sinking rates of this detritus ($\omega^0$, metres day⁻¹) are varied as a function of phytoplankton concentration (in a similar fashion to half-saturation coefficients described earlier). This approach is to emulate observations of varying sinking speeds (Riley et al., 2012) and that such variations may be strongly dependent on phytoplankton community composition (Bach et al., 2016).

In accordance with a more general Navier-Stokes drag equation and using a compilation of particle sinking speeds, Cael et al. (2021b) identified that the sinking velocity of particles ($\omega$, meters per day) is proportional to their diameter raised to the power of roughly 0.63, such that

$$\omega \propto d^{0.63} . \tag{A60}$$

Knowing that $d = \left(\frac{6V}{\pi}\right)^{\frac{1}{3}}$ and given that the average volume of phytoplankton cells can be approximated by $V = B_p^{0.65}$ (Wickman et al., 2024), we can relate $\omega$ to the biomass concentration of phytoplankton multiplied by the scaler $\omega^0$:

$$\omega = \omega^0 (B_p)^{0.21} . \tag{A61}$$

This formula is identical to that presented by Cael et al. (2021b) in their Eq. 3, with the exception that we have related sinking rates to the biomass concentration of phytoplankton ($B_p$) by assuming that $V = B_p^{0.65}$ (Eq. A26) based on marine phytoplankton data (Wickman et al., 2024).

However, phytoplankton concentrations are negligible beneath the euphotic zone. Using in situ $B_p$ would therefore result in negligible sinking speeds throughout most of the dark ocean. To address this, we use $B_p$ in the upper-most grid cell (k=1) within Eq. A61 ($B_p^{k=1}$). This assumes that the sinking velocities of marine aggregates can be related to phytoplankton community composition (Bach et al., 2016; Iversen and Lampitt, 2020), which varies more horizontally across the ocean than vertically. Moreover, because we do not include dissolved/suspended organic matter as a tracer in WOMBAT-lite, we must also account for the large fraction of organics that are suspended and thus neutrally buoyant in the gyres. As such, we vary Eq. A61 to include a phytoplankton biomass threshold ($B_p^{thresh}$, in mmol C m⁻³) above which sinking accelerates and beneath which any produced detritus emulates dissolved (neutrally buoyant) organic matter:

$$\omega = \omega^0 \cdot \max\left(0.0, B_p^{k=1} - B_p^{thresh}\right)^{0.21} . \tag{A62}$$



Finally, we apply a linear increase to sinking speeds with depth to ensure that the trend in the concentration of detritus with depth exhibits a power-law behaviour, which is widely observed (Berelson, 2001; Martin et al., 1987), thought to be associated with a greater attenuation of slower sinking particles, and shows better performance than a constant sinking rate in models (Tjiputra et al., 2020). This is applied after Eq. A62 as

$$\omega = \omega + \max\left(0.0, \frac{\text{depth}}{5000} \cdot (\omega^{\max} - \omega)\right). \tag{A63}$$

**A2.9 Sinking CaCO₃**

No changes have been made to CaCO$_3$ dynamics in WOMBAT-lite, except for permanent burial in the sediments (see section A.2.11) and changes to the parameter values (Table 1). Calcium carbonate (CaCO$_3$) is produced alongside organic detritus but scaled according to the CaCO$_3$ to organic detritus ratio, $R_{\frac{CaCO_3}{detritus}}$. It sinks at a predefined rate of 6 metres per day ($\omega^0_{CaCO_3}$) and dissolves at a temperature-independent rate of 0.01 day$^{-1}$ ($\gamma^0_{CaCO_3}$).

**A2.10 Iron cycling outside the ecosystem component**

Treatment of dissolved iron (dFe, μmol m$^{-3}$) follows Aumont et al. (2015). Equilibrium concentrations of ligand-bound dissolved iron (Fe$_{Lig}$, μmol m$^{-3}$) and free iron (Fe′, μmol m$^{-3}$) are estimated using the concentration of ligand (Lig, μmol m$^{-3}$) and an equilibrium constant (K$^{Fe}_{eq}$, μmol m$^{-3}$) dependent on temperature in degrees Kelvin (T$_K$ = T + 273.15)

$$K^{Fe}_{eq} = 10^{17.27 - \frac{1565.7}{T_K}} \cdot 1 \times 10^{-9}, \tag{A64}$$

$$Fe_x = 1 + K^{Fe}_{eq} Lig - K^{Fe}_{eq} dFe, \tag{A65}$$

$$Fe' = \frac{-Fe_x + \sqrt{(Fe_x)^2 + 4K^{Fe}_{eq} dFe}}{2K^{Fe}_{eq}}. \tag{A66}$$

Ligand-bound dissolved iron is then the difference between total dissolved iron in seawater and free iron

$$Fe_{Lig} = dFe - Fe'. \tag{A67}$$

Following the equilibrium partitioning of dFe into Fe′ and Fe$_{Lig}$, we estimate losses of dFe due to precipitation, scavenging and coagulation. These processes work to increase the quota of iron within the detritus as it sinks deeper, emulating observations of increased iron content of particulates with increasing depth (Bressac et al., 2019).



Precipitation of Fe′ onto nanoparticles ($Fe_{precip}^{dFe\rightarrow}$), which represents a permanent loss of dFe from the model domain (hence

the superscript dFe →), occurs when the concentration of Fe′ is in excess of the solubility of Fe(III) in solution, $Fe(III)_{sol}$.

This solubility is calculated using experimentally derived coefficients ($c_{1-5}^{Fe(III)}$) dependent on temperature, $T_K$ (ºK), salinity,

sal (psu), and pH:

$$sal_x = \frac{19.924\, sal}{1000 - 1.005\, sal}\,,$$ (A68)

$$c_1^{Fe(III)} = 10^{-13.486 - 0.1856\sqrt{sal_x} + 0.3073 sal_x + \frac{5254}{T_K}}\,,$$ (A69)

$$c_2^{Fe(III)} = 10^{2.517 - 0.8885\sqrt{sal_x} + 0.2139 sal_x + \frac{1320}{T_K}}\,,$$ (A70)

$$c_3^{Fe(III)} = 10^{0.4511 - 0.3305\sqrt{sal_x} - \frac{1996}{T_K}}\,,$$ (A71)

$$c_4^{Fe(III)} = 10^{-0.2965 - 0.7881\sqrt{sal_x} - \frac{4086}{T_K}}\,,$$ (A72)

$$c_5^{Fe(III)} = 10^{4.4466 - 0.8505\sqrt{sal_x} - \frac{7980}{T_K}}\,,$$ (A73)

$$Fe(III)_{sol} = c_1^{Fe(III)}\left(pH^3 + c_2^{Fe(III)}pH^2 + c_3^{Fe(III)}pH + c_4^{Fe(III)} + \frac{c_5^{Fe(III)}}{pH}\right).$$ (A74)

Precipitation of nanoparticles is then estimated as that in excess of solubility, multiplied by a parameterised rate, $K_{nanop}^{Fe}$

(day$^{-1}$), via:

$$Fe_{precip}^{dFe\rightarrow} = \max(0, Fe' - Fe(III)_{sol})\, K_{nanop}^{Fe}\,.$$ (A75)

Scavenging of Fe′ onto particles ($Fe_{scav}^{dFe\rightarrow}$) is linearly dependent on the concentration of particles in solution, which we

estimate roughly as the sum of detrital carbon and calcium carbonate:

$$particles = B_d + B_{CaCO_3}\,.$$ (A76)

Scavenging then occurs at a parameterised rate controlled by $K_{scav}^{Fe}$ ((mmol C m$^{-3}$)$^{-1}$ day$^{-1}$) via

$$Fe_{scav}^{dFe\rightarrow} = Fe'\left(3 \times 10^{-5} + K_{scav}^{dFe} particles\right).$$ (A77)

$Fe_{scav}^{dFe\rightarrow}$ represents the total scavenging rate of Fe′, which is subtracted from the dFe pool. However, the proportion of

scavenging due to detrital particles ($Fe_{scav}^{dFe\rightarrow B_d^{Fe}}$) is apportioned to detrital iron ($B_d^{Fe}$) according to the proportion of $B_d$ in the

total mass of particles.



Coagulation of colloidal iron, which represents a transfer of dissolved iron to detrital iron, is estimated as linearly dependent

on the concentration of detrital organic carbon. First, we assume that half of all $Fe_{Lig}$ is colloidal, where

$$Fe_{coll} = \frac{Fe_{Lig}}{2}.$$ (A78)

The concentration of colloidal iron is multiplied by the coagulation rate, $K_{coag}^{Fe}$ ((mmol C m$^{-3}$)$^{-1}$ day$^{-1}$) and the concentration

of detrital carbon to control the transfer of $Fe_{coll}$ to the detrital iron pool ($B_d^{Fe}$):

$$Fe_{coag}^{dFe \rightarrow B_d^{Fe}} = Fe_{coll}(0.001 + B_d K_{coag}^{Fe}).$$ (A79)

Colloidal aggregation rates are scaled down by a factor of 100 beneath the mixed layer to reflect a reduction in encounter

rates associated with reduced mixing.

Finally, we elevate scavenging in waters close to the coasts, specifically in waters shallower than 200 metres, by setting dFe

to a maximum of 1.0 µmol m$^{-3}$ in these environments. Precipitation, scavenging and colloidal coagulation rate constants are

described in Table 1.

### A2.11 Boundary fluxes

WOMBAT-lite has been updated to include boundary fluxes from rivers, hydrothermal vents and burial in the sediments.

Riverine fluxes are annually repeating climatologies of dissolved inorganic carbon (DIC), alkalinity and nitrate. For nitrate,

the flux is based on GLOBAL-NEWS2 (Mayorga et al., 2010) and combines their estimates of inorganic and organic

nitrogen loads at a total of 35.8 Tg N yr$^{-1}$. For DIC the fluxes are based on Ludwig et al. (1996) and amount to a total of

0.587 Pg DIC yr$^{-1}$. Alkalinity is added at a rate of 0.216 Pg C equivalents yr$^{-1}$ to correct for a long-term positive trend in

global alkalinity. Hydrothermal fluxes include constant release of dissolved iron to the ocean at a rate of 9.9 Gmol Fe yr$^{-1}$

(Tagliabue et al., 2014a). The iron cycle in legacy WOMBAT already includes a monthly climatology of atmospheric flux of

dissolved iron from Mahowald et al. (2005) at a rate of 1.1 Gmol Fe yr$^{-1}$, and this has not been updated in this version.

We consider burial of detrital organic carbon, nitrogen, iron and CaCO$_3$. The fraction of incoming organic matter (C, N and

Fe) and inorganic matter (CaCO$_3$) that is permanently buried in the sediments ($f_{bury}$) and therefore removed from the model

domain is computed according to the metamodel of Dunne et al. (2007), where

$$f_{bury} = 0.013 + 0.53 * \frac{(F_{org})^2}{(7 + F_{org})^2}.$$ (A80)

Here, $F_{org}$ is the flux of detrital organic carbon to the sediments in units of mmol C m$^{-2}$ day$^{-1}$. The fraction of organic and

inorganic matter that is not buried is routed to a labile sedimentary pool, which is acted upon by remineralisation/dissolution

at each timestep. Sedimentary pools of iron and calcium carbonate are tracked explicitly, such that the





remineralisation/dissolution of this material to the overlying water occurs at ratios with detrital organic carbon and nitrogen that differ spatially and temporally. The remineralisation of sedimentary detrital organic carbon, nitrogen, iron and calcium carbonate is computed according to temperature via Eq. A51, with the exception that the $\alpha_h$ terms scaling the rate of remineralisation are equal to 0.02 day$^{-1}$ and 0.0035 day$^{-1}$ for organics (C, N and Fe) and inorganics (CaCO$_3$), respectively.

A fraction of the sedimentary organic matter that is remineralised is remineralised anaerobically, specifically via denitrification. We compute this fraction ($f_{denit}$) using the metamodel of Middelburg et al. (1996), where

$$f_{denit}^{log} = -2.2567 - 1.185 \log_{10}\left(F_{org}\right) - 0.221\left(\log_{10}\left(F_{org}\right)\right)^2 - 0.3995 \log_{10}\left(NO_3^-\right) \log_{10}(O_2) + 2 \log_{10}\left(NO_3^-\right) +$$
$$0.04721 \log_{10}(O_2) - 0.0996 \log_{10}(z) + 0.4256 \log_{10}\left(F_{org}\right) \log_{10}(O_2) , \tag{A81}$$

$$f_{denit} = 10^{f_{denit}^{log}} \cdot \frac{NO_3^-}{NO_3^- + 1} , \tag{A82}$$

$$\text{denitrification} = f_{denit} \left(\frac{94.4}{122}\right) . \tag{A83}$$

Here, $NO_3^-$ is the concentration of nitrate in mmol m$^{-3}$, $O_2$ is the concentration of oxygen in mmol m$^{-3}$ and z is the depth in metres. Eq. A82 differs slightly from that of Middelburg et al. (1996) in that we accelerate sedimentary denitrification when nitrate concentrations are high by scaling by 2 instead of 1.25 in the 5$^{th}$ term. Eq. A84 includes the theoretical stoichiometry

of the denitrification nitrate demand of 94.4/122 (Paulmier et al., 2009). To ensure balance in the nitrogen cycle, we track the total rate of denitrification at each timestep and add this evenly to surface waters. This proxy for nitrogen fixation will be updated in future versions to account for environmental conditions favourable for diazotrophs, ensuring a more realistic distribution. When nitrate is consumed or added to the ocean model via denitrification and nitrogen fixation, respectively, we add and remove alkalinity in equal measure.

**Appendix B**

**B.1 Interior tracer distributions**

One hundred years of simulation is sufficient to understand the impacts of WOMBAT-lite on the distributions of key tracers at mesopelagic depths. Distributions of nitrate, dissolved iron, oxygen, dissolved inorganic carbon (DIC) and alkalinity at 500 metres depth are shown in Figure S5. Nitrate, DIC and alkalinity all show accumulation in the eastern Pacific that is well

in excess of the observations, and coincident severe oxygen depletion in this region. Despite this and other biases, WOMBAT-lite shows some slight improvements in the distribution of these tracers within the mesopelagic. Oxygen is slightly more depleted in the subarctic North Pacific in WOMBAT-lite consistent with the observations. Alkalinity distributions in WOMBAT-lite show less bias compared with legacy WOMBAT, although the North Atlantic and Indian Oceans still have too much alkalinity compared with the observations. The biggest change has been to the iron cycle, which



now shows less uniform interior distributions. Concentrations of dFe are highest in the tropics and subarctic North Pacific and this is consistent with the broad patterns observed (Huang et al., 2022). However, there is too much dFe in the mesopelagic of WOMBAT-lite, meaning that the ferricline is placed too shallow. This bias is most important in the Southern Ocean, where concentrations of dFe should be low even at 500 metres but instead are too high in WOMBAT-lite. Despite overestimating dFe in the mesopelagic Southern Ocean, primary production in the Southern Ocean is still dominantly limited

by iron during the spring and summer (Fig. 8t). This suggests that future versions of WOMBAT-lite may benefit from elevating the iron scavenging term on particles to help strip out the excess dFe in the interior.