# Peer review of "Optimisation of the World Ocean Model of Biogeochemistry and Trophic-dynamics (WOMBAT) using surrogate machine learning methods."

_EGUsphere, 2024_

## Referee Comment (RC1)

**Reviewer's comments on Buchanan et al. "*Optimisation of the World Ocean Model of Biogeochemistry and Trophic-dynamics (WOMBAT) using surrogate machine learning methods*"**

*March 17, 2025*
* * *
**Overview**

**Research summary**

**(1)** In this paper, the authors introduce a new version of the marine biogeochemical model, WOMBAT-lite. The updated model incorporates several new processes and requires tuning using observational datasets. The authors leverage machine learning methods to optimize the model. First, a machine learning model is trained on outputs from 512 experiments, each with different parameter sets (same parameters but different values), to predict 8 root mean square errors (RMSEs) relative to observational products. This trained machine learning model is used to perform a sensitivity analysis, identifying the parameters that most significantly impact the RMSEs — a total of 11 parameters are identified. Then, a second machine learning model is trained using outputs from 512 new experiments, focusing solely on the 11 selected parameters, to replicate a cost function that measures the deviation from observations. This trained model is utilized for optimization analysis to determine the optimal values and ranges of the 11 parameters. The optimized version of the model is then compared to observational data and a previously unoptimized version of WOMBAT. The new optimization approach proves to be successful, with the optimized model showing improvements across all evaluated variables, albeit to varying degrees.

**Overall assessment**

**(2)** This technical paper is clearly written and presents a novel method for optimizing the new version of the biogeochemical model. The authors have made great efforts to describe the machine learning approach, detailing the sequential steps used first to identify key parameters and then to optimize them. I have a few major comments, that summarize as: 1) Additional justifications are needed to ensure that ten years of simulations are long enough, in relation to the model's intended usage and the influence of the optimization strategies. 2) Further discussion on alternative model tuning approaches would be beneficial. 3) The advantages of this optimization approach over simply selecting the best of the 512 experiments need to be further explained. 4) Further details on the machine learning algorithms may be necessary. I elaborate on these major comments below. Most of these comments can probably be addressed by adding text in the introduction or conclusive section of the manuscript. Additionally, I have a list of minor or specific comments and suggestions. Finally, given the length of the manuscript and the number of figures, any content that could be shortened would be beneficial. I tried to provide some suggestions. Overall, this is a quality piece of work, and I recommend it for publication after some minor revisions.
* * *
**Major comments**

**(3)** My first concern is the ten-year-long simulations used for conducting the sensitivity/optimization analysis. I understand the constraints associated with running a large number of longer simulations. Nonetheless, I would like to have a clearer picture of the equilibrium achieved in ten years. What are the drifts at various depths? Although ten years may be enough for the surface ocean, what about the deeper ocean that can impact the surface on longer time scale? The authors have noted a drift away from the optimal fit when simulations are extended to 100 years. Could the authors discuss the risk of further drift in these simulations? Additionally, is the drift of the optimized simulation weaker compared to the best initial simulation or any of the 512 simulations? If the drift is indeed weaker, this would be a strong point in favor of the optimization strategy.

**(4)** Somewhat connected with the previous point, could the authors specify/discuss the intended usage for which the model is optimized? For instance, will the model perform equally well when incorporated into an Earth System model for climate simulations? I am curious whether the optimization based on ten-year simulations and focusing on the surface ocean would remain effective during the longer processes, such as a thousands of years spin-up to achieve preindustrial equilibrium, followed by hundreds of years for simulating historical and future climate conditions.

**(5)** Another major comments reside in emphasizing the utility of this optimization approach. Considering that the ML approach will identify the optimal parameter set in the space that has been explored, why not using the best performing of the 512 experiment? Could the authors comment a bit around that? I guess the optimization approach also provide some uncertainty and sensitivity quantification but can't we get the same from the 512 experiments? What about comparing the optimal simulations with the best of the 512 initial simulations?

**(6)** I would appreciate further discussion on model tuning and alternative approaches, particularly those employing ensembles/surrogate models/emulators. For instance, the work of Singh et al. (2025) and Williamson et al. (2017) seem relevant.

**(7)** Last major comment, while the approach is explained very well, I believe it lacks a detailed description of the machine learning techniques used. This aspect is beyond my area of expertise, so I cannot adequately assess it. However, a thorough explanation might be essential for reproducibility and for the benefit of readers interested in machine learning methods applied to a specific problem.
* * *
**Minor and specific comments**

**Introduction**

**(8)** l. 47: Reformulation suggestion for clarification: "This component represents the growth..." -> "This component, in its simplest form, represents the growth..."

**(9)** l. 96: There is a typo, remove "apt" in "As an apt example, ..."

**Methods**

**(10)** l. 163-179: I first counted 7 observation products before understanding that chlorophyll data where actually providing 2 product: depth of the maximum and surface chlorophyll. Can the authors reformulate to clarify?

**(11)** l. 172: Why optimizing to the $CO_2$ flux and not $pCO_2$? The $CO_2$ flux is itself estimated from $pCO_2$, thus introducing an additional source of potential errors in the observational target. Can the authors clarify/justify the choice?

**(12)** l. 225: Can the authors gives some example and eventually some reference (if they exists) to clarify what they have in mind?

**(13)** l. 239: I find misleading using "Southern Ocean" for all the ocean south of 20°S. Here it includes some of the subtropical gyres. I found this misleading in some other part of the manuscript. Can the authors use an other term?

**(14)** l.271: For clarifying "... by comparing GPR predictions with WOMBAT-lite RMSE...", the authors can be specific and say that they compare the GPR predictions of RMSE with WOMBAT-lite RMSE.

**(15)** l. 267-268: Is there one training for the 8 RMSE or 8 training? i.e 1 GPR model predicting 8 RMSE or 8 GPR model? Can the authors clarify?

**(16)** l. 276: Why not doing the optimization directly with the 24 parameters? Why having first a sensitivity analysis? (ie. why not doing step 10 just after step 5?) Can the authors clarify?

**(17)** l.285-297: Similar to comment 15, can the authors clarify? Is there one training for the 8 cost functions? Or one for the global cost function? Or 8 training for each cost function? To be entirely clear, the model is trained on a cost function (defined in Eq. 1 or 2), whereas for the sensitivity analysis, it focuses on the RMSE. Is that correct?

**Results**

**(18)** l. 349-356: Why not including $PI^0$ in the most important parameters? On figure 5b it has a value of 0.05 for the depth of chlorophyll maximum, i.e. 5% if I understood correctly the threshold. Can the authors clarify?

**(19)** l. 405: The depth of the chlorophyll maximum is indeed heavily influenced by $\alpha_a$ and $\gamma_p^0$ but only "in interaction" (i.e. panel b of Figure 5). It is the only target (with maybe iron but it is not as marked) that have 2 parameters standing out so much in interaction while no parameters seems to strongly influence it at first order. Can the authors comment one that?

**(20)** l. 412: It reads a bit strange to mention 2 additional parameters and not have them specified right away. There are mentioned just a bit after, but maybe the authors can slightly reformulate the paragraph so that 2 additional parameters are specified right after being mentioned.

**(21)** l. 413-418: Any reason why these parameters where not included in the sensitivity analysis? Can the authors clarify?

**(22)** l. 437: "... due to its complexation with organics..." is not clear to me. What is organics? Organic form of dFe? Can the authors clarify?

**(23)** l. 506: In the subtropical or subpolar gyres?

**(24)** l. 504-506: It seems to me there is a very shallow chlorophyll maximum in the subtropical gyres that is not present in the observation (Fig. 8k, l). Any explanation? This is a bit contradictory with the "too deep maxima in the gyres." Can the authors clarify?

**(25)** l. 486-506: What about the CO2 flux? Can the authors mention that it receive a detailed analysis further in the paper.

**(26)** l. 483-484: "Our optimised WOMBAT-lite manages to show some improvement..." I don't find Figure 10b to be the most effective way to illustrate this. Instead, Figure 11 demonstrates the improvement in the seasonality of the CO2 flux through better local correlation, whereas the correlation on Figure 10b actually decreases. I agree nonetheless that on figure 10 the amplitude seems to be a bit closer. It also seems to me that comparing a transition zone based on average is rather complex. Could the authors reformulate to provide a more precise explanation/description?

**(27)** l. 593-599: This paragraph seems to be a repetition of the the one just before. Can the authors merge it and combine figure 10 and 11 (maybe not all panel of figure 10)?

**(28)** l. 593: The improvement are clear for the equatorial band (10S to 10N) and the high latitude (>50N and <30S). For the rest (10S-30S and 10N-50N) the unoptimized model is already well correlated with the observations and even better (darker red) in a few places. Can the authors reformulate to provide a more precise explanation/description?

**(29)** l. 593-599: Can the authors mentioned that the observations also have some caveats (Hauck et al., 2023; Gloege et al., 2021)?

**(30)** l. 602: "the unoptimised version of the model exceeded the observations." The optimized version also exceed the observation on figure 12a. Can the authors clarify their point?

**(31)** l.601-611: On figure 12 we see that the optimized version has a much better seasonal cycle in the high latitude (in line with what we see on figure 11). This important improvement does not appear so much on the annual average because of compensation between the errors in summer and winter. This should be stated more clearly.

**Appendix**

**(32)** l. 1430: There is typo, "is remineralised" is repeated.

**Figures**

**(33)** Fig. 1: In the text of step (6) in the figure, consider to keep the same format as the other steps. Maybe something like "Run a global sensitivity..."

**(34)** Fig. 4: Can the authors add the performance of the optimized model and the 20 optimal experiments after 10 and 100 years.

**(35)** Fig. 4: Why is the standard deviation of sinking detritus, NPP and chlorophyll so weak for many experiments? Can the authors explain/comment?

**(36)** Fig. 5: Should the sum of each row in the table be equal to one, including both tables a) and b)? Could the authors clarify the difference between the first-order and higher-order values? It would be helpful to include an explanation of this in the methods section, for example at the end of section 2.4.1.

**(37)** Fig. 6: The figure is not much referenced in the text, and because the values are normalized, it is difficult to compare them with Table 1 (I assume that the optimal values and ranges are derived from this figure in some

way) and it does not provide a finer understanding of the optimal range. Perhaps it could be moved to the supplementary materials.

**(38)** Fig. 10: As mentioned in comment 13, I find misleading the term Southern Ocean in the caption.

**(39)** Fig. 11: Can the authors add the latitude on the map.

**References**

Tarkeshwar Singh, François Counillon, Jerry Tjiputra, and Yiguo Wang. A Novel Ensemble-Based Parameter Estimation for Improving Ocean Biogeochemistry in an Earth System Model. *Journal of Advances in Modeling Earth Systems*, 17(2):e2024MS004237, 2025. ISSN 1942-2466. doi: 10.1029/2024MS004237.

Daniel B. Williamson, Adam T. Blaker, and Bablu Sinha. Tuning without over-tuning: Parametric uncertainty quantification for the NEMO ocean model. *Geoscientific Model Development*, 10(4):1789–1816, April 2017. ISSN 1991-959X. doi: 10.5194/gmd-10-1789-2017.

Judith Hauck, Cara Nissen, Peter Landschützer, Christian Rödenbeck, Seth Bushinsky, and Are Olsen. Sparse observations induce large biases in estimates of the global ocean CO2 sink: An ocean model subsampling experiment. *Philosophical Transactions of the Royal Society A: Mathematical, Physical and Engineering Sciences*, 381(2249):20220063, May 2023. doi: 10.1098/rsta.2022.0063.

Lucas Gloege, Galen A. McKinley, Peter Landschützer, Amanda R. Fay, Thomas L. Frölicher, John C. Fyfe, Tatiana Ilyina, Steve Jones, Nicole S. Lovenduski, Keith B. Rodgers, Sarah Schlunegger, and Yohei Takano. Quantifying Errors in Observationally Based Estimates of Ocean Carbon Sink Variability. *Global Biogeochemical Cycles*, 35(4):e2020GB006788, February 2021. ISSN 1944-9224. doi: 10.1029/2020GB006788.

---

## Author Comment (AC3)

**Overview**

**Research summary**

(1) In this paper, the authors introduce a new version of the marine biogeochemical model, WOMBAT-lite. The updated model incorporates several new processes and requires tuning using observational datasets. The authors leverage machine learning methods to optimize the model. First, a machine learning model is trained on outputs from 512 experiments, each with different parameter sets (same parameters but different values), to predict 8 root mean square errors (RMSEs) relative to observational products. This trained machine learning model is used to perform a sensitivity analysis, identifying the parameters that most significantly impact the RMSEs — a total of 11 parameters are identified. Then, a second machine learning model is trained using outputs from 512 new experiments, focusing solely on the 11 selected parameters, to replicate a cost function that measures the deviation from observations. This trained model is utilized for optimization analysis to determine the optimal values and ranges of the 11 parameters. The optimized version of the model is then compared to observational data and a previously unoptimized version of WOMBAT. The new optimization approach proves to be successful, with the optimized model showing improvements across all evaluated variables, albeit to varying degrees.

**Overall assessment**

(2) This technical paper is clearly written and presents a novel method for optimizing the new version of the biogeochemical model. The authors have made great efforts to describe the machine learning approach, detailing the sequential steps used first to identify key parameters and then to optimize them. I have a few major comments, that summarize as: 1) Additional justifications are needed to ensure that ten years of simulations are long enough, in relation to the model's intended usage and the influence of the optimization strategies. 2) Further discussion on alternative model tuning approaches would be beneficial. 3) The advantages of this optimization approach over simply selecting the best of the 512 experiments need to be further explained. 4) Further details on the machine learning algorithms may be necessary. I elaborate on these major comments below. Most of these comments can probably be addressed by adding text in the introduction or conclusive section of the manuscript. Additionally, I have a list of minor or specific comments and suggestions. Finally, given the length of the manuscript and the number of figures, any content that could be shortened would be beneficial. I tried to provide some suggestions. Overall, this is a quality piece of work, and I recommend it for publication after some minor revisions.

We thank the reviewer for their thorough and constructive suggestions that have substantially improved the manuscript.

**Major comments**

(3) My first concern is the ten-year-long simulations used for conducting the sensitivity/optimization analysis. I understand the constraints associated with running a large number of longer simulations. Nonetheless, I would like to have a clearer picture of the equilibrium achieved in ten years. What are the drifts at various depths? Although ten years may be enough for the surface ocean, what about the deeper ocean that can impact the surface on longer time scale? The authors have noted a drift away from the optimal fit when simulations are extended to 100 years. Could the authors discuss the risk of further drift in

these simulations? Additionally, is the drift of the optimized simulation weaker compared to the best initial simulation or any of the 512 simulations? If the drift is indeed weaker, this would be a strong point in favor of the optimization strategy.

Both reviewers are correct to point out one of the biggest caveats of our work. 10 years is sufficient time for the upper ocean biogeochemistry (higher frequency cycling) to equilibrate into different states given the input parameter set, but it is certainly not long enough for the whole ocean biogeochemistry to equilibrate. Typically, this requires thousands of years of continuous simulation.

The choice of 10 years was due to three factors: (1) that we assess only the high frequency turnover of the ecosystem and biogeochemical components of the model, ignoring the low frequency, long timescale effects such as the drift the reviewer mentions; (2) due to computational constraints we cannot run 512 simulations for 100+ years each; and (3) even if computational constraints were not an issue, we wouldn't choose 100+ years because of the confounding effect of error in the ocean's physical state that propagates error into the biogeochemical fields. Because we initialise the model with observed temperature and salinity fields, the longer we continue the simulations the more that the model physical state will drift from the observed fields.

We fully acknowledge and address the reviewers concerns here by adding additional text discussing the advantages and disadvantages of our 10-year simulations.

Lines 253:263 – *"We chose to run the experiments for only 10 years, making a total of 5120 model years and at a nominal horizontal resolution of 1º. This short timescale was sufficient to assess the skill of the biogeochemical model, at least regarding its ecosystem component. Marine phytoplankton contribute half of all primary production in the Earth system (Field, 1998) but represent less than 1% of photosynthetic biomass (Friedlingstein et al., 2023; Le Quéré et al., 2005), meaning that they turn over quickly. Changes to key parameters within the ecosystem component therefore result in a rapid realisation of different patterns in biological states (e.g., chlorophyll and net primary production, among others). Our analyses and optimisation thus focus on the ecosystem component using 10-year model runs. We do acknowledge that longer-term, low frequency modes of variation exist in biogeochemical models, and to partially address this we completed 100-year simulations with optimal parameter sets. However, we also note that longer integrations risk the compounding of physical and biogeochemical model errors as the physical state drifts further from the observations. Optimisation of the biogeochemistry under these conditions can cause over-tuning of the biogeochemical parameter set that compensate for physical errors (e.g., Singh et al., 2025)."*

Lines 559:564 – *"Continuing to run the model forward for 100 years post initialization showed some degradation in the performance (red bars in Fig. 7). This is expected, since our optimisation procedure was trained on model output only 10 years post initialization due to computational constraints. Model outcomes drift further away from the target fields with longer integrations. Lower frequency variability and trends are thus missed by the optimisation that are nonetheless present in the biogeochemical model, and these play out as the model is integrated forward for longer."*

*Lines 807:816: "Even with its optimal parameters, WOMBAT-lite suffered a loss in performance when run over 100 years compared to when run over only 10 years. Future iterations of surrogate-based optimisation would therefore benefit from extending the length of simulations done by initial set of sensitivity experiments. That said, significant savings in computation efficiency would be needed before this is possible with computationally demanding models, such as ocean biogeochemical models, but could be feasible by running the biogeochemical model offline from the ocean physics (e.g., Séférian et al., 2013). This approach would also eliminate any confounding errors caused by an evolution of the ocean's physical state since the physical state would not be allowed to evolve. Future versions of WOMBAT, including WOMBAT-lite, WOMBAT-mid and WOMBAT-full, and their deployment into different configurations (e.g., higher resolution versions) would benefit from this methodology of optimisation."*

(4) Somewhat connected with the previous point, could the authors specify/discuss the intended usage for which the model is optimized? For instance, will the model perform equally well when incorporated into an Earth System model for climate simulations? I am curious whether the optimization based on ten-year simulations and focusing on the surface ocean would remain effective during the longer processes, such as a thousands of years spin-up to achieve preindustrial equilibrium, followed by hundreds of years for simulating historical and future climate conditions.

The optimisation is immediately valid for ocean-only simulations and the parameter set defined here will be applied directly. It's application within Australia's Earth System Model, the ACCESS-ESM1.6, is however not so simple and will require additional tuning. Ideally, we would run the same surrogate machine learning optimisation procedure on 512 simulations with the ESM, but this is computationally expensive and prohibitive. We therefore accept that the optimal parameter set from these ocean-only experiments will need to be varied to deliver similar skill in the ESM.

We have added a sentence clarifying this in the introduction.

Lines 111:114 – *"This optimised version of WOMBAT-lite is intended for use within the Australian Community Climate and Earth System Simulator (ACCESS) umbrella of ocean-only model configurations (e.g., Kiss et al., 2020), while slight changes would be needed for the full earth system configurations to accommodate the differences in the physical state of the ocean (e.g., Law et al., 2017)."*

(5) Another major comments reside in emphasizing the utility of this optimization approach. Considering that the ML approach will identify the optimal parameter set in the space that has been explored, why not using the best performing of the 512 experiment? Could the authors comment a bit around that? I guess the optimization approach also provide some uncertainty and sensitivity quantification but can't we get the same from the 512 experiments? What about comparing the optimal simulations with the best of the 512 initial simulations?

This comparison is made in the current Figure 7, where we show (yellow star) that the optimal parameter set outperforms the best of the 512 simulations in terms of the global cost

function. We also provide a new supplementary figure that shows the distribution of performances between the optimal and full set of sensitivity experiments.

This figure shows that the optimal parameter sets always perform well, with the majority of optimal experiments always performing better than at least 75% of sensitivity experiments (with the exception of surface dissolved iron, which performs better than at least 50%). For air-sea flux of $CO_2$ and particle sinking flux, one optimal experiment has skill exceeding the skill of all sensitivity experiments. For other target fields, the best performing of the optimal experiments either has slightly greater skill or slightly less skill than the best of the 512 sensitivity experiments. The consistently good performance of the optimal experiments, particularly related to the skill improvements in air-sea flux of $CO_2$, shows that there is value in selecting the optimal parameter set over the best of the 512 sensitivity experiments. However, we acknowledge that good solutions do exist from within the sensitivity experiments.

Lines 557:560 – *"These optimal versions of WOMBAT-lite show good fidelity to the target fields, with all registering good performances in terms of the global cost function that were as good as or better than the best of the 512 sensitivity experiments for the majority of the target fields (yellow bars in Fig. 7; Fig. S6)."*

[Figure]

*Figure S6. Performance of sensitivity and optimal experiments. Performance measured by the cost function of all 512 sensitivity experiments (black) against all 20 optimal experiments (gold) for each of the 8 target variables. The best performing of the 512 experiments (grey star) is compared with the best performing of the optimal experiments (red star).*

(6) I would appreciate further discussion on model tuning and alternative approaches, particularly those employing ensembles/surrogate models/emulators. For instance, the work of Singh et al. (2025) and Williamson et al. (2017) seem relevant.

We have included a greater discission of these papers and their relevance to our work in the discussion.

Lines 756:764 – *"Here we applied the surrogate-based method to WOMBAT-lite, optimised the model parameters and delivered improved performance in reproducing 8 target datasets. Other approaches do exist for optimisation of biogeochemical models, such as iterative ensemble methods (Singh et al., 2025), which do not rely on surrogate methods but instead uses ensembles to iteratively adjust either the initial state or the parameters towards their optimal values through regular data assimilation (Dowd et al., 2014). While this approach can provide time-evolving parameter optimisation, the surrogate approach employed herein represents another way to optimise the parameter set of a complex biogeochemical model, albeit statically, but importantly without large computation overhead. Once the surrogate is trained, it can be deployed cheaply to explore the parameter space in different ways, offering flexibility to select a new set of parameters with perhaps an emphasis on one target field (e.g., air-sea $CO_2$ flux) over another, if required. One disadvantage, however, is that the optimisation can only be as good as the skill of the surrogate, meaning that careful training is critical."*

(7) Last major comment, while the approach is explained very well, I believe it lacks a detailed description of the machine learning techniques used. This aspect is beyond my area of expertise, so I cannot adequately assess it. However, a thorough explanation might be essential for reproducibility and for the benefit of readers interested in machine learning methods applied to a specific problem.

We point the reviewer to the code for sensitivity analysis and optimisation that is available at https://github.com/Jyoteesh38/Bayesian-Optimization-of-the-World-Ocean-Model-of-Biogeochemistry-and-Trophic-dynamics-WOMBAT-. Following this code will allow any reader with access to python to reproduce the analysis in this study and this link is available under the "Code Availability" section.

**Minor and specific comments**

**Introduction**

(8) l. 47: Reformulation suggestion for clarification: "This component represents the growth..." -> "This component, in its simplest form, represents the growth..."

Done.

(9) l. 96: There is a typo, remove "apt" in "As an apt example, ..."

Done.

**Methods**

(10) l. 163-179: I first counted 7 observation products before understanding that chlorophyll data where actually providing 2 product: depth of the maximum and surface chlorophyll. Can the authors reformulate to clarify?

Done.

Lines 190:192 - "The extension of chlorophyll to depth allows for an assessment of patterns in the vertical, including spatiotemporal variations in the position of the deep chlorophyll maximum, *which we assess in addition to surface chlorophyll concentrations.*"

(11) l. 172: Why optimizing to the CO2 flux and not pCO2? The CO2 flux is itself estimated from pCO2, thus introducing an additional source of potential errors in the observational target. Can the authors clarify/justify the choice?

Done.

Lines 194:196 - "*We opt for $CO_2$ fluxes rather than $pCO_2$ concentrations to also account for any error in the windspeed-dependent gas exchange formulation, which we do not update or optimise at this stage.*"

(12) l. 225: Can the authors gives some example and eventually some reference (if they exists) to clarify what they have in mind?

Clarified.

Lines 261:263 – "*However, we also note that longer integrations risk the compounding of physical and biogeochemical model errors as the physical state drifts further from the observations. Optimisation of the biogeochemistry under these conditions can cause over-tuning of the biogeochemical parameter set that compensate for physical errors (e.g., Singh et al., 2025).*"

(13) l. 239: I find misleading using "Southern Ocean" for all the ocean south of 20°S. Here it includes some of the subtropical gyres. I found this misleading in some other part of the manuscript. Can the authors use an other term?

Replaced "Southern Ocean" with "Southern Oceans" in all places where appropriate.

(14) l.271: For clarifying "... by comparing GPR predictions with WOMBAT-lite RMSE...", the authors can be specific and say that they compare the GPR predictions of RMSE with WOMBAT-lite RMSE.

Done.

Lines 309:311 – "*This is repeated across all folds, and predictions are aggregated. The GPR model accuracy is assessed through the goodness-of-fit ($R^2$) metric by comparing GPR predictions of RMSE with WOMBAT-lite RMSE data, which indicates high accuracy (Fig. S2).*"

(15) l. 267-268: Is there one training for the 8 RMSE or 8 training? i.e 1 GPR model predicting 8 RMSE or 8 GPR model? Can the authors clarify?

8 GPRs are trained, one for each of the 8 target fields. Clarified this.

Lines 306:307 - "Next, *GPR models are trained for each observational target* using these parameter samples as inputs and the normalized RMSE as the output."

(16) l. 276: Why not doing the optimization directly with the 24 parameters? Why having first a sensitivity analysis? (ie. why not doing step 10 just after step 5?) Can the authors clarify?

Clarified.

Lines 327:329 - *"This set of 512 sensitivity experiments are different from those generated during the sensitivity analysis because we now use a reduced set of only the most important parameters. This second iteration using a reduce parameter set and denser sampling of the parameter space is essential for the G-BO process (Reddy et al., 2024)."*

(17) l.285-297: Similar to comment 15, can the authors clarify? Is there one training for the 8 cost functions? Or one for the global cost function? Or 8 training for each cost function? To be entirely clear, the model is trained on a cost function (defined in Eq. 1 or 2), whereas for the sensitivity analysis, it focuses on the RMSE. Is that correct?

Clarified. And yes.

Lines 337:338 - *"Unlike the sensitivity analysis that measured model performance via the RMSE, this cost function penalizes poor correlations and bias and error in the variance by using an uncentered NRMSE."*

**Results**

(18) l. 349-356: Why not including PI0 in the most important parameters? On figure 5b it has a value of 0.05 for the depth of chlorophyll maximum, i.e. 5% if I understood correctly the threshold. Can the authors clarify?

Lines 415:416 - *"The initial slope of the photosynthesis-irradiance curve ($PI^0$) was not included because it fell just below the 5% threshold (shown rounded up in Figure 5b) in its higher order effects."*

(19) l. 405: The depth of the chlorophyll maximum is indeed heavily influenced by αa and γ0p but only "in interaction" (i.e. panel b of Figure 5). It is the only target (with maybe iron but it is not as marked) that have 2 parameters standing out so much in interaction while no parameters seems to strongly influence it at first order. Can the authors comment one that?

Lines 473:475 - *"Increasing $\alpha_a$ while simultaneously decreasing $\gamma_p^0$, for example, would cause an accelerated deepening of the nitracline since phytoplankton consume more nutrients but can also survive for longer under light limitation at depth."*

(20) l. 412: It reads a bit strange to mention 2 additional parameters and not have them specified right away. There are mentioned just a bit after, but maybe the authors can slightly reformulate the paragraph so that 2 additional parameters are specified right after being mentioned.

Structure improved.

Lines 478:491 - *"Our optimisation procedure involved another 512 experiments that varied 13 parameters: the 11 parameters identified in the sensitivity analysis plus 2 additional parameters that were missed by our sensitivity analysis. The additional 2 parameters included wider variations in the biomass threshold of phytoplankton for allometric scaling ($B_p^{thresh}$) from 0.01 to 1.0 (previously this had been varied from 0.01 to 0.1), as well as variations in the base scaler of temperature-dependent autotrophy ($\beta_a$), which was previously held constant during the sensitivity experiments at a value of 1.066 ($Q_{10}$ = 1.89), following* Eppley (1972), *but have nonetheless been shown to vary between phytoplankton types (Anderson et al., 2021). We took the opportunity during the optimisation to vary these parameters. In our optimisation step, we explored a range of 1.040 to 1.080 ($Q_{10}$ from 1.48 to 2.16) in $\beta_a$ motivated by the results of* Anderson et al. (2021). *All remaining, insensitive parameters were held at their default values (Table 1). Thus, we explored variations in 13 parameters (11 parameters from the sensitivity analysis and 2 additional parameters) and sought their optimal values for best reproducing the 8 target fields (Fig. 3) by minimizing the global cost function (Eq. 2). The 512 experiments were used to calibrate the global cost function synthetically using the machine learning model."*

(21) l. 413-418: Any reason why these parameters where not included in the sensitivity analysis? Can the authors clarify?

This was an oversight. We hope that the rephrasing in the previous response (#20) is sufficient to clarify this.

(22) l. 437: "... due to its complexation with organics..." is not clear to me. What is organics? Organic form of dFe? Can the authors clarify?

This is a detail of the cycling of dissolved iron in the ocean (Shaked et al., 2025; Tagliabue et al., 2017). As it is a very large topic, we refer the interested reader to these references.

(23) l. 506: In the subtropical or subpolar gyres?

Clarified.

(24) l. 504-506: It seems to me there is a very shallow chlorophyll maximum in the subtropical gyres that is not present in the observation (Fig. 8k, l). Any explanation? This is a bit contradictory with the "too deep maxima in the gyres." Can the authors clarify?

Lines 599:601 - *"We note that in places where the chlorophyll maxima appear very shallow in the subtropical gyres is where chlorophyll concentrations are so low as to not form any appreciable maximum at depth, essentially where the nitracline is so deep as to be placed in darkness."*

(25) l. 486-506: What about the CO2 flux? Can the authors mention that it receive a detailed analysis further in the paper.

Done.

Lines 595:596 – *"In the annual averages presented in Figure 8, there is little obvious change in air-sea $CO_2$ fluxes, but this hides compensating improvements in the seasonality which we detail in section 3.4.4."*

(26) l. 483-484: "Our optimised WOMBAT-lite manages to show some improvement..." I don't find Figure 10b to be the most effective way to illustrate this. Instead, Figure 11 demonstrates the improvement in the seasonality of the CO2 flux through better local correlation, whereas the correlation on Figure 10b actually decreases. I agree nonetheless that on figure 10 the amplitude seems to be a bit closer. It also seems to me that comparing a transition zone based on average is rather complex. Could the authors reformulate to provide a more precise explanation/description?

This is why we include both figures. It is also essential to show where the changes to parameter values may have altered things for the worse locally while improving them globally. We respectfully wish to keep Figure 10 as it is. We also note that we are forthcoming that there is much room for improvement regarding the model's representation of $CO_2$ fluxes.

(27) l. 593-599: This paragraph seems to be a repetition of the the one just before. Can the authors merge it and combine figure 10 and 11 (maybe not all panel of figure 10)?

Done. Combined the paragraphs, however we respectfully opt to maintain Figures 10 and 11 as separate.

Lines 674:703 – *"We directly compared the monthly $CO_2$ fluxes between an observation product (Chau et al., 2022) and WOMBAT-lite as well as the unoptimised model from January 1985 to December 2018. With optimal parameters, WOMBAT-lite shows improvement in its seasonality and regional agreement of $CO_2$ fluxes compared with an unoptimised version of the model (Fig. 10; Fig. 11). While $CO_2$ fluxes are strongly controlled by thermal processes in the subtropics and are thus well approximated by optimised and unoptimised versions alike (Fig. 10a; Fig. 11), $CO_2$ fluxes at higher latitudes are, however, more affected by biological drawdown and release (Mongwe et al., 2018; Takahashi et al., 2002). In the transition from subtropical to subantarctic zones (35ºS-50ºS) the observations show overall oceanic uptake of $CO_2$, but importantly a greater uptake in the summer (Fig. 10b). Optimised WOMBAT-lite manages to show some improvement over the unoptimised model, with lower uptake in the winter and a trend towards uptake in the spring/summer. Nonetheless, this improvement is marginal in this zone and suggests that further improvement can be made in the future. The best match between WOMBAT-lite and the data is achieved in the Antarctic Circumpolar Current zone (50ºS-65ºS), where WOMBAT-lite shows good climatological correlations with the observations, while the unoptimised model shows negative correlations (Fig. 10c; Fig. 11). The flip from poor to good performance is caused by the net outgassing in the late winter and a trend towards oceanic uptake in the spring summer (Fig. 10c). Better seasonal correlations (0.87 → 0.96) are also*

*achieved in the Antarctic Zone (65ºS-80ºS), although with WOMBAT-lite potentially overestimating the summer flux of $CO_2$ into these waters (Fig. 10d).*

*Improvement in air-sea $CO_2$ fluxes is noteworthy from a zonally integrated perspective that incorporates the Northern Hemisphere (Fig. 12). North of 40ºN, the oceanic uptake of $CO_2$ in the unoptimised version of the model exceeded the observations and this is somewhat reduced in WOMBAT-lite due to a substantial reduction in winter ingassing (Fig. 12b), while summer uptake is increased (Fig. 12c), resulting in a better match to observed $CO_2$ fluxes in the Northern Hemisphere (Fig. 12a) that is also visible in the temporal correlations (Fig. 11). Meanwhile, there is little difference between the optimised and unoptimised versions of the model in the low latitudes, emphasising how thermal changes dominate air-sea $CO_2$ fluxes in this region (Takahashi et al., 2002). Once again, in the Southern Ocean, we see clear improvements from a zonally integrated perspective. Winter outgassing is now achieved in WOMBAT-lite, although the zone of peak outgassing occurs too far north (Fig. 12c). Similarly, summer oceanic uptake is now achieved and is a closer match to the observations, although again the zones of maximum uptake are shifted too far north by roughly ~5º (Fig. 12b). Overall, the changes in the biogeochemical functionality of WOMBAT-lite show some improvements in reproducing observed air-sea $CO_2$ fluxes, although we note some degree of caution is required given the uncertainty in the observational-based product itself (Gloege et al., 2021; Hauck et al., 2023a)."*

(28) l. 593: The improvement are clear for the equatorial band (10S to 10N) and the high latitude (>50N and <30S). For the rest (10S-30S and 10N-50N) the unoptimized model is already well correlated with the observations and even better (darker red) in a few places. Can the authors reformulate to provide a more precise explanation/description?

*Lines 677:679 - "While $CO_2$ fluxes are strongly controlled by thermal processes in the subtropics and are thus well approximated by optimised and unoptimised versions alike (Fig. 10a; Fig. 11), $CO_2$ fluxes at higher latitudes are, however, more affected by biological drawdown and release (Mongwe et al., 2018; Takahashi et al., 2002)."*

(29) l. 593-599: Can the authors mentioned that the observations also have some caveats (Hauck et al., 2023; Gloege et al., 2021)?

*Done.*

*Lines 700:703 - "Overall, the changes in the biogeochemical functionality of WOMBAT-lite show some improvements in reproducing observed air-sea $CO_2$ fluxes, although we note some degree of caution is required given the uncertainty in the observational-based product itself (Gloege et al., 2021; Hauck et al., 2023)."*

(30) l. 602: "the unoptimised version of the model exceeded the observations." The optimized version also exceed the observation on figure 12a. Can the authors clarify their point?

*Clarified.*

*Lines 692:695 - "North of 40ºN, the oceanic uptake of $CO_2$ in the unoptimised version of the model exceeded the observations and this is somewhat reduced in WOMBAT-lite due to a substantial reduction in winter ingassing (Fig. 12b), while summer uptake is increased (Fig. 12c), resulting in a better match to observed $CO_2$ fluxes in the Northern Hemisphere (Fig. 12a) that is also visible in the temporal correlations (Fig. 11)."*

(31) l.601-611: On figure 12 we see that the optimized version has a much better seasonal cycle in the high latitude (in line with what we see on figure 11). This important improvement does not appear so much on the annual average because of compensation between the errors in summer and winter. This should be stated more clearly.

Stated earlier as per your request and wording.

**Appendix**

(32) l. 1430: There is typo, "is remineralised" is repeated.

Corrected.

**Figures**

(33) Fig. 1: In the text of step (6) in the figure, consider to keep the same format as the other steps. Maybe something like "Run a global sensitivity..."

Done.

(34) Fig. 4: Can the authors add the performance of the optimized model and the 20 optimal experiments after 10 and 100 years.

Yes this can be added, but we opt for these to be shown as supplementary figures to not distract or clutter an already lengthy manuscript. This has been added as Supplementary Figure 6.

(35) Fig. 4: Why is the standard deviation of sinking detritus, NPP and chlorophyll so weak for many experiments? Can the authors explain/comment?

Likely that our NPP and sinking fluxes are too high in the gyres and too low in the productive regions. Or also that the measurements in the case of sinking particulate fluxes are highly localised in space and time, while the model necessarily produces smoothed (i.e., temporally and spatially averaged) output.

(36) Fig. 5: Should the sum of each row in the table be equal to one, including both tables a) and b)? Could the authors clarify the difference between the first-order and higher-order values? It would be helpful to include an explanation of this in the methods section, for example at the end of section 2.4.1.

The total order (the sum of both tables) does not necessarily sum to 1. The first-order indices sum to **at most** 1, while the total-order indices sum to **at least** 1. If there are no interactions, then the first and total-order indices are equal, and both the first and total-order indices sum to 1. We have included this information in the Figure legend.

(37) Fig. 6: The figure is not much referenced in the text, and because the values are normalized, it is difficult to compare them with Table 1 (I assume that the optimal values and ranges are derived from this figure in some way) and it does not provide a finer understanding of the optimal range. Perhaps it could be moved to the supplementary materials.

It is possible to move this to the supplement, but it is also a visualisation of a key outcome of the work. We defer to the editor, but have also added another reference to Figure 6 in the text as a way to show its importance.

(38) Fig. 10: As mentioned in comment 13, I find misleading the term Southern Ocean in the caption.

Corrected to "Southern Oceans".

(39) Fig. 11: Can the authors add the latitude on the map.

Yes.

**References**

Tarkeshwar Singh, François Counillon, Jerry Tjiputra, and Yiguo Wang. A Novel Ensemble-Based Parameter Estimation for Improving Ocean Biogeochemistry in an Earth System Model. *Journal of Advances in Modeling Earth Systems*, 17(2):e2024MS004237, 2025. ISSN 1942-2466. doi: 10.1029/2024MS004237.

Daniel B. Williamson, Adam T. Blaker, and Bablu Sinha. Tuning without over-tuning: Parametric uncertainty quantification for the NEMO ocean model. *Geoscientific Model Development*, 10(4):1789–1816, April 2017. ISSN 1991-959X. doi: 10.5194/gmd-10-1789-2017.

Judith Hauck, Cara Nissen, Peter Landschützer, Christian Rödenbeck, Seth Bushinsky, and Are Olsen. Sparse observations induce large biases in estimates of the global ocean CO2 sink: An ocean model subsampling experiment. *Philosophical Transactions of the Royal Society A: Mathematical, Physical and Engineering Sciences*, 381(2249):20220063, May 2023. doi: 10.1098/rsta.2022.0063.

Lucas Gloege, Galen A. McKinley, Peter Landschützer, Amanda R. Fay, Thomas L. Frölicher, John C. Fyfe, Tatiana Ilyina, Steve Jones, Nicole S. Lovenduski, Keith B. Rodgers, Sarah Schlunegger, and Yohei Takano. Quantifying Errors in Observationally Based Estimates of Ocean Carbon Sink Variability. *Global Biogeochemical Cycles*, 35(4):e2020GB006788, February 2021. ISSN 1944-9224. doi: 10.1029/2020GB006788.

---

## Author Comment (AC4)

The authors present the use of a machine learning-based approach to emulate the output, or more specifically, the model-data misfit as quantified by a cost function of a NPZD-style global ocean biogeochemical model to estimate model parameters. The approach is interesting, appears to work well in a relatively high-dimensional parameter space; however, some of the implementation details are not described well, and a key model shortcoming is not emphasized enough.

We thank the reviewer for their thoughtful suggestions and have attended to them in our answers below, and we feel that these comments have improved the manuscript.

**general comments**

One aspect of the study that may and is not emphasized enough in the current version of the manuscript is the use of a single phytoplankton and a single zooplankton tracer/variable in a global ocean model. Phytoplankton growth rates and other key parameters vary greatly between different phytoplankton groups, for example those that dominate coastal ecosystems in comparison to phytoplankton more commonly found in the open ocean. Thus, in some way, this study aims to do the impossible: fit the parameters of a one phytoplankton variable to a global dataset which is based on a complex spatially and temporally varying phytoplankton population. Here, I am not arguing that such as study should not be conducted -- the use of a machine learning surrogate model is very interesting -- but many of the shortcomings of the model identified in the study may be due to the very simple plankton representation in the model. For example, the manuscript describes underestimates of net primary production and issues in bloom phenology, and provides possible reasons for these model shortcomings, but the use of a single phytoplankton variable is not among them. Yet, a single phytoplankton model is unlikely to provide good primary production and bloom phenology estimates globally.

We agree wholeheartedly that a single phytoplankton and a single zooplankton functional group is not sufficient to represent the global marine ecosystem, particularly the transition from coastal to open ocean biomes. The reviewer is correct to assert that the model is simple, and that this simplicity limits its potential to represent the processes of primary production, remineralisation, carbon export and air-sea gas exchange, among others.

However, we note that extending the model to many functional types of phytoplankton and zooplankton is beyond the scope of this work. The biogeochemical model itself is called WOMBAT-lite precisely because of its computational efficiency, which relies on having less tracers and process complexity. That said, we would also hasten to point the reviewer to the

extensive developments that were made during this work to the model architecture (Appendix A), which form a considerable jump in model complexity compared to the previous WOMBAT model, while still maintaining its "lite" tracer number.

To acknowledge and address the reviewers concern, we have:

- Acknowledged in the introduction that the ocean-biogeochemical model is simple compared to others, but nonetheless is complex enough to demand optimization with surrogate techniques, and this optimization is required to realize the full potential of the model.

  Lines 93:99 - *In this study, we optimise a relatively simple ocean-biogeochemical model designed to represent open-ocean biomes using surrogate machine learning techniques: version "lite" of the World Ocean Model of Biogeochemistry And Trophic dynamics (WOMBAT-lite) (Fig. 1). This surrogate approach is crucial. Although WOMBAT-lite has few tracers and is computationally efficient, making it viable for high resolution configurations (Kiss et al., 2020; Matear et al., 2015; Menviel and Spence, 2024; Oke et al., 2013) and large ensembles (Mackallah et al., 2022; Rashid, 2022; Ziehn et al., 2020), it is nonetheless a global, three-dimensional, biogeochemical model with complex non-linear process interactions. This makes it computationally demanding enough to prevent parameter calibration via traditional techniques.*

- Acknowledged in section 3.4.3 (Phytoplankton bloom phenology) that

  Lines 635:637 - *"Alternatively, it is possible that representing the global marine phytoplankton community with only one functional type limits the model's potential to realize the full variation."*

- Acknowledged in the Summary that

  Lines 797:803 - *"Alternatively, spatial variations in $\varepsilon$ that capture transitions from nano- to meso-zooplankton from oligotrophic to eutrophic regimes (Rohr et al., 2024) may serve to accelerate the phytoplankton bloom at the beginning of the growth season. Furthermore, the succession of different types of phytoplankton is important for the biological carbon pump (Tréguer et al., 2018). Therefore, representing these shifts in community with additional functional types of plankton beyond that explored herein might be important for the phenology of the annual spring bloom, and by extension Southern Ocean $CO_2$ fluxes."*

One more example: the authors highlight the "difficulty in reproducing the seasonality of air-sea CO2 exchange in the Southern Ocean" (l. 654) as a chief issue of the model. There are

many studies that emphasize the unique phytoplankton composition in the Southern Ocean and the role of Southern Ocean diatoms in modulating carbon export, for example "Influence of diatom diversity on the ocean biological carbon pump" (Tréguer et al., 2018, DOI: 10.1038/s41561-017-0028-x). Thus, it does not seem surprising that a single phytoplankton model, optimized on a global dataset, does not perform particularly well in the Southern Ocean.

In the revised manuscript, we have acknowledged and addressed that the model has limits in its potential to more completely represent the succession of different forms of phytoplankton due to only having one functional type, and that this may feed directly or indirectly into error in other features of the model, such as air-sea exchange of $CO_2$. (See above).

A related aspect to the comments above is that model performance can easily be biased based on the observation locations and the amount of data from each location. That is, if more open-ocean than nearshore observations are included in the RMSE-based cost function used for parameter estimation, then the phytoplankton variable will be parameterized more like a general open-ocean species (likely adapted to low-nutrient conditions) whereas the parameter estimates may be quite different if mostly nearshore observations are used. In summary, I suggest emphasizing the drawbacks of simple biogeochemical models more. Yes, parameter estimation becomes easier with fewer plankton variables, but even optimized parameters cannot capture the complex plankton ecosystem on a global scale for simple NPZD-style models.

We agree with the reviewer that the simplicity of the model will necessitate some acceptance of model-data misfit. We accept that this is the case, and we chose to optimize towards observational target fields with a bias towards open-ocean environments. Thus, our global ocean biogeochemical model is optimised towards representing open-ocean biomes.

We acknowledge this directly in the introduction:

Lines 93:95 – *"In this study, we optimise a relatively simple ocean-biogeochemical model designed to represent open-ocean biomes using surrogate machine learning techniques: version "lite" of the World Ocean Model of Biogeochemistry And Trophic dynamics (WOMBAT-lite) (Fig. 1)."*

Building suitable and balanced cost functions for parameter estimation experiments is not easy and the authors include 8 data products in their cost function (Eq. 2), making it quite complex and interesting. Here modelers and readers like me might be interested in a bit more detail that can be obtained without additional model experiments: One question if

100 interest is how "orthogonal" the different cost function parts are, i.e. which cost function parts/data products provide additional constraints on the parameters. Calculating the linear correlation of the cost functions parts for the 512 simulations or visualizing them using a "scatter plot matrix" could be of interest and would show which cost function parts can be optimized together and which move the parameter estimates in different directions.

105 This is a great idea. We show in the figure below the linear correlations between all cost functions for the 512 sensitivity experiments. Some are poorly correlated, and others are relatively well correlated. Among those most well correlated ($r \geq 0.7$) are:

- Air-sea flux of $CO_2$ (20ºS-90ºS) – sinking particle flux
- Air-sea flux of $CO_2$ (20ºS-90ºS) – surface chlorophyll concentration
110 - Sinking particle flux – surface chlorophyll concentration
- Air-sea flux of $CO_2$ (20ºS-90ºS) – surface $NO_3$ concentration (20ºS-20ºN)
- Sinking particle flux – surface $NO_3$ concentration (20ºS-20ºN)

This suggests that these target fields are not providing completely orthogonal constraints on the parameter set, and that optimising for one is optimising for another. Mechanistically, it
115 suggests that how we represent the biological pump has important effects on the air-sea flux of $CO_2$.

All other associations between target fields are moderately to poorly correlated, meaning that they are more orthogonal and that their optimisation can proceed without always affecting the other fields.

120 We can include this figure as a supplementary figure 5 for the manuscript and reference it here:

Lines 344:346 - *"Although expected, we note that the predicted cost functions using the GPR models are not completely orthogonal, with some being well correlated (Fig. S5), indicating that the optimization of parameters towards one target field will affect other target fields."*

[Figure]

*Figure S5. Pair-wise correlations between cost functions of the 8 target fields.* *If positive and significant, the gains in the skill of the model to reproduce one target field will positively affect the models skill in reproducing the other.*

Relatedly, it would be good to know if the optimal parameter values that were obtained perform well (have low values) for all cost function parts or if they perhaps do not fit certain datasets very well at all.

The optimal sample sets (sampling of the posterior distributions in Figure 6) performed well, as shown in Figure 7 in terms of the global cost function. However, what the reviewer is asking for is how the optimal parameter sets behaved in terms of the individual cost functions for all 8 target fields. This is presented below in the box and whisker plot.

Some fields are optimised more effectively than others. The air-sea flux of CO2, surface chlorophyll concentrations, sinking particle flux and surface NO3 concentrations are well reproduced. Surface dissolved iron and the primary limiting nutrient for phytoplankton growth are not as well reproduced.

[Figure]

*The performance of each of the 20 "optimal" parameter sets sampled from the posterior distributions of Figure 6 in the main text against all 8 of the target fields.*

Although not made explicit, the simulations in this study appear to rely on a 9-year spin up (output from year 10 was used for model data-comparison). The authors

state that "Continuing to run the model forward for 100 years post initialization showed some degradation in the performance" (l. 469). Parameter estimation relies on not having too much drift in the model and for the parameter values to have taken effect (there is no longer drift introduced by the change in parameter values). How did the authors ensure that the 9-year spin up was sufficient?

We explicitly discuss the choice of 10-year simulations in the final summary paragraph, acknowledging that this is a key caveat of our work. The degradation in the performance of the model that occurs by running the model forward for 100 years, rather than just 10 years, is an unfortunate symptom of only running our 512 training experiments for only 10 years.

We stress that this choice was unavoidable because of computational demands of running the ocean model. Moreover, because the ocean model was run online with the physics, our choice of using only 10 years was also informed by a desire to optimise the biogeochemical ecosystem component, which overturns quickly, while limiting physical biases, which grow as the simulation progresses. Optimising the biogeochemical component of an ocean model should be done in a physical setting that is as close to the observations as possible. If we optimised the model using simulations that extended for centuries or even millennia, then it is likely that we would be correcting for mismatches that are generated due to physical error, rather than biogeochemical error.

We fully acknowledge and discuss this caveat at several points in the manuscript and suggest future fixes:

Lines 253:263 – *"We chose to run the experiments for only 10 years, making a total of 5120 model years and at a nominal horizontal resolution of 1°. This short timescale was sufficient to assess the skill of the biogeochemical model, at least regarding its ecosystem component. Marine phytoplankton contribute half of all primary production in the Earth system (Field, 1998) but represent less than 1% of photosynthetic biomass (Friedlingstein et al., 2023; Le Quéré et al., 2005), meaning that they turn over quickly. Changes to key parameters within the ecosystem component therefore result in a rapid realisation of different patterns in biological states (e.g., chlorophyll and net primary production, among others). Our analyses and optimisation thus focus on the ecosystem component using 10-year model runs. We do acknowledge that longer-term, low frequency modes of variation exist in biogeochemical models, and to partially address this we completed 100-year simulations with optimal parameter sets. However, we also note that longer integrations risk the compounding of physical and biogeochemical model errors as the physical state drifts further from the observations. Optimisation of the biogeochemistry under these conditions can cause over-tuning of the biogeochemical parameter set that compensate for physical errors (e.g., Singh et al., 2025)."*

Lines 559:564 – *"Continuing to run the model forward for 100 years post initialization showed some degradation in the performance (red bars in Fig. 7). This is expected,*

*since our optimisation procedure was trained on model output only 10 years post initialization due to computational constraints. Model outcomes drift further away from the target fields with longer integrations. Lower frequency variability and trends are thus missed by the optimisation that are nonetheless present in the biogeochemical model, and these play out as the model is integrated forward for longer.”*
*Lines 807:816: “Even with its optimal parameters, WOMBAT-lite suffered a loss in performance when run over 100 years compared to when run over only 10 years. Future iterations of surrogate-based optimisation would therefore benefit from extending the length of simulations done by initial set of sensitivity experiments. That said, significant savings in computation efficiency would be needed before this is possible with computationally demanding models, such as ocean biogeochemical models, but could be feasible by running the biogeochemical model offline from the ocean physics (e.g., Séférian et al., 2013). This approach would also eliminate any confounding errors caused by an evolution of the ocean's physical state since the physical state would not be allowed to evolve. Future versions of WOMBAT, including WOMBAT-lite, WOMBAT-mid and WOMBAT-full, and their deployment into different configurations (e.g., higher resolution versions) would benefit from this methodology of optimisation.”*

One of the metrics used in the cost function is primary limiting nutrient data, but there is not much information about its use and how model-data misfit is quantified. The model seems to include carbon and iron, all other elemental quotas/ratios are considered to be fixed. Fig. 3g shows nitrogen, iron and phosphorus as limiting nutrients in the data, how are these compared to model output? How is the RMSE computed? Some more information is needed.

We have included a greater discussion of how this comparison is made between the observations and the model.

Lines 204-209: *“For the primary limiting nutrient dataset, we only consider nitrogen and iron as limiting (excluding phosphorus) and ascribe nitrogen limitation, iron-nitrogen co-limitation and iron limitation as equal to 1.0, 1.5 and 2.0, respectively. This is compared directly to the degree of limitation by the model, which varies continuously from strong nitrogen limitation to strong iron limitation (1.0 to 2.0). Although the model does not include co-limitation as a process, simulated values between 1.0 and 2.0 can represent seasonal variations between nitrogen and iron limitation over an annual timescale.”*

Several aspects of study design and implementation details are not described well and can lead to confusion. For example, the generation of 512 parameter samples is mentioned in several places independently: "We undertook 512 simulations that each sampled randomly from predefined ranges of 24 key parameters related to the ecosystem component of the model" (l. 189). Then, later: "Initially, a Quasi-Monte Carlo Sobol sequence is applied to generate 512 parameter samples using the Uncertainty Quantification Python Laboratory package" (l. 263). Here, it is not clear if the 512 parameter samples are the same ones mentioned before; these methods sections need to be connected better. Furthermore, the text mentions that

512 parameter samples are drawn initially and then a sample size of 512 was selected based on the sample size sensitivity experiments (l. 267). Was it sheer luck that the initial number of experiments was also the right number of experiments confirmed in later tests?

We have added extra text to explain why 512 experiments were done for both steps: the sensitivity analysis and the optimisation. This information is already in the supplementary material, but we have been more explicit in pointing the reader towards it and have also added more sentences to explain why two sets of 512 experiments were done.

Lines 220:222 – "*A total of 512 experiments was selected because it was enough for training the surrogate machine learning model to an acceptable standard (Fig. S1).*"

Lines 301:302 – "*The analysis focuses on the target fields detailed in Fig. 3 and section 2.3.1. To generate the 512 sensitivity experiments required to train the surrogate machine learning model,*"

Lines 322:332 – "*Following sensitivity analysis, which identified the most important parameters for model performance (RMSE) of the eight target fields, we performed parameter optimisation (Fig. 1). This study uses Gaussian Process Regression-based Bayesian Optimisation (G-BO) (Reddy et al., 2024a) to identify the optimal parameter distributions so that WOMBAT-lite can best reproduce the eight target fields simultaneously. The process begins by generating another 512 parameter samples via the same Quasi Monte-Carlo (QMC) Sobol sequence design implemented through the Uncertainty Quantification Python Laboratory (UQ-PyL) package (Wang et al., 2020). This set of 512 sensitivity experiments are different from those generated during the sensitivity analysis because we now use a reduced set of only the most important parameters, which also provides a denser sampling of the parameter space essential for the G-BO process (Reddy et al., 2024a). These 512 sample parameter sets are used as input to WOMBAT-lite, and the model is run forward for 10 years (see above). A sample size of 512 is selected based on the sample size sensitivity experiments (Fig. S3).*"

**specific comments**

L 12: "Optimisation of the model parameters is crucial to ensure model performance based on process representation, rather than poor parameter values.": This sentence is not very clear, please rephrase.

Rephrased for clarity:

Lines 12-13 - "*Optimisation of the model parameters is crucial to ensure that model performance is based on process representation (i.e., functional forms), rather than poor choices of input parameter values.*"

L 19: Isn't the size of the training dataset (512) somewhat in conflict with the previous claim that "(tens of) thousands of simulations are required to accurately

estimate optimal parameter values" (l 14). Perhaps it would be useful to add some qualifiers that this large number of simulations would be required by naive approaches like grid search and for a certain number of estimated parameters.

We have added a clarifying clause to this sentence:
Lines 18:20 - "A computationally inexpensive surrogate machine learning model based on Gaussian Process Regression was trained on a set of 512 simulations with WOMBAT-lite *and was used to produce synthetic results emulating tens of thousands of simulations*."

L 22: What are "optimal posterior distributions"? From a Bayesian technique, one would expect to obtain (samples from) the posterior distribution, or perhaps maximum a posteriori (MAP) point estimates. But there is no "optimal" posterior distribution.

Replaced "optimal" with "constrained"

L 66: "However, even if our understanding and observational network were complete, there exist many tuneable and potentially inter-dependent parameters that control many target outcomes...": True, but with complete understanding, we would know the relevant rates that these parameters aim to represent. We would not even need models if our understanding and observational network were complete. I would suggest rephrasing the issue and avoiding the fictional scenario of "complete" knowledge.

Rephrased.

Lines 69:71 – *"However, even with greater understanding and observations, there exist many tuneable and potentially inter-dependent parameters that control many target outcomes (air-sea $CO_2$ fluxes, nutrient fields, chlorophyll concentrations, etc.) that must be reproduced simultaneously."*

L 74: Does the "a priori ranges" imply that the prior distributions are uniform? As a first-time reader familiar with Bayesian techniques, I would have expected the term "prior estimates" here or a brief explanation of the "a priori ranges".

The prior distributions are indeed uniform over the prescribed range. We have added "uniform" as an adjective.

L 118: "We have developed a new ocean biogeochemical model called WOMBAT-lite": This statement is a bit confusing to the reader, as a previous sentence appears to suggest that WOMBAT-lite has been applied in previous studies: "Although WOMBAT-lite has few tracers and is computationally efficient, making it viable for high-resolution configurations (Kiss et al., 2020; Matear et al., 2015; Menviel and Spence, 2024; Oke et al., 2013)" (l 90). I would suggest clarifying.

Changed to "updated".

L 118: Fig. 2 is referenced before Fig. 1.

Figure 1 is referenced in the Introduction.

L 119: It would be useful to many readers to add a brief description of Sobol sensitivity analyses or at least a reference.

Reference added.

L 145: How is cell size included in the model using just one phytoplankton tracer?

See the Appendix for this explanation. Variable cell size is emulated by considering a relationship between the density of phytoplankton and the average cell size. This affects half-saturation coefficients for nutrient uptake, the packaging effect on light transmission and also on the sinking rates of detritus.

L 174: The connection between CbPM, VGPM and the data used for model evaluation needs a better explanation.

Added a sentence to explain why we use CbPM-based NPP rather than chlorophyll-based NPP.

Lines 200:202 – *"Net primary production (NPP) built from the CbPM products should therefore provide more independent constraints on model assessment than an NPP product built from chlorophyll concentrations."*

L 189: Where samples obtained "randomly" or via a more sophisticated sampling strategy like Latin hypercube sampling? How were the parameter ranges selected?

Additional information given:
Lines 218:220 - "We undertook 512 simulations that each sampled randomly from predefined ranges of 24 key parameters related to the ecosystem component of the model (Table 1; Fig. 1) *using a Quasi-Monte Carlo Sobol sequence (see section 2.4).*"

L 288: How does the max-min scaling work? Are max and min values taken from the 512 samples, are outliers considered in the normalization (for example those in the depth of the DCM, Fig. S2 d)?

Clarified.

Lines 334:335 - "is the normalized root mean square error (scaled by the max-min*, such that the worst experiment has an NRMSE of one and the best is zero*)"

Eq. 2: I would recommend placing the superscript into parentheses "(n)" to avoid confusing it for an exponent.

Done.

L 301: The "p(z)" here is not very helpful without providing additional context.

Removed.

Fig. 4: It is unclear how the primary limiting nutrient is turned into a number and how the misfit is quantified.

This information has been added earlier in the manuscript within section 2.3.1 Observational target fields for assessment.

Fig. 5: What kind, if any, normalization was applied to the RMSE values here? Not using any would make the values dependent on the scale and units of the properties/target field shown. Using max-min (previously denoted as NRMSE) would make it dependent on the variability that can be introduced by any changes in the parameter values -- which might vary greatly between properties.

The RMSE values are presented here as normalised by the max-min scaling. We have added an extra note of this in the figure legend.

Lines 441:445 – *"Performance is measured by the root mean square error (RMSE), which is here normalised by the full range (max-min scaling). First-order indices are direct individual effect of a parameter on the target field, while higher-order indices indicate that interaction effects with other parameters are important. First-order indices sum to at most one for a given target field, while first-order + higher-order indices sum to at least one. If there are no interaction effects, then higher-order effects are nil."*

L 444: The units here would raise fewer eyebrows by changing the "m^6 (mmol C)^-2" to "(mmol C m^-3)^-2".

Agreed!

L 450: "The fact that our optimisation always chose the lowest values (near 0.01 day-1) suggests that the proportion of the community that is stressed is considerably lower than we assumed.": I would suggest being careful in generalizing from the "optimal" value of a single global phytoplankton tracer to properties of the full plankton community.
Agreed. Rephrased.

Lines 535:537 - *"The fact that our optimisation always chose the lowest values (near 0.01 day$^{-1}$) perhaps suggests that, at least with our simple model, the proportion of the community that is stressed is lower than initially assumed."*

---

## Referee Report (RR1)

**Reviewer's comments on Buchanan et al. "*Optimisation of the World Ocean Model of Biogeochemistry and Trophic-dynamics (WOMBAT) using surrogate machine learning methods*"**

*June 19, 2025*
* * *
**Overview**

Thanks to the authors for their thorough response to my comments. The additional text in the revised manuscript addresses my concerns. In particular I find the additional figure S6 complementing very well the figure 7 to show the interest of using the surrogate model to run an extensive optimisation. One remaining question to me: how the optimisation affect the drift of the model? If it could be shown that the optimisation reduce the drift that would be an additional strength of the method. Maybe by comparing the drifts in the best of the sensitivity experiments and the best of the optimal experiments, both run on 100 years? In addition, I have a couple of minor and specific comments, some of which I did not catch during the previous review. Once these minor adjustments are made, the manuscript will be ready for publication.
* * *
**Minor and specific comments**

**Abstract**

**(1)** l. 27: I think "earth" in "...of earth system models..." is more commonly written with a capital E ("...of Earth system models...")

**Methods**

**(2)** l. 318-333: Is it one surrogate model trained for each cost function J (so 8 surrogate models in total) or one for the global cost function (equation 2)? Can you clarify? I had a similar question for the sensitivity analysis during the first round of review. The authors clarified very well, notably by using plural forms or emphasizing singular terms when appropriate.

**Results**

**(3)** l. 463-464: "The 512 experiments were used to calibrate the global cost function synthetically using the machine learning model." I do not think this sentence really state what was done. What I understood is that: the 512 experiment are used to trained the machine learning model to reproduce the global cost function. The trained model is then used to look for the optimal values of the parameters, i.e. the ones minimizing the global cost function. It do not think we can say that the global cost function is calibrated. Can the authors rephrase?

**(4)** l. 478-479: "We also note that the model predicted optimal values that often aligned well with ecological theory." Is the initial range of the parameters' values much wider than the theory? Otherwise it seems expected

that the optimal values are aligned with the ecological theory and thus the argument is not very strong and could be removed.

**(5)** l. 531: "unoptimised biogeochemical model". I assume the authors mean the former version of the model that even if not optimised was considered good enough to conduct analysis and numerical experiment. Correct? This model was still tuned. Can the author specify this? Otherwise, it sounds like the new optimised version is compared to a version that should even not be used for any study.

**(6)** Sec. 3.4.3: Is the bloom phenology better than with the unoptimised model version?

**Summary**

**(7)** l. 666-668: The authors should mention here that the surrogate model can provide the large number of samples at a low computational cost.

**(8)** l. 677: Shouldn't it be "statistically" instead of "statically"?

**Figures**

**(9)** Fig. 5: I do not think emphasizing "at most" and "at least" with bold font is necessary.

---

## Referee Report (RR2)

**General comments**

The paper by Buchanan et al., provides an innovative and elegant case study for parameter sensitivity analysis and optimization of a ocean biogeochemical model using surrogate models and Bayesian methods. In the study the authors use Gaussian Processes (GPs) to predict model error (specifically, root mean square error - RMSE) for an ensemble of process-based model runs (from the WOMBAT-lite model). These surrogate models are then used to (1) test sensitivity analysis of the WOMBAT-lite model and (2) optimize key WOMBAT-lite parameter values. Using the method the authors show the importance of key parameters, and illustrate significant model improvements, including for non-optimized metrics such as bloom phenology.

The authors have made significant improvements to the manuscript since the initial submission. However, due to 1) the intended audience of process-based biogeochemical modelers - which do no necessarily have statistical backgrounds and 2) the general novelty of the methods I would like to see a bit more discussion and a few more clarifications which are suggested in detail below.

**High-level description of method**

The surrogate modeling is novel and should be explained conceptually at a higher level. For instance, it is not initially clear that the surrogate model is used to predict model error - which can be confusing to researchers more familiar with surrogate models that predict the full model fields. While this detail is provided in the methods, stating this clearly in the overview (section 2.1) of the methods section, as well as in the conclusion, would aid in interpretation.

**Definition of priors**

The authors use uniform priors which are scaled to range between 0-1. However: 1) the scaling is not mentioned in the text and should mentioned, justified and explained; 2) how the priors were chosen is not clear - ideally references for each value should be provided in table 1; 3) the uniform priors seem to work nicely, but the downside and potential implications of a uniform prior on the posterior estimates should be discussed - in particular since the parameters are likely Gamma or Normally distributed in reality.

**Choice of surrogate model objective**

The authors train the surrogate models to predict RMSE of each parameter configuration. This is an elegant and cost effective approach, but the benefits of this approach should be highlighted and it's use justified in text. It would also be useful to highlight any downsides of this approach compared to a spatially-resolving emulator. For instance, it is conceivable that sensitivity of each parameter is not spatially uniform (e.g. parameters influencing predictive

performance in the Southern Ocean could be quite different from equatorial upwelling regions).

**Specific comments**

- L294: include the short name of the package (UQ-PyL)

- L296: consider defining this as rRMSE (relative RMSE) or NRMSE (as used further down in the text) and then using NRMRSE/rRMSE where appropriate for clarity

- L298: why was a K value of 8 chosen?

- L329: which hyperparameters were used for these kernels? Was any hyperparameter optimization conducted to find the best values?

- L342: how was convergence assessed? A plot illustrating chain convergence should be included, and if any test (e.g. Rhat) was used, these values should be reported.

- L538-545: these estimates should be compared to the literature (e.g. field et al., 1998, falkowski et al., 1998, johnson et al., 2021 - or others as relevant)

---

## Author Response (AR3)

**Overview**

*Thanks to the authors for their thorough response to my comments. The additional text in the revised manuscript addresses my concerns. In particular I find the additional figure S6 complementing very well the figure 7 to show how the interest of using the surrogate model to run an extensive optimisation. One remaining question to me: how the optimisation affect the drift of the model? If it could be shown that the optimisation reduce the drift that would be an additional strength of the method. Maybe by comparing the drifts in the best of the sensitivity experiments and the best of the optimal experiments, both run on 100 years?*

*Because we train the GPR model on the $10^{th}$ year of output from the full model (with coupled and evolving physics), we do not expect the selection of optimal parameters to counteract any long-term drift in the ocean state. We are therefore unable to show that the method reduces the long-term drift in the ocean nor do we necessarily expect it to, although we acknowledge that it may provide some guardrails against rapid deviations.*

*We have added the following sentence to clarify this:*
*Line 261: "Our optimisation therefore does not counteract the potential for long-term drift but does provide some guardrails against rapid deviations from the initial state with respect to dissolved nutrient and carbon fields. Optimisation of the biogeochemistry under these conditions can cause over-tuning of the biogeochemical parameter set that compensate for physical errors (e.g., Singh et al., 2025). However, by assessing the cost function after 100 years and selecting the best performing of the optimal experiments we likely select for a parameter set with minimal long-term drift."*

*In addition, I have a couple of minor and specific comments, some of which I did not catch during the previous review. Once these minor adjustments are made, the manuscript will be ready for publication.*

**Minor and specific comments**

**Abstract**

*(1) l. 27: I think "earth" in "...of earth system models..." is more commonly written with a capital E ("...of Earth system models...")*

*Corrected.*

**Methods**

*(2) l. 318-333: Is it one surrogate model trained for each cost function J (so 8 surrogate models in total) or one for the global cost function (equation 2)? Can you clarify? I had a similar question for the sensitivity analysis during the first round of review. The authors clarified very well, notably by using plural forms or emphasizing singular terms when appropriate.*

*We thank the reviewer for their thoroughness. We have clarified this in the text on line 346 by extending the sentence:*

*"Therefore, we aim to select parameter sets that optimise overall model performance"*

*To:*

*"Therefore, we select parameter sets that optimise overall model performance by summing the prediction errors from all eight surrogate GPR models."*

**Results**

*(3) l. 463-464: "The 512 experiments were used to calibrate the global cost function synthetically using the machine learning model." I do not think this sentence really state what was done. What I understood is that: the 512 experiment are used to trained the machine learning model to reproduce the global cost function. The*

*trained model is then used to look for the optimal values of the parameters, i.e. the ones minimizing the global cost function. It do not think we can say that the global cost function is calibrated. Can the authors rephrase?*

*Agreed. We have corrected this by including your interpretation within the text.*

*(4) l. 478-479: "We also note that the model predicted optimal values that often aligned well with ecological theory." Is the initial range of the parameters' values much wider than the theory? Otherwise it seems expected that the optimal ₃₁ values are aligned with the ecological theory and thus the argument is not very strong and could be removed.*

*Here, we refer to and immediately discuss the fact that the optimal values of two parameters were in a ratio that reflected theory and/or observational evidence. So, while we chose prior ranges that are legitimate, there are ratios of two parameters that are not, and it was encouraging to see that the optimal parameter set included parameter ratios that were in line with observations/theory.*

*(5) l. 531: "unoptimised biogeochemical model". I assume the authors mean the former version of the model that even if not optimised was considered good enough to conduct analysis and numerical experiment. Correct? This model was still tuned. Can the author specify this? Otherwise, it sounds like the new optimised version is compared to a version that should even not be used for any study.*

*Agreed. When referring to this prior version of the model, we have now introduced it as a prior model "that did not undergo the same optimisation process but was nonetheless tuned manually". This has been clarified in the text.*

*(6) Sec. 3.4.3: Is the bloom phenology better than with the unoptimised model version?*

*The previous version did not have prognostic chlorophyll concentrations, so a direct comparison was not possible.*

**Summary**

*(7) l. 666-668: The authors should mention here that the surrogate model can provide the large number of samples at a low computational cost.*

*We have followed the reviewer's suggestion. The sentence now reads:*

*Line 712:*
*"To do so, we used a surrogate machine learning model trained on a limited sample of real model output that provided many synthetic estimates of model performance (i.e., global NRMSE and the global cost function) at low computation cost."*

*(8) l. 677: Shouldn't it be "statistically" instead of "statically"?*

*Statically is correct here. We refer to the idea of having time-evolving, or even space-evolving, parameter values, which could potentially even better performance.*

*We have nonetheless altered the sentence to improve clarity:*

*Line 725:*
*"While these approaches can provide optimal parameter values that evolve in time and/or space, the surrogate approach employed herein represents a simple yet valid approach that provides globally optimised parameter values that are fixed in time and space, but importantly without large computation overhead."*

**Figures**

*(9) Fig. 5: I do not think emphasizing "at most" and "at least" with bold font is necessary.*

*Agreed. Corrected.*

**General comments**

The paper by Buchanan et al., provides an innovative and elegant case study for parameter sensitivity analysis and optimization of a ocean biogeochemical model using surrogate models and Bayesian methods. In the study the authors use Gaussian Processes (GPs) to predict model error (specifically, root mean square error - RMSE) for an ensemble of process-based model runs (from the WOMBAT-lite model). These surrogate models are then used to (1) test sensitivity analysis of the WOMBAT-lite model and (2) optimize key WOMBAT-lite parameter values. Using the method the authors show the importance of key parameters, and illustrate significant model improvements, including for nonoptimized metrics such as bloom phenology.

The authors have made significant improvements to the manuscript since the initial submission. However, due to 1) the intended audience of process-based biogeochemical modelers - which do no necessarily have statistical backgrounds and 2) the general novelty of the methods I would like to see a bit more discussion and a few more clarifications which are suggested in detail below.

**High-level description of method**

The surrogate modeling is novel and should be explained conceptually at a higher level. For instance, it is not initially clear that the surrogate model is used to predict model error - which can be confusing to researchers more familiar with surrogate models that predict the full model fields. While this detail is provided in the methods, stating this clearly in the overview (section 2.1) of the methods section, as well as in the conclusion, would aid in interpretation.

*The reviewer is right to ask for greater clarity. Following their request, we have included additional remarks within sections 2.1 and the conclusion that remind the reader that the surrogate GPR model is predicting the univariate, global statistics of model error.*

*Section 2.1; Lines 133 – 135:*
*"This computationally inexpensive surrogate was trained on hundreds of real simulations with WOMBAT-lite, predicted global univariate statistics of model error and was able to produce large samples of synthetic results that enabled sensitivity analysis and optimisation (Fig. 1)."*

*Summary; Lines 712 – 713:*
*"To do so, we used a surrogate machine learning model trained on a limited sample of real model output that provided many synthetic estimates of model performance (i.e., global NRMSE and the global cost function) at low computation cost."*

**Definition of priors**

The authors use uniform priors which are scaled to range between 0-1. However: 1) the scaling is not mentioned in the text and should mentioned, justified and explained; 2) how the priors were chosen is not clear - ideally references for each value should be provided in table 1; 3) the uniform priors seem to work nicely, but the downside and potential implications of a uniform prior on the posterior estimates should be discussed - in particular since the parameters are likely Gamma or Normally distributed in reality.

*We acknowledge the reviewers concerns and we believe we have addressed them completely:*

1) *We do normalize the prior distributions to vary between 0 and 1 via a Max-Min scaling, since we know the lower and upper bounds of each parameter. The reason for normalising the prior distribution is to ensure more reliable and efficient exploration of the high-dimensional parameter space during MCMC.*
2) *We have added the appropriate references that guide the a priori range choices in Table 1.*
3) *We want the optimisation to determine the posterior distribution from as generic a distribution as possible. However, we also note that the burn-in phase of the MCMC is designed to forfeit the uniform distribution and its effect on the posterior, so although we do assume a uniform as input to the MCMC procedure, the burn-in substantially reduces its effect on the posterior.*

*These concerns have been addressed in what is now written on lines 361 – 379:*

*''Bayesian optimisation enables iterative learning of optimal model parameters using observational data. Here, we apply uniform priors under the assumption that there is equal probability of the optimal values falling anywhere between what are known lower and upper bounds (Table 1). The priors are normalised from 0 to 1 via max-min scaling, and the normalized cost function value predicted by the GPR model serves as the likelihood function (Reddy et al., 2024a). Since computing marginal likelihood directly is often complex, Markov Chain Monte Carlo (MCMC) sampling is employed, which estimates the posterior distribution without explicit calculation of this constant (Issan et al., 2023). Normalisation of priors between 0 and 1 ensures more efficient exploration of the high-dimensional parameter space (i.e., better "mixing") and therefore more reliable convergence, while normalisation of the cost functions ensures equal weighting of each target field to the solution. Among MCMC methods, Affine invariant ensemble sampling, implemented using the "emcee" Python package (Foreman-Mackey et al., 2013), is selected for its efficient convergence properties. This method uses an ensemble of chains to simplify sampling from anisotropic distributions. Fifty walkers and a stretch move of two are applied, with the first 10,000 steps used as a burn-in phase to ensure convergence, followed by 90,000 additional steps to achieve stable posterior distribution estimates (Foreman-Mackey et al., 2013; Goodman and Weare, 2010). This substantial burn-in phase converts our uniform priors into distributions that look more like the posterior (Foreman-Mackey et al., 2013). In total, 90,000 steps for each of 50 walkers means a total of 4,500,000 random walks, and with a mean autocorrelation time of 690 steps, we achieve 6500 independent samples of the posterior. This number of samples, made possible by the efficiency of the surrogate GPR model, is more than sufficient for convergence. Convergence was also evident by a mean acceptance fraction of 0.22 (Foreman-Mackey et al., 2013), a Gelman-Rubin statistic of between 1.005 and 1.009 for each parameter and visual assessment of the chains (Fig. S6)"*

[Figure]

*Figure S6. Mean (black line) and range (pink shading) of sampling by all 50 walkers during the Markov Chain Monte Carlo (MCMC) optimisation of the 13 parameters. The blue dashed line demarcates the transition from the burn-in phase to the actual sampling used to construct the posterior. The x-axis is on a log10-scale, and the y axis is the same for all parameters because we normalised their ranges to 0-1 based on Max-Min scaling. A total of 100,000 steps is taken by the 50 workers.*

*As well as the new Table 1 with a new column "Reference".*

[revised manuscript text omitted]

[iii]Parameter space not explored.

**Choice of surrogate model objective**

*The authors train the surrogate models to predict RMSE of each parameter configuration. This is an elegant and cost effective approach, but the benefits of this approach should be highlighted and it's use justified in text. It would also be useful to highlight any downsides of this approach compared to a spatially-resolving emulator. For instance, it is conceivable that sensitivity of each parameter is not spatially uniform (e.g. parameters influencing predictive performance in the Southern Ocean could be quite different from equatorial upwelling regions).*

*Lines 312 – 313:*
*"NRMSE was chosen as a metric of performance because of our diversity of target fields each with different units, which required normalization to ensure that each contributed equally to an assessment of global performance."*

*A spatially resolving emulator was avoided in this first instance because we wanted to determine how the simplest possible approach performed. A more sophisticated emulator could be developed later.*

**Specific comments**

*• L294: include the short name of the package (UQ-PyL)*
*Addressed.*

*• L296: consider defining this as rRMSE (relative RMSE) or NRMSE (as used further down in the text) and then using NRMRSE/rRMSE where appropriate for clarity*
*Addressed.*

*• L298: why was a K value of 8 chosen?*
*Addressed.*

*Line 313: "K-fold cross-validation is used to evaluate the GPR model's accuracy and we chose 8 folds (K=8) to cleanly divide the 512 experiments and provide a balance between sufficient training (448) and testing data (64) (Reddy et al., 2024a)."*

*• L329: which hyperparameters were used for these kernels? Was any hyperparameter optimization conducted to find the best values?*
*Parameter and kernal choices were guided by previous work.*
*Line 348: "Using a composite kernel—constant, Matern, and white noise kernels with hyperparameters guided by previous work (Reddy et al., 2024a)—"*

*• L342: how was convergence assessed? A plot illustrating chain convergence should be included, and if any test (e.g. Rhat) was used, these values should be reported.*
*See answer above referencing Fig S6. We also note here that we do assess convergence and have now reported the Rhat and other statistics in the text.*

*• L538-545: these estimates should be compared to the literature (e.g. field et al., 1998, falkowski et al., 1998, johnson et al., 2021 - or others as relevant)*
*We make this comparison later in the Summary section on Line 739.*